# Design of a self-driven probiotic-CRISPR/Cas9 nanosystem for sono-immunometabolic cancer therapy

Jifeng Yu[1,5], Bangguo Zhou[2,3,4,5], Shen Zhang[2,3,4,5], Haohao Yin ®[1,2,3,4] ✉,
Liping Sun[2,3,4], Yinying Pu[2,3,4], Boyang Zhou[1], Yikang Sun[1], Xiaolong Li[1],
Yan Fang[2,3,4], Lifan Wang[1], Chongke Zhao[1], Dou Du[2,3,4], Yan Zhang[2,3,4] &
Huixiong Xu ®[1] ✉

Reprogramming the tumor immunosuppressive microenvironment is a promising strategy for improving tumor immunotherapy efficacy. The clustered regularly interspaced short palindromic repeat (CRISPR)/CRISPR-associated protein 9 system can be used to knockdown tumor immunosuppression-related genes. Therefore, here, a self-driven multifunctional delivery vector is constructed to efficiently deliver the CRISPR-Cas9 nanosystem for indoleamine 2,3-dioxygenase-1 (*IDO1*) knockdown in order to amplify immunogenic cell death (ICD) and then reverse tumor immunosuppression. *Lactobacillus rhamnosus GG* (LGG) is a self-driven safety probiotic that can penetrate the hypoxia tumor center, allowing efficient delivery of the CRISPR/Cas9 system to the tumor region. While LGG efficiently colonizes the tumor area, it also stimulates the organism to activate the immune system. The CRISPR/Cas9 nanosystem can generate abundant reactive oxygen species (ROS) under the ultrasound irradiation, resulting in ICD, while the produced ROS can induce endosomal/lysosomal rupture and then releasing Cas9/sgRNA to knock down the *IDO1* gene to lift immunosuppression. The system generates immune responses that effectively attack tumor cells in mice, contributing to the inhibition of tumor re-challenge in vivo. In addition, this strategy provides an immunological memory effect which offers protection against lung metastasis.

Immunotherapy has been revolutionizing cancer management by boosting protective immune responses to promote tumor regression. Immune checkpoint blockade therapies, such as those with anti-cytotoxic T lymphocyte-associated protein 4, anti-programmed death-1 or anti-programmed death ligand-1, have demonstrated brilliant clinical achievements in recent years[1-4]. However, only a small proportion of patients respond to current tumor immunotherapy, primarily because tumor, stromal and infiltrating immune cells, like bone marrow derived cells (BMDCs), regulatory T cells (Tregs) and M2 macrophage, create a highly complex tumor immunosuppressive microenvironment (TIME) that inhibits the response of solid tumors to immunotherapy[5-7]. In particular, immune metabolism has a critical role

[1]Department of Ultrasound, Zhongshan Hospital, Institute of Ultrasound in Medicine and Engineering, Fudan University, Shanghai 200032, P. R. China.
[2]Department of Medical Ultrasound, Shanghai Tenth People's Hospital, School of Medicine, Tongji University, Shanghai 200072, P. R. China. [3]Department of Medical Ultrasound, Shanghai Tenth Hospital, School of Clinical Medicine of Nanjing Medical University, Shanghai 200072, P. R. China. [4]Shanghai Engineering Research Center of Ultrasound Diagnosis and Treatment, Shanghai 200072, P. R. China. [5]These authors contributed equally: Jifeng Yu, Bangguo Zhou, Shen Zhang. ✉e-mail: yin.haohao@zs-hospital.sh.cn; xu.huixiong@zs-hospital.sh.cn

in regulating the immune cells responses, which involves multiple intracellular metabolic pathways, such as glycolysis, tricarboxylic acid cycle and amino acid metabolism[8]. Specially, amino acid metabolism such as tryptophan (Trp) and arginine, influence tumor progression, proliferation and differentiation of immune cells[9]. Indoleamine-2,3-dioxygenase-1 (*IDO1*) is an endogenous immunosuppressive mediator that can stimulate the accumulation of FOXP3+ Tregs and suppress T cells activity by depleting Trp in the microenvironment[10,11]. The Cancer Genome Atlas (TCGA) database analysis reveals that the expression level of *IDO1* is significantly upregulated in triple-negative breast cancer (TNBC) compared to the normal breast tissue[12]. Thus, *IDO1* is a potential immunotherapeutic target to reprogram the TIME by improving amino acid metabolism[13]. Nevertheless, small molecule inhibitors generally cannot provide durable responses due to the presence of drug resistance[14,15], and a phase III clinical trial of IDO inhibitor combination therapy was declared a failure[16]. Whereas the RNA interference technique commonly used to target gene therapy by suppressing gene expression, the technique often suffers from inadequate transfection efficiency and transient gene silencing[17,18]. Therefore, alternative approaches are urgently needed to interfere with amino acid metabolism in order to reshape the TIME for cancer immunotherapy.

The clustered regularly interspaced short palindromic repeats (CRISPR)/CRISPR-associated endonuclease protein 9 (Cas9) technology is an attractive gene editing tool in various systems for treating diseases by precisely reprogramming or activating the specific genes[19–23]. The guidance of a single-guide RNA (sgRNA) leads to double-stranded breaks (DSBs) in Cas9 initiation of site-specific genomic DNA sequences, and the endogenous repair of the target site can lead to gene disruption or template restoration[24–26]. To date, most CRISPR/Cas9 delivery systems have primarily relied on physical approaches or viral vectors, which limit the applicability and present significant immunological challenges[27,28]. Crucially, effective delivery of CRISPR/Cas9 systems into target tissues remains elusive because of many extracellular and intracellular barriers, which severely limit the adoption of CRISPR/Cas9 in vivo[29]. Recently, the integration of nanotechnology with microbial carriers has become a promising strategy which can effectively overcome various physiological constraints, such as tissue penetration, tumor hypoxia and blood-brain barriers, etc. through self-, light- and magnetic-driven characteristics to achieve enhanced anti-tumor effects[30–32]. *Lactobacillus* is a probiotics in both humans and animals, which has the characteristic to maintain microecological balance by inhibiting colonization, translocation, and infection of pathogens[33]. At the same time, *Lactobacillus* belongs to the family of parthenogenic anaerobic bacteria with hypoxia metabolic properties, which makes it have the ability to target solid tumors as a carrier[34]. In addition, it has been demonstrated that oral live *Lactobacillus rhamnosus GG* (LGG) in combination with immune checkpoint therapy can significantly increase the number of dendritic cells (DCs) that activate CD8+ T cells and be recruited into the tumor microenvironment, resulting in powerful inhibition of tumor growth, metastasis and recurrence[35]. Therefore, LGG is a promising application in tumor therapy not only as a carrier for nanomedicine delivery, but also for regulating tumor microenvironment to activate the immune system.

Herein we present a self-driven CRISPR/Cas9 nanosystem for comprehensive TIME modulation to suppress tumor growth and avoid lung metastasis and antagonize re-challenge (Fig. 1). Zeolitic imidazolinium framework (ZIF-8) is a metal-organic framework (MOF) with a large specific surface area, tailored pore size, pre-designed morphology, biocompatibility and controlled degradability that bring such materials closer to pharmaceutical and medical translation[36]. Hence, ZIF-8 (M) is used as an excellent non-viral CRISPR/Cas9 delivery vehicle for delivery of the sonosensitizer hematoporphyrin monomethyl ether (H) and CRISPR/Cas9 system (S), named as MHS. Subsequently, live

LGG is compounded with the CRISPR/Cas9 system, named LGG-MHS, where LGG-MHS still maintains LGG activity and the therapeutic properties of MHS. Finally, MHS is electrostatically adsorbed onto the surface of LGG after being magnetically agitated with it in PBS at room temperature. Utilizing the hypoxia targeting ability of LGG, the ultrasound (US)-controlled CRISPR/Cas9 gene editing system (MHS) is delivered to the hypoxia tumor core, thus promoting effective accumulation of MHS in tumors. Therefore, when LGG-MHS is enriched in the tumor hypoxic microenvironment, the decrease in pH value improves the release of MHS from LGG. The as-obtained CRISPR/Cas9 system generates reactive oxygen species (ROS) upon US triggering, which induces the release of tumor-associated antigens, immunogenic cell death of tumor cells and DCs maturation. In addition, ROS effectively disrupt the structure of the endosomal/lysosomal membrane, allowing Cas9/sgRNA to escape from the endosomal/lysosomal and transport to the nucleus for efficient *IDO1* knockdown, reducing Treg cells to cluster in the tumor microenvironment. The system generates powerful immune responses that effectively attack tumor cells in mice, contributing to the inhibition of tumor metastasis in vivo. In addition, this strategy provides a powerful immunological memory effect which offers protection against tumor re-challenge after elimination. In summary, the designed self-driven CRISPR/Cas9 nanosystem not only reprograms TIME from multiple pathways to activate the immune system against tumors, but also provides an example of microbial vector for CRISPR/Cas9 delivery, which is crucial for the further clinical applications of gene editing technology in vivo.

## Results

### Investigating the function of *IDO1* genes in vivo

At the beginning of this study, to investigate whether *IDO1* promotes 4T1 growth in vivo, we constructed stable overexpression of *IDO1* and interference with *IDO1* in 4T1 cell lines and injected into the left second breast pad of Balb/C mice to construct a tumor transplantation model. Tumors of mice with knockdown of *IDO1* in 4T1 cells exhibited significant growth inhibition compared to vector control group, suggesting that *IDO1* knockdown reduced tumor burden compared to the control group. Tumors from 4T1 cells stably overexpressing *IDO1* grew faster and exhibited greater tumor volume compared to vector control groups (Supplementary Fig. 1a). In addition, immunofluorescence staining of tumor sections and corresponding quantitative analysis showed lower levels of IDO protein expression in 4T1 tumors with knockdown *IDO1*, while 4T1 tumors overexpressing *IDO1* had higher levels of IDO protein expression (Supplementary Fig. 1b). In conclusion, the results indicate that overexpression of *IDO1* significantly promotes the development of breast cancer.

### Synthesis and characterizations of self-driven CRISPR/Cas9 nanosystem

To construct a self-driven CRISPR/Cas9 nanosystem, HMME was pre-dropped in a mechanically stirred dimethylimidazole solution prior to the dropwise of zinc nitrate in order to obtain ZIF-8 encapsulated with HMME, named as MH. Since Cas9 and sgRNA are composed in a set ratio (molar ratio = 1:1), Cas9/sgRNA as a system incubated with MH, called as MHS (Fig. 2a). Specifically, during the synthesis of ZIF-8, HMME was added dropwise to form MH through in situ encapsulation, and MH was incubated with CRISPR/Cas9 to produce MHS via the inherent dispersion force of ZIF-8 coupled with surface energy between substances adsorption of CRISPR/Cas9, and grafting of CRISPR/Cas9 by imidazole-like ligands provided by ZIF-8. Due to the pore size limitation, only a relatively small amount of CRISPR/Cas9 has been internalized in the mesopores larger than 10 nm of the MH, while most of it will be grafted on the surface of the MH. Different mass ratios of MH to Cas9/sgRNA were used to prepare MHS in order to achieve optimal Cas9/sgRNA loading efficiency, and the amount of sgRNA in the nanosystem was determined utilizing agarose gel

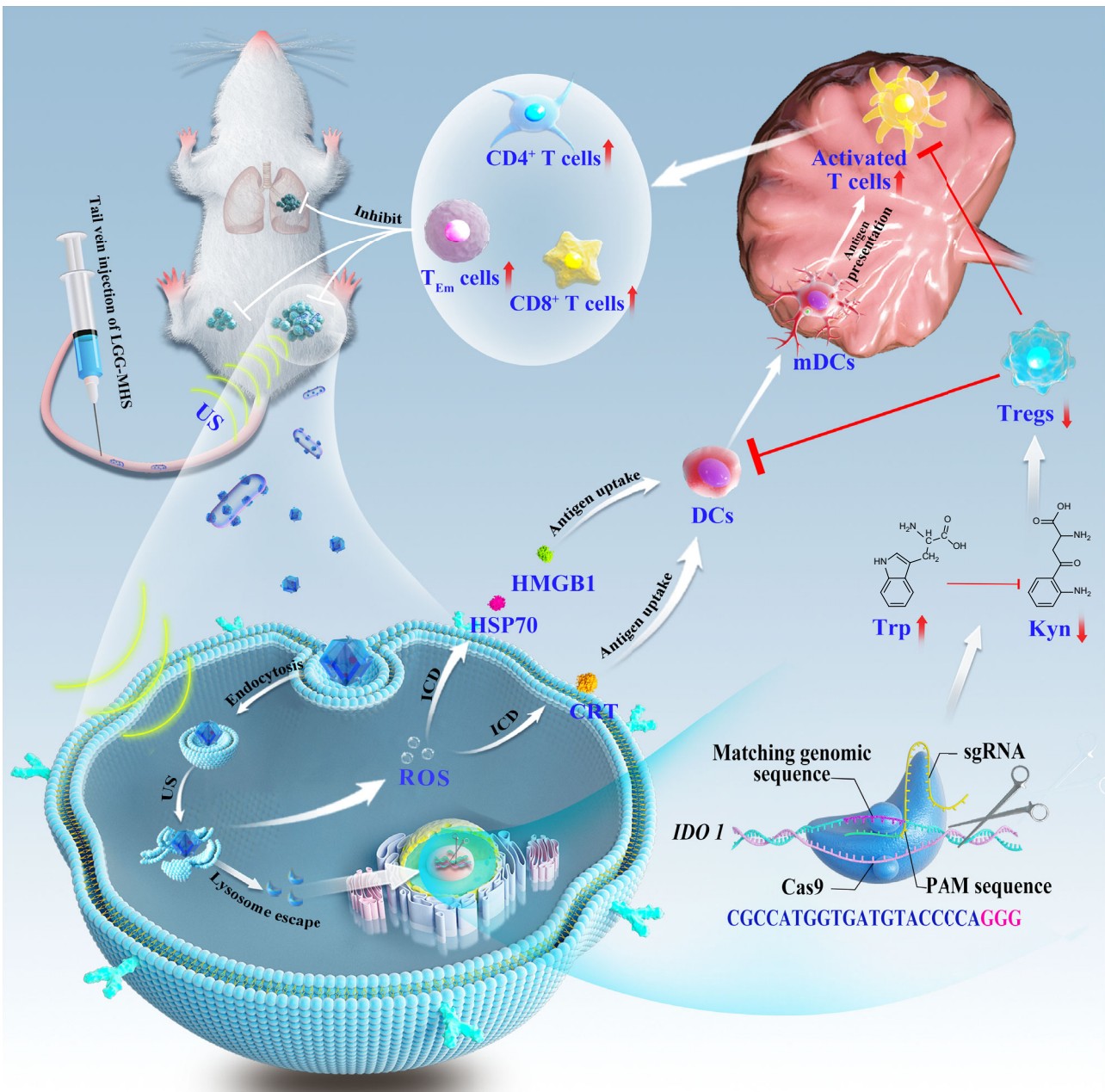

**Fig. 1 | Schematic of the LGG-MHS nanosystem delivery of CRISPR/Cas9 system for reprogrammed the TIME via activation of immune response.** The use of a US-triggered Cas9/sgRNA delivery system improved the efficiency of delivering Cas9/sgRNA to the nucleus of tumor cells for gene editing.

electrophoresis (AGE) (Supplementary Fig. 2a). The outcome shows that a ratio of 4 for MH: Cas9/sgRNA results in the optimal loading efficiency of Cas9/sgRNA. Simultaneously, nitrogen adsorption isotherms were utilized to confirm the successful loading of Cas9/sgRNA, showing a significant decrease in Brunauer Emmett Teller surface area and adsorption isotherms of MHS. Moreover, the total pore volume of MHS also showed a decrease relative to MH, demonstrating that part of the Cas9/sgRNA successfully entered the interior of ZIF-8 via permeation (Fig. 2b and Supplementary Fig. 2c). Following that, transmission electron microscopy (TEM) and Powder X-ray diffraction (PXRD) were used to examine the morphologies and structures of ZIF-8, MH and MHS, which showed no changes in nanoparticles morphology except for the slightly increase in particle size of MH and MHS compared to ZIF-8. The elemental profile corresponds to a denser P element within MHS than ZIF-8 and MH, which further suggests that Cas9/sgRNA and MH successfully formed a complex (Fig. 2c–e,

Supplementary Table 1). Dynamic light scattering (DLS) was used to determine the particle size of ZIF-8, MH, and MHS, which revealed that the average diameter of ZIF-8 is 79.43 nm. When HMME and Cas9/sgRNA loading succeed, the diameter of MH and MHS increased to 111.60 nm and 125.80 nm, respectively (Fig. 2f). Meanwhile, the average zeta potential of MH and MHS was 64.80 mV and 34.63 mV, respectively, which further indicates the successful loading of Cas9/sgRNA. (Fig. 2g).

To assess the stability of the CRISPR/Cas9 nanosystem (MHS), Cas9/sgRNA and MHS was incubated in the PBS containing serum (10% v/v) respectively for different durations (0 h, 6 h, 12 h, 24 h), followed by agarose gel and SDS-PAGE electrophoresis, to examine the stability of sgRNA and Cas9 protein in MHS. The results of sgRNA stability are shown in Supplementary Fig. 2d, e. The sgRNA with MH remained stable after 12 h. On the contrary, the free sgRNA was almost completely degraded, which further indicates that Cas9/sgRNA can

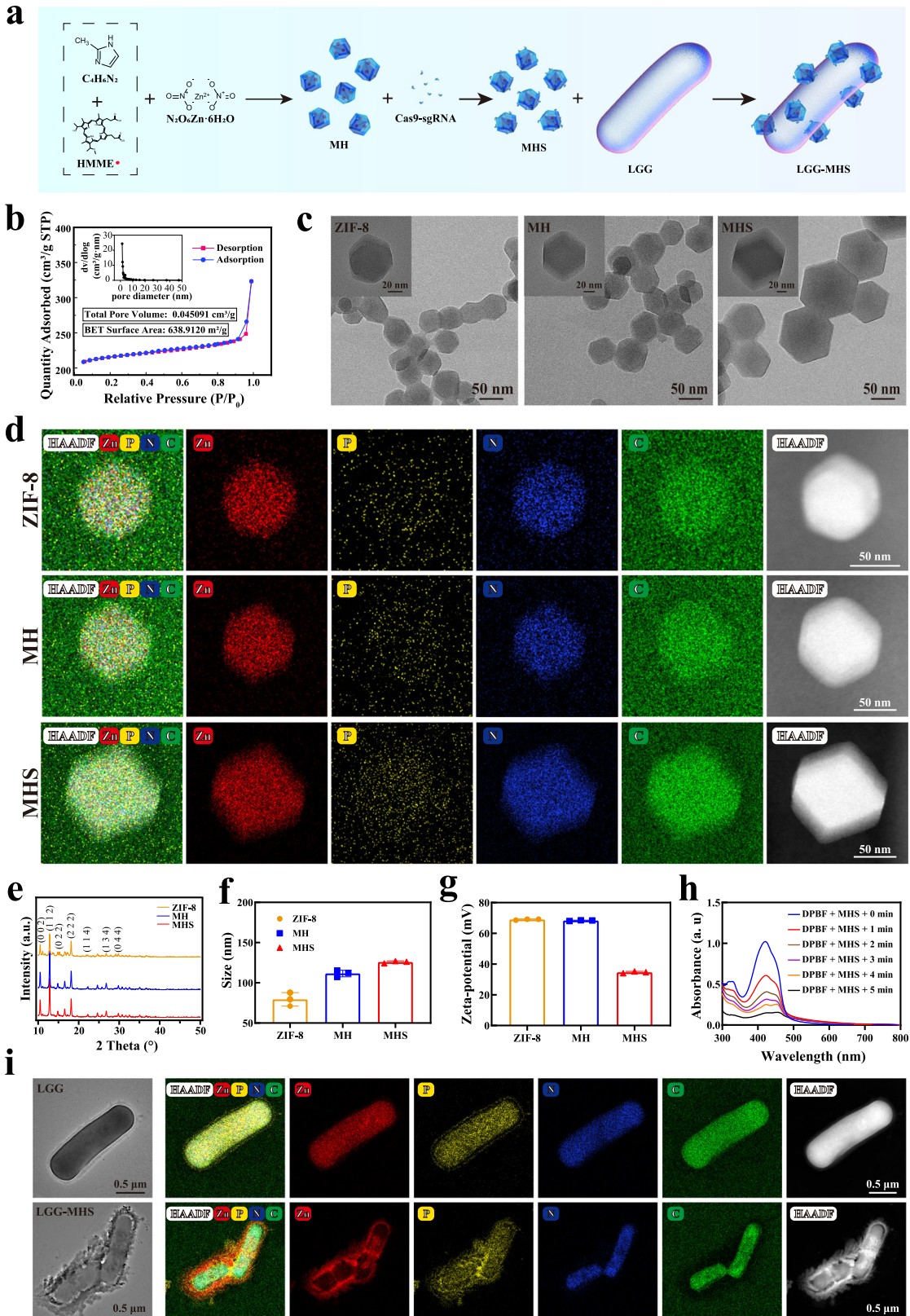

**Fig. 2 | Synthesis and structural characterization of ZIF-8, MH, MHS, LGG, and LGG-MHS. a** Synthesis procedure of LGG-MHS. **b** N$_2$ adsorption-desorption isotherms and of MHS. The inset shows its corresponding total pore volume and specific surface area. **c** Transmission electron microscopic (TEM) and **d** elemental mappings of ZIF-8, MH and MHS. **e** PXRD of ZIF-8, MH and MHS. **f** Particle size and **g** Zeta- potential of ZIF-8, MH, and MHS ($n = 3$ independent samples, data were expressed as means ± SD). **h** UV-vis absorption spectra of 1,3-diphenyliso-benzofuran (DPBF) in the presence of MHS upon prolonged US irradiation. **i** Transmission electron microscopic (TEM) and corresponding elemental mappings of LGG and LGG-MHS. The experiments for **b, c, d, e, h,** and **i** were repeated three times independently with similar results. Source data are provided as a Source Data file.

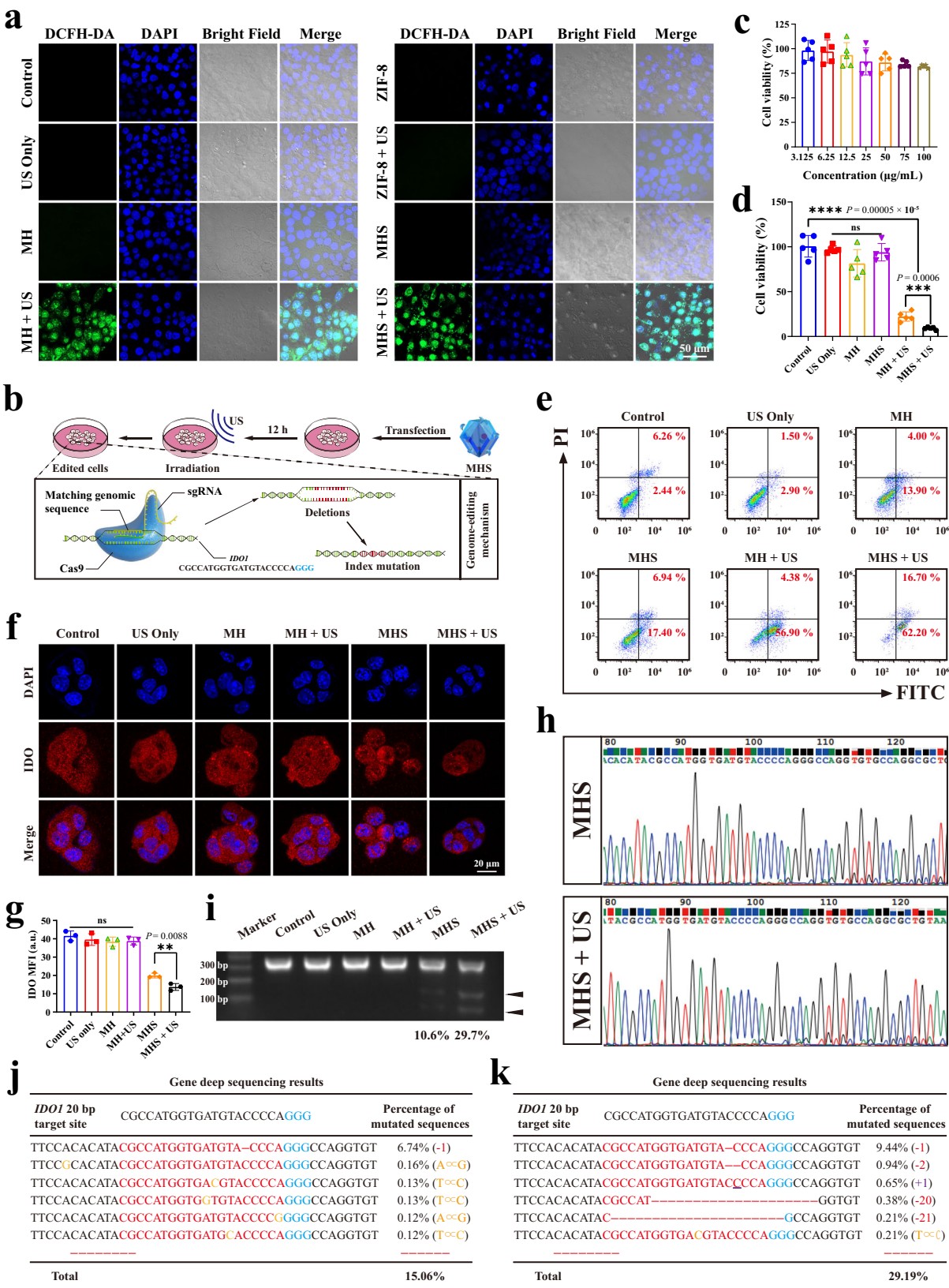

minimize degradation after being loaded by MH. And the stability of Cas9 protein was not affected by either naked Cas9/sgRNA or MHS (Supplementary Fig. 2f, g). In addition, the MHS nanosystem in the acidic environment (pH = 5.0) led to a greater Cas9/sgRNA release, which provides the foundation for effective release of Cas9/sgRNA in the acidic tumor microenvironment (Supplementary Fig. 2h). Next, 1,3-

diphenylisobenzofuran (DPBF) was employed as a single-linear oxygen ($^1O_2$) trapping agent for testing the capability to generate ROS of the MHS nanosystem under US irradiation. The results reveal that the absorbance of DPBF at 410 nm significantly decreases with increasing ultrasound exposure time, indicating that the MHS nanosystem efficiently produce $^1O_2$ under US irradiation (Fig. 2h and Supplementary

**Fig. 3 | Evaluation of US-associated *IDO1* genome editing in vitro.**
**a** Representative CLSM images of 4T1 cells with different treatments from three biologically independent samples. Concentration = 100 μg/mL. Incubation time = 12 h. **b** Illustration of transfection process of 4T1 cells by MHS upon US. **c** Toxicity evaluation in 4T1 after incubated with different concentrations of MHS and (**d**) Cell viability of 4T1 cells after various treatments for 24 h (*n* = 5 biologically independent samples). **e** Flow cytometry analysis of apoptosis of 4T1 cells with various treatments (*n* = 3 biologically independent samples). **f** Representative CLSM images and **g** corresponding mean fluorescence intensity of 4T1 cells treated with various treatments after IFNγ-stimulation, followed by staining with fluorescent anti-IDO antibody (red). DAPI was used to stain the nucleus of the cell (blue) (*n* = 3

biologically independent samples) **h** In vitro DNA sequencing of *IDO1* in 4T1 cells after treatment with MHS and MHS + US. **i** Representative image of T7EI cleavage analysis after 4T1 cells with different treatments. **j** Deep sequencing analysis of gene editing in 4T1 cells in the presence of MHS and (k) MHS + US. The experiments for **h**, **i**, **j**, and **k** were repeated three times independently with similar results. Statistical differences for **c**, **d**, and **g** were calculated using two-tailed unpaired Student's t-test for comparisons between two groups, ordinary one-way ANOVA for comparisons of more than two groups not containing Control, and Dunnett's multiple comparisons post test for comparisons of more than two groups containing Control. Data were expressed as means ± SD. *$P < 0.05$, **$P < 0.01$, ***$P < 0.001$, ****$P < 0.0001$. Source data are provided as a Source Data file.

Fig. 2i). Aiming to use the CRISPR/Cas9 nanosystem to actively target the tumor region, the MHS system was compounded with parthenogenic anaerobic LGG (LGG-MHS). Following this process, the average zeta potential of LGG-MHS dramatically decreased to −35.90 mV (Supplementary Fig. 2j). TEM results show that the LGG surface was not smooth with numerous nanoparticles attached after compounding. The corresponding elemental mapping reveals the presence of more Zn elements on the surface of LGG, which further implies that LGG was successfully compounded with the MHS nanosystem (Fig. 2i). Later, the effect of loading process on the activity of CRISPR/Cas9 nanosystem was investigated. Different states of Cas9/sgRNA were incubated with targeted DNA fragments, and agarose gel electrophoresis was performed. Quantitative analysis of the cut bands indicates that with the loading process or the application of US, the activity of Cas9/sgRNA is maintained at a high level, although a slight decrease occurs (Supplementary Fig. 2k, l). The coating of MRS agar plates was employed to detect the effect of MHS loading on the biological activity of LGG. As shown in Supplementary Fig. 2m, n, the effect on LGG activity was not statistically significant compared to the control group when the concentration of MHS was 2 mg/mL with stirring for 24 h, whereas the CFU decreased substantially when the concentration of MHS reached 4 mg/mL. The results indicate that the concentration of LGG-loaded MHS (1 mg/mL) in our strategy does not negatively affect the activity of LGG.

**Nanoparticle cellular uptake and the treatment effectiveness**
In order to thoroughly investigate the cellular absorption mechanism and confirm clathrin-mediated endocytosis, caveolae-mediated endocytosis, and micro-pinocytosis, three endocytosis inhibitors—sucrose, methyl-cyclodextrin (MβCD), and amiloride—were applied, respectively. The confocal laser scanning microscopy (CLSM) images show that endocytosis efficiency was decreased in cells pretreated with MβCD and amiloride, indicating that caveolae-dependent endocytosis were the primary routes for the endocytic uptake of MHS (Supplementary Fig. 3a, b). The ROS levels were compared in cells using different nanoparticles with or without US irradiation to confirm that the MHS nanosystem can generate ROS under US irradiation. The CLSM images show that the MH + US group and MHS + US group produce a large amount of ROS compared to other groups, demonstrating that the presence of HMME is one of the necessary components for ROS production (Fig. 3a and Supplementary Fig. 3c). On the other hand, ROS can destroy endosomal/lysosomal and improve gene editing efficiency. Z-stack CLSM was used to observe the location of Cas9/sgRNA in the organelle and found that the more cyanine 5.5 (Cy5.5)-labeled Cas9/sgRNA (red fluorescence) co-localized with the lysosome (green fluorescence) without US irradiation. Notably, under US irradiation, the Cy5.5-labeled red fluorescence signal was separated from the green fluorescence signal of lysosomes, while Cy5.5-labeled red fluorescence was detected in the nucleus, indicating that US irradiation is required for Cas9/sgRNA endosomal/lysosomal escape (Supplementary Fig. 3d). The endosomal/lysosomal escape associated with the MHS nanosystem was interfered with the protonation of the imidazole ring, which was

followed by the release of Cas9/sgRNA to the cytoplasm, and the US irradiation enhanced the process[37,38].

Subsequently, to explore the therapeutic efficacy and mechanism of the CRISPR/Cas9 nanosystem, murine breast cancer cells 4T1 were employed. Six groups were designed, which including Control, US only, MH, MH + US, MHS, and MHS + US, among which the processing of MHS + US group is shown in Fig. 3b. Initially, the biosafety of the MHS nanosystem was examined using standard cell counting kit-8 (CCK-8) assay. After 24 h of incubation with 4T1 cells, various concentrations of the MHS nanosystem exhibited negligible cytotoxicity (Fig. 3c). Then, the efficacy of different treatment regimens in 4T1 cells was measured after 24 h. The average lethal rate of tumor cells reached 91% (Fig. 3d), demonstrating the efficiency of the MHS nanosystem in killing tumor cells with the assistance of US. Simultaneously, flow cytometry and CLSM were used to evaluate the therapeutic efficacy of the MHS nanosystem with US irradiation. The apoptosis rate of 4T1 cells in the MHS + US group was 78.90%, which was significantly higher than other groups (Fig. 3e and Supplementary Fig. 3e). The MHS + US group exhibited a stronger red fluorescence single in propidium iodide (PI)-stained dead cells than other groups. For comparison, the control, US only, MH and MHS group showed weak red fluorescence (Supplementary Fig. 3f, g).

To investigate the gene editing efficacy of the MHS nanosystem under US irradiation, Cas9/sgRNA-mediated *IDO1* degradation was examined in 4T1 cells by employing immunofluorescence staining and Western blotting. As the results reveal that IDO protein expression levels were significantly reduced in the MHS and MHS + US group, indicating that Cas9/sgRNA effectively mediated the *IDO1* knockdown (Fig. 3f, g and Supplementary Fig. 3h, i). Then Sanger sequencing was used to analyze the gene-editing effect of the MHS nanosystem under US irradiation in vitro. The *IDO1* mutation peaks in the MHS and MHS + US group were higher than the groups without the CRISPR/Cas9 nanosystem treatment (Fig. 3h and Supplementary Fig. 3j). Later, fragment amplification of the target gene *IDO1* was performed by extracting genomic DNA from 4T1 cells after different treatments. After digestion of the amplified target gene using T7 endonuclease I, grayscale analysis for the target bands showed that the MHS + US group produced more cleavage products relative to the MHS group (Fig. 3i and Supplementary Fig. 3k). Subsequently, next-generation sequencing (NGS) was further performed to quantify the efficiency of the *IDO1* indel, revealing a genome disruption efficiency was 15.06% and 29.19% for the MHS and MHS + US group, respectively, compared with only 6.35% for the control group (Fig. 3j, k and Supplementary Fig. 4a). In addition, NGS reveals that the insertion and deletion mutation rates of the *IDO1* motif in the MHS + US group were 1.80% and 16.61%, respectively, while the deletion mutation rate of the *IDO1* motif in the MHS group was 7.79%, which further indicating that US-generated ROS disruption of the lysosomal membrane could significantly improve genome editing efficiency (Supplementary Fig. 4b). In conclusion, the above results further prove that the MHS nanosystem under US irradiation efficiently delivers the CRISPR/Cas9 system and performs target gene loci knockdown for the gene editing purposes.

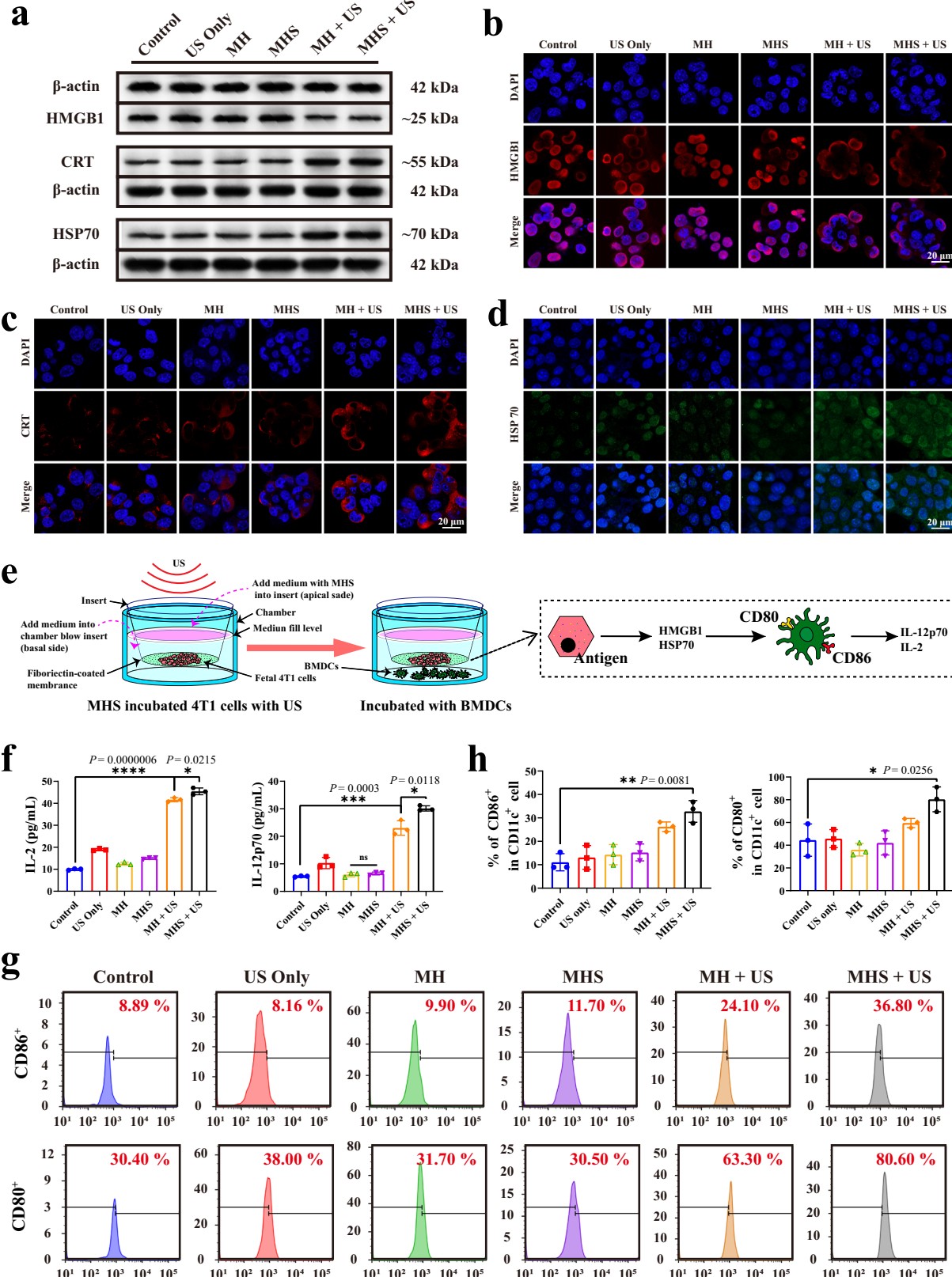

## In vitro exploration of sono-immunometabolic therapy

The potential ability of the MHS nanosystem to trigger ICD was then explored. The damage-associated molecular patterns (DAMPs) were detected, which included high-mobility group box 1 protein (HMGB1), calmodulin (CRT) and heat shock protein 70 (HSP70), the release of which is commonly considered to be the signature of ICD occurrence in tumor cells[39–41]. For this purpose, 4T1 cells were previously incubated with the MH or MHS nanosystem and then upon to US irradiation. From the Western blot assay, co-incubation of 4T1 cells with MH + US or MHS + US caused a decrease of HMGB1 band intensity and an increase of CRT and HSP70 band intensity (Fig. 4a and Supplementary Fig. 5a–c). This indicates that ICD was generated by the MH

**Fig. 4 | ICD facilitates antitumor immunity against 4T1 cells in vitro. a** Western blot analysis of specific proteins expression after DAMPs (HMGB1, CRT and HSP70). 4T1 cells were left untreated, treated with US only, co-incubated with MH, MHS, MH + US and MHS + US. Concentration = 100 μg/mL. Incubation time = 12 h (Four times each experiment was repeated independently with similar results). **b**–**d** Immunofluorescence analysis of specific proteins expression after DAMPs, including HMGB1 (red), CRT (red) and HSP70 (green). 4T1 cells were left untreated, treated with US only, co-incubated with MH, MHS, MH + US and MHS + US. DAPI was used to stain the nucleus of the cell (blue). **e** Schematic diagram of the experiment to explore DC cells maturation in vitro. In the upper chamber, 4T1 cells were cultured without any treatment, treated with US only, co-incubated with MH, MH + US, MHS and MHS + US. And in the lower chamber, BMDCs were cultured. After co-cultured for 24 h, BMDCs are collected for analysis. **f** Secretion of IL-12p70 and IL-2 from the supernatant of BMDCs ($n = 3$ biologically independent samples). **g** Representative flow cytometry plots and **h** corresponding statistical data of matured BMDCs (CD80⁺CD86⁺CD11c⁺) after various treatments, including control, US only, MH, MH + US, MHS and MHS + US ($n = 3$ biologically independent samples). A representative image or plot of three biologically independent samples from each group is shown in **b**, **c**, **d**, and **g**. Statistical differences for **f** and **h** were calculated using two-tailed unpaired Student's *t*-test, data were expressed as means ± SD. *$P < 0.05$, **$P < 0.01$, ***$P < 0.001$, ****$P < 0.0001$. Source data are provided as a Source Data file.

and MHS under the US trigger. These outcomes were further validated by immunofluorescence results, which have showed the similar trends with DAMPs generations. CLSM results show that the MH + US and MHS + US group markedly promoted the outflow of the HMGB1 protein from tumor cells (Fig. 4b and Supplementary Fig. 5d). In addition, the CLSM results for CRT protein changes demonstrate that the MH and MHS with US irradiation can produce significantly more CRT protein efflux on the cell membrane than other groups (Fig. 4c and Supplementary Fig. 5e). Moreover, the fluorescence intensity of HSP70 protein was significantly higher in the MH + US and MHS + US group than other groups (Fig. 4d and Supplementary Fig. 5f). These findings demonstrate that 4T1 cells co-incubated with the MH or MHS nanosystem under US irradiation could provoke the occurrence of ICD.

DCs are important antigen-presenting cells that act as a connection between the innate and adaptive immune systems. To promote maturation, immature DCs can phagocytize and remove antigens from injured tumor cells. These mature DCs migrate to lymph nodes (LNs) and present antigens to naive T cells there[42]. Based on the mechanism, we further evaluated the immunogenicity of 4T1 cells after co-incubation with the MHS nanosystem, and the transwell system was utilized to examine the ICD-induced DCs maturation in order to mimic these evolutionary processes (Fig. 4e). Juvenile mouse bone marrow dendritic cells (BMDCs) were obtained from mouse bone marrow for the study. The enzyme-linked immunosorbent assay (ELISA) revealed that the US-irradiated MHS nanosystem increased the expression of IL-12p70 and IL-2 by BMDCs (Fig. 4f). The presence of double staining with CD80 and CD86, which are the characteristic surface markers of matured DCs, confirming the maturation of BMDCs. Following incubation for 24 h, the MHS nanosystem significantly promoted DCs maturation under US irradiation, whereas US irradiation alone did not have a such significant effect (Fig. 4g, h and Supplementary Fig. 5g). IL-12p70 and IL-2 production coupled with an increase in the number of CD80⁺CD86⁺ T cells confirmed that the conditioned medium of MH + US or MHS + US treated 4T1 cells promoted the activation of immature DCs. These results reveal that the strategy of the MHS nanosystem under US irradiation holds great promise for inducing ICD and thus promoting immune cell infiltration in tumor tissues.

## Hypoxia targeting characterization and sequencing of LGG

It is essential to deliver the MHS nanosystem to the tumor site when performing CRISPR/Cas9-based oncogene therapy in vivo. Several anaerobic bacteria possess the ability to actively target and colonize the hypoxic microenvironment of tumors, among which LGG is one of the most distinctive and prevalent parthenogenic anaerobic probiotics[43,44]. To explore the tumor-targeting ability of LGG, the tumor tissues and major organs of 4T1 tumor-bearing mice were homogenized and smeared at different time points (0, 2, 6, 24, and 72 h) after injection of the LGG ($1 \times 10^8$ CFU per mouse). By counting the colony forming units (CFU) in each plate, 72 h after LGG injection, the bacteria were almost completely eliminated from the heart, spleen, kidneys, and lungs, although residual LGG remained in the liver. The amount of LGG was increased dramatically over time in tumors within 24 h after injection. Interestingly, LGG enrichment in the tumor was

higher than in the liver at 72 h with ~ 2-fold difference in CFU, which was attributed to the more favourable hypoxic microenvironment in the tumor for LGG proliferation, which further supports that LGG has relatively better hypoxic targeting and proliferative capacity (Supplementary Fig. 6a, b).

Subsequently, six 4T1 tumor-bearing mouse models were established, which were randomly divided into LGG groups and control groups. When the tumor volume reached 200 mm³, RNA sequencing was performed on the tumors in order to investigate the potential biological mechanisms of LGG to promote therapeutic efficacy. Correlation analysis shows that the two groups had similar gene expression levels (Supplementary Fig. 6c). Transcriptome sequencing results identify 160 genes were regulated, of which 106 were upregulated and 54 were downregulated in the LGG group ($P < 0.05$, log2FoldChange | > 1) (Fig. 5a, b). Analysis of these differential genes using gene ontology (GO) and Kyoto Encyclopedia of Genes and Genomes (KEGG) reveals that they are associated with multiple signaling pathways, which including immune infiltration of the tumor microenvironment (red) and promotion of tumor cell apoptosis (Fig. 5c and Supplementary Fig. 6d). The more representative evidence is the Toll-like receptor signaling pathway, where Toll-like receptors can be bound by exogenous and possibly endogenous ligands to trigger pro-inflammatory signaling cascades in various immune cells linking natural immunity and inflammation. Upon ligands bindings, TLR homo- or heterodimers are formed that activate MyD88-dependent and/or independent signaling pathways via different junctional proteins such as myeloid differentiation primary response 88 (MyD88), TIR domain-containing adaptor protein (TIRAP), translocation associated membrane protein (TRAM) and/or toll-like receptor adaptor molecule 1 (TICAM1). In the MyD88-dependent signaling pathway, MyD88 can recruit interleukin-1 receptor-associated kinases (IRAKs) and cause them to be phosphorylated, which then recruits tumor necrosis factor receptor-associated factors (TRAF) ubiquitin ligases. Typically, TRAF6 forms a complex with TGF−activated kinase (TAK) and TGF-beta-activated kinase1/MAP3K7 binding protein(TAB), and TAK activates the downstream IκB kinase-nuclear factor-κB (IKK-NF-κB) and mitogen-activated protein kinases (MAPK) cascades through phosphorylation modifications, which in turn leads to activation of the transcription factors nuclear factor-κB (NF-κB) and activating protein-1 (AP-1) and controls the expression of pro-inflammatory cytokines and other immune-related genes[45,46]. In addition, the TNF signaling pathway that causes apoptosis in tumor cells, is an exogenous pathway triggered by the cell surface death receptor tumor necrosis factor receptor family. The participating ligands and corresponding death receptors are TNF-α/TNFR1. The junctional protein (FADD/TRADD) recruits pro-caspase-8, leading to the formation of the death-inducing signaling complex, caspase-8 oligomerization, and activation by autocatalysis. The activated caspase-8 then induces apoptosis[47,48]. In summary, LGG may possess the ability to enhance the effect of immune therapy for tumor.

Accordingly, the combination of LGG with the MHS nanosystem to form LGG-MHS complexes has promised to amplify the oncological therapeutic efficacy by utilizing the self-driven effect of LGG for carrying the MHS nanosystem to the tumor sites. Notably, it has been shown that the acidic nature of the tumor microenvironment reduces

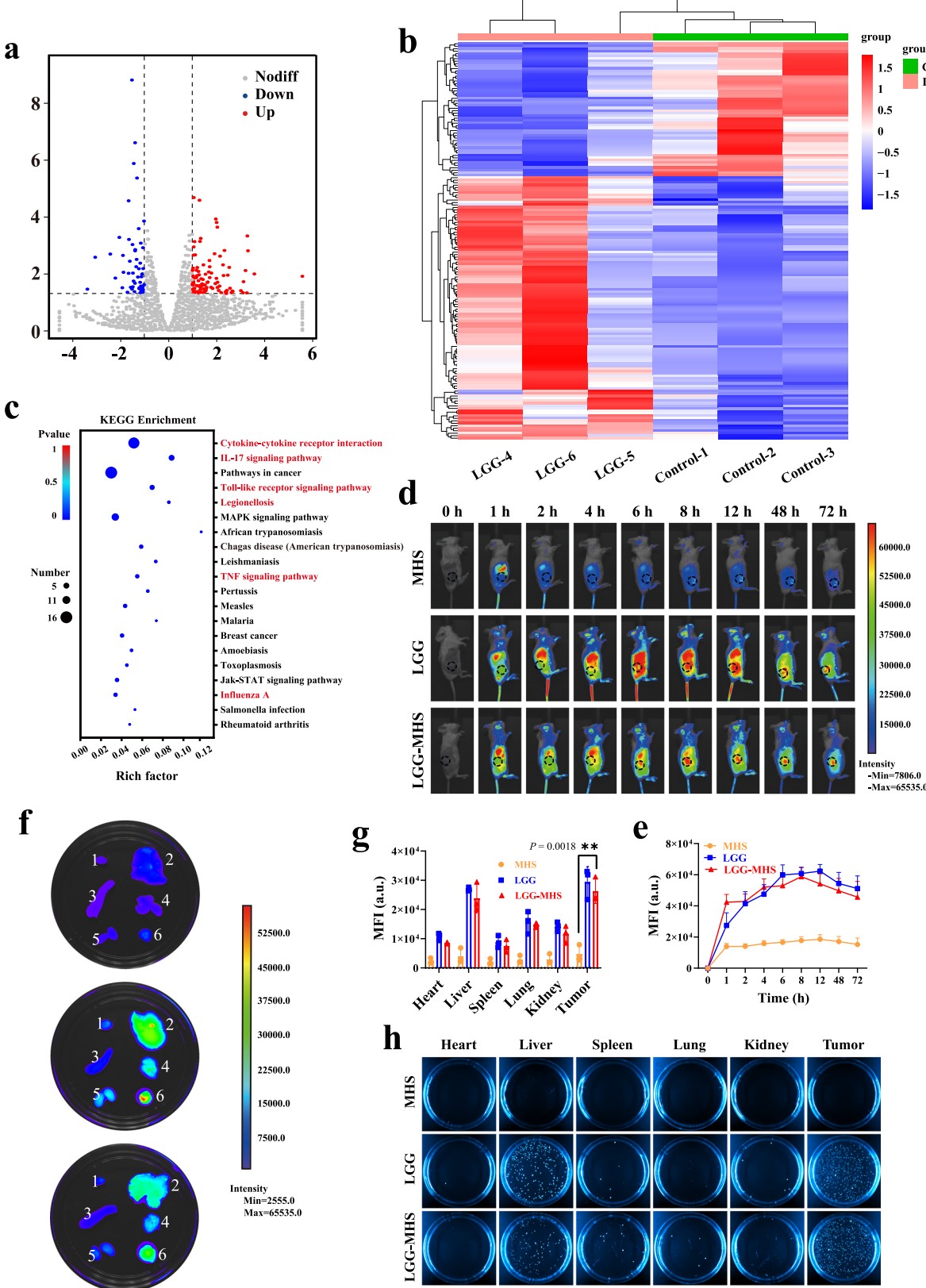

the forces between the drug molecule and the carrier material, such as electrostatic interaction, which facilitate the release of the drug. To examine the tumor tropism of the LGG-MHS, pre-prepared Cy5.5-labeled MHS (MHS-Cy5.5), Cy5.5-labeled LGG (LGG-Cy5.5) and both complexes (LGG-MHS-Cy5.5) were intravenously administered into tumor-bearing female Balb/c mice ($1 \times 10^7$ CFU per mouse), followed

by in vivo fluorescence imaging at different time points. In vivo fluorescence images and semi-quantitative analysis indicate that the fluorescent intensity of Cy5.5 at the tumor site increased over time after intravenous injection of LGG-Cy5.5 and LGG-MHS-Cy5.5, revealing that the LGG-MHS complex has relatively better tumor targeting properties (Fig. 5d, e). In addition, when tumor tissues and major

**Fig. 5 | Bacterial hypoxia targeting characterization and bacterial sequencing. a** Volcano map and **b** Heatmap of genes alteration with or without LGG treatment ($P < 0.05$, |fold change| ≥ 2). **c** RNAseq-based KEGG analysis of differential gene expression profiles after LGG treatment ($n = 3$ mice per group). Statistical difference was calculated using Fisher's exact test. **d** In vivo imaging and **e** corresponding fluorescence intensity of Cy5.5-labeled MHS, Cy5.5-labeled LGG and Cy5.5-labeled LGG-MHS in mice, respectively ($1 \times 10^7$ CFU LGG per mouse, $n = 3$ mice per group, data were expressed as means ± SD). **f** Accumulation and **g** corresponding mean

fluorescence intensity of Cy5.5-labeled MHS, Cy5.5-labeled LGG and Cy5.5-labeled LGG-MHS in major organs (1. Heart, 2. Liver, 3. Spleen, 4. Lung, 5. Kidney, 6. Tumor. $n = 3$ mice per group, data were expressed as means ± SD). Statistical differences were calculated using two-tailed unpaired Student's $t$-test. **h** Photographs of bacterial colonization in various organs harvested from 4T1-bearing mice at various time points after injection of MHS, LGG, and LGG-MHS on solid MRS agar plates (The representative imaging from three independent samples). Source data are provided as a Source Data file.

organs (heart, liver, spleen, lung, and kidney) were harvested for fluorescence imaging 72 h after injection of MHS-Cy5.5, LGG-Cy5.5, and LGG-MHS-Cy5.5, LGG-treated group and the LGG-MHS-treated group accumulated significantly stronger signals in tumors than the other organs (Fig. 5f, g). The collected tissues were homogenized and the obtained homogenates were t 100-fold dilution and split on MRS agar plates. The results suggest that no nascent LGG was found in the MHS group, while the bacteria mainly accumulated in the tumors in the LGG and LGG-MHS groups, demonstrating relatively good tumor-targeting properties of LGG-MHS (Fig. 5h and Supplementary Fig. 6e).

### Effectiveness of LGG-MHS in the treatment of tumors and anti-rechallenge in vivo

Initially, the biosafety of the LGG-MHS nanosystem was evaluated by analyzing the blood biochemistry and major organs of mice after tail vein injection of the LGG-MHS nanosystem for 30 days. The major organs were homogenized and spread after 100-fold dilution on MRS agar plates for incubation, which displayed that the heart, spleen, lung, and kidney were free of LGG growth except for minor LGG residues in the liver after 30 days of LGG-MHS nanosystem injection (Supplementary Fig. 7a, b). Subsequently, the major mouse organs (heart, liver, lung, and kidney) were extracted and stained for hematoxylin-eosin staining at different time points (1, 3, 7, and 30 days) following LGG-MHS injection, showing physiological morphology with no obvious necrosis or inflammation comparable to those of the pre-injection mice (Supplementary Fig. 7c). In addition, the biochemical indexes of blood (white blood cells, red blood cells, and platelets), liver (albumin, globulin, total protein, and total bilirubin) and kidney (creatinine and blood urea nitrogen) functions in the experimental mice were also not significantly different from those in the control group (Supplementary Fig. 7d, e). Furthermore, another US-imposed experiment was carried out to assess the biosafety of LGG-MHS under US stimulation. As shown in Supplementary Fig. 8, even mice injected with 2-fold of the treatment dose showed no abnormalities in hematological parameters and organ HE sections compared to untreated mice. The above results indicate that the LGG-MHS nanosystem has a high biosafety and biocompatibility.

Based on the results of the above in vitro studies, the therapeutic efficacy of LGG-MHS nanosystems in solid tumors and beneficial ICD induction were explored. A mouse 4T1 subcutaneous breast cancer tumor model was first constructed (Fig. 6a). To compare the merits between traditional small molecule inhibitors and emerging gene editing technologies, the IDO inhibitor NLG919 was included in the experimental group as a control group (LGG-MHI + US). Mice were randomly divided into 8 groups once the tumor volume reached an approximate size of 60-80 mm³, which including Control, LGG, MHS, LGG-MHS, MHS + US, LGG-MH + US, LGG-MHI + US and LGG-MHS + US, and were treated on days 7-14. The tumor growth was monitored every other day until day 21. As a result, the LGG-MHI + US group showed excellent ability to inhibit tumor growth and completely eliminated primary tumors in some mice (2/5) compared to the control group, while the LGG-MHS + US group had a more powerful treatment effect with tumor elimination reaching 4/5 (Fig. 6b, c). The findings suggest that reducing the expression level of *IDO1* can help inhibit the proliferation of tumor. More importantly, it was also demonstrated that the CRISPR/Cas9 nanosystem can precisely achieve *IDO1* knockdown

under US irradiation, avoiding the drug resistance defection of traditional inhibitors. The treated mice showed no significant change in body weight except for greatly prolonged survival, which further demonstrating the excellent biosafety and efficient therapeutic efficacy of the LGG-MHS + US nanosystem (Supplementary Fig. 9a, b).

To elucidate the mechanism of effective tumor growth inhibition by the LGG-MHS + US nanotherapy system, parts of the mice were euthanized and tumor tissues were collected at the time point of 24 h after the first US irradiation, followed by detection of Trp and Kyn levels in tumors. The Trp levels in the LGG-MHI + US and LGG-MHS + US group were increased relative to other groups, while the Kyn levels were reduced (Fig. 6d, e). In addition, HE and TUNEL staining were utilized to detect pathological changes in tumor tissue, while Ki-67 staining was employed to examine proliferation status of cells in tumor sections. The results of HE and TUNEL staining exhibit substantial cell necrosis in the MHS + US and LGG-MHS + US groups, which is particularly evident in the LGG-MHS + US group. Conversely, necrosis was either absent or minimal in control, LGG, MHS, and LGG-MHS groups (Supplementary Fig. 9c–e). In addition, Ki-67 staining shows a significant proliferation inhibition in the LGG-MHS + US group compared to other groups, which further demonstrates the high efficacy of the self-driven LGG-MHS nanosystem for tumor therapy (Fig. 6f). Subsequently, tumor sections immunostaining for DAMP-related proteins (HMGB1, CRT, and HSP70) was performed to explore the capability of the above treatments to induce ICD production. As the results indicate the related DAMPs proteins were highly expressed in the LGG-MHS + US group compared to the other groups, suggesting that the LGG-MHS nanosystem could effectively induce the production of ICD in vivo upon US irradiation (Fig. 6g, Supplementary Fig. 9f). Meanwhile, the obtained primary tumor sections were subjected to IDO immuno-fluorescence staining, which found that the IDO protein level was significantly decreased in the LGG-MHS + US group (Supplementary Fig. 9g, h). The above results demonstrate that the composite system consisting of LGG and MHS nanosystems (LGG-MHS) can accurately target and enrich the tumor site, with more powerful therapeutic effects under US radiation.

The mice were re-challenged on the time of 60 days with a 3-fold higher inoculum implanted subcutaneously into the left axillary to see if the survivors ($n_{LGG-MHI + US} = 2$, $n_{LGG-MHS + US} = 4$) had established long-term immunity against 4T1 cancer cells (Supplementary Fig. 9i). For the survivors that had been treated with LGG-MHS + US, the second tumor challenge was rejected at a 100% rate. Though animals treated with the LGG-MHI + US initially demonstrated 2/5 survival rate, all with tumor progression observed after the tumor re-challenge, indicating inefficient development of an adaptive immune response against 4T1 cells. These results show that, while IDO inhibitor combination LGG exhibits antitumor activity under the US exposure, it is not as efficient as the CRISPR/Cas9 at eliciting long-lasting immunity.

### Anti-tumor immune mechanism

To identify the mechanism underlying the superb antitumor effectiveness of LGG-MHS nanosystems under US radiation, the immune cells response in tumors and spleens, as well as the major immune cytokines in tumors were evaluated in treated mice. Given the importance of DCs in initiating innate and adaptive immunity, we first investigated whether the self-driven nanosystem therapeutic strategy

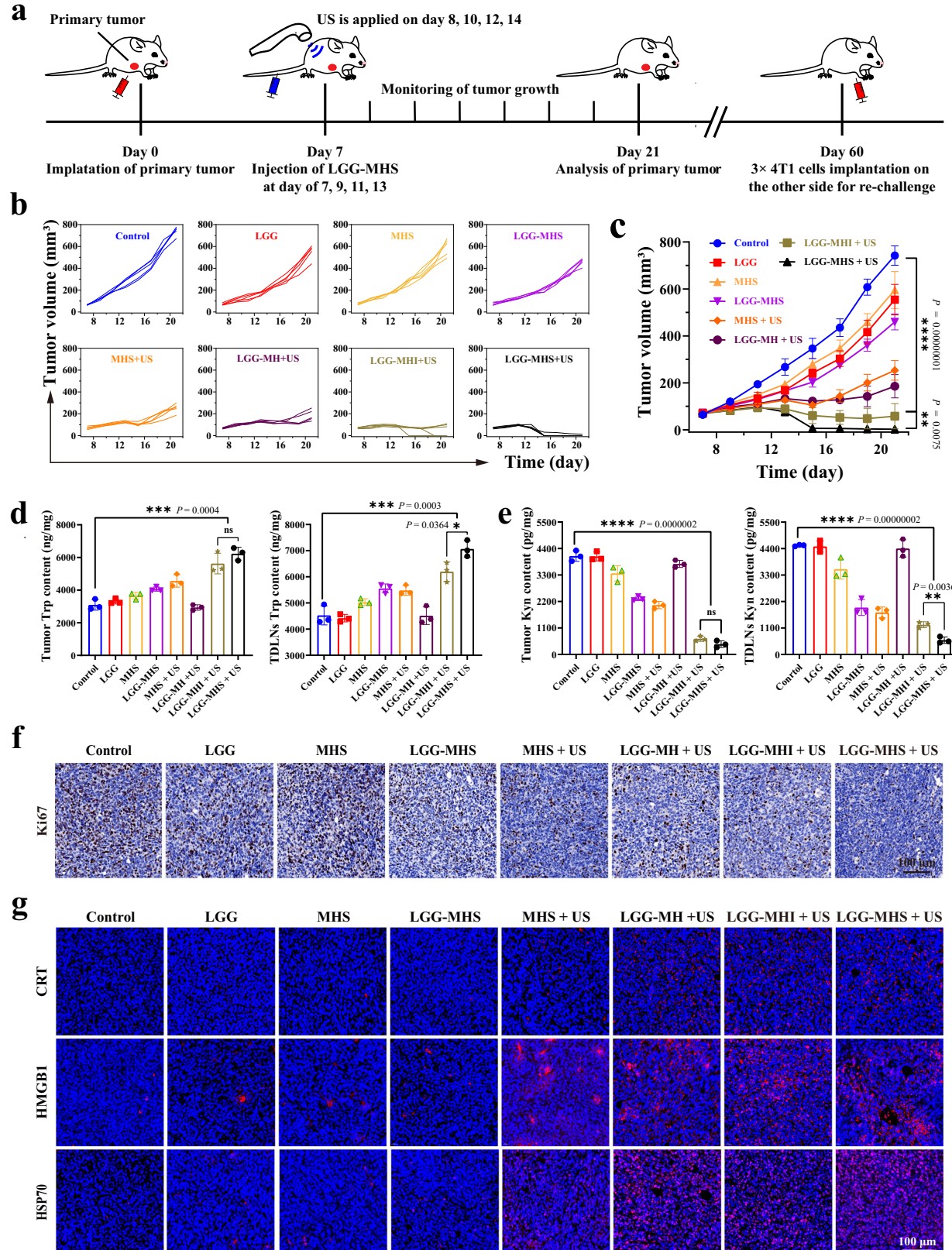

described above could promote DCs maturation in tumors. The results reveal that the proportions of mature DCs (CD11c⁺CD80⁺CD86⁺) in the Control, LGG, MHS, LGG-MHS, MHS + US, LGG-MH + US, LGG-MHI + US, and LGG-MHS + US groups were average ~10.1%, ~13.9%, ~13.3%, ~16.6%, ~19.7%, ~21.0%, ~25.5%, and ~29.7%, respectively, with LGG-MHS + US group levels being remarkably higher than other groups

(Fig. 7a and Supplementary Fig. 10a). Simultaneously, a population of individual CD8⁺ cytotoxic T lymphocytes (CTLs) and CD4⁺ helper T lymphocytes (HTLs) was detected in the spleen tissue. The results show that the populations of CD8⁺ T and CD4⁺ T cells in the mice spleen in the LGG-MHS + US group were significantly higher than other groups, indicating that self-driven probiotics play an essential role in

**Fig. 6 | LGG-MHS + US against 4T1 tumor in vivo. a** Schematic diagram of primary tumor treatment process in vivo. **b** Tumor growth curves of 4T1 after being treated by PBS, LGG, MHS, LGG-MHS, MHS + US, LGG-MH + US, LGG-MHI + US, and LGG-MHS + US (*n* = 5 mice per group). **c** Average tumor growth curves in different groups (*n* = 5 mice per group). Statistical differences were calculated using two-way ANOVA with the Geisser-Greehouse correction, match values are stacked into a subcolumn. **d** HPLC assay of the Trp content in primary tumors and TDLNs of tumor-bearing mice after different treatments (*n* = 3 biologically independent samples). **e** Elisa assay Kyn content in primary tumors and TDLNs of tumor-bearing mice after different treatments (*n* = 3 biologically independent samples). **f** Antigen Ki-67 staining in tumor sections from each experiment group. **g** Immunofluorescence images of specific proteins expression after DAMPs (HMGB1, CRT, and HSP70) from tumor tissue. DAPI was used to stain the nucleus of the cell (blue). A representative image of three biologically independent samples from each group is shown in **f** and **g**. Statistical differences for **d** and **e** were calculated using two-tailed unpaired Student's *t*-test between two groups, Dunnett's multiple comparisons post-test for comparisons more than two groups containing group Control. Data were expressed as means ± SD in **c**, **d**, and **e**. *$P < 0.05$, **$P < 0.01$, ***$P < 0.001$, ****$P < 0.0001$. Source data are provided as a Source Data file.

remodeling the immunosuppressive microenvironment following the delivery of the MHS nanosystem (Fig. 7b and Supplementary Fig. 10b). To further demonstrate that the LGG-MHS + US therapeutic strategy could reduce immunosuppression through improved CRISPR/Cas9 *IDO1* knockdown efficiency, the infiltration of regulatory CD3⁺CD4⁺Foxp3⁺ in tumors and overexpression of IDO resulting in accumulation of Tregs and thus blocking the anti-tumor immune CTLs response were investigated. The results show that the proportion of Tregs in the LGG-MHI + US and LGG-MHS + US group was significantly less than that in other groups (Fig. 7c and Supplementary Fig. 10c). In the spleen, the percentage of M1- and M2-like macrophages was determined simultaneously, showing that the ratio of M1/M2 was significantly higher in the LGG-MHS + US groups compared to the other groups, with the LGG-MHS + US group reaching an average 18.17 (Fig. 7d and Supplementary Fig. 10d). Immunofluorescence staining was used to assess the proliferation of CD4⁺ HTLs and CD8⁺ CTLs in order to further illustrate the anti-tumor immune response of the self-driven probiotic CRISPR/Cas9 delivery nanosystem strategy. As a result, the LGG-MHI + US and LGG-MHS + US therapy significantly facilitated the proliferation and activation of tumor-infiltrating HTLs (CD3⁺CD4⁺) and CTLs (CD3⁺CD8⁺), suggesting that LGG combined with IDO reduction strategy in the presence of US therapeutic arrangement modality can effectively activate the immune system to kill the tumor cells (Fig. 7e and Supplementary Fig. 11). Furthermore, intra-tumor levels of several vital biomarkers, including Interleukin-2 (IL-2), interleukin-12p70 (IL-12p70), tumor necrosis factor-α (TNF-α), and interferon-γ (IFN-γ), which are cytokines secreted by immune cells in TIME to promote T-cell responses, were considerably elevated in the LGG-MHS + US group (Fig. 7f–i). Collectively, these results confirm that a self-driven probiotic CRISRP/Cas9 delivery nanosystem therapeutic strategy enables a potential transformation of immune "cold" into immune "hot" tumors.

## Effectiveness in combating distal tumors and lung metastases
It is crucial to ensure that LGG can colonize distal tumors before the LGG-MHS self-driven nanosystem elicits systemic immune effects. Therefore, 4T1 cells were injected into the left side of the second breast pad of mice, and the same operation was performed on the right side 7 days later to establish an in situ dual tumor model as primary and distal tumors, respectively. When the primary tumor size reached approximately 200–250 mm³ and the distal tumor volume was approximately 60–80 mm³, LGG was injected via tail vein. After 24 h, the primary and distal tumors were harvested and homogenized for dish coating. As shown in Supplementary Fig. 12a, b, both primary and distal tumors showed LGG colonization. The difference in CFU may be due to the different levels of hypoxia in the primary and distal TIME.

Additionally, the dual tumor in situ model has been employed to investigate the against distal tumors of LGG-MHS + US therapeutic strategy. The necessary US irradiation was only performed on the primary tumor, but not on the distant tumor, detecting the size of the bilateral tumors during the treatment and analyzing the recorded data after 21 days (Fig. 8a). The tumor growth curves indicate that the primary tumors in the LGG-MHI + US and LGG-MHS + US group were distinctly suppressed compared to those in the control groups (Fig. 8b and Supplementary Fig. 12b), while distal tumors were significantly suppressed in LGG-MHS + US compared with LGG-MHI + US (Fig. 8c, d). At the end of therapy, the mice were euthanized, while bilateral tumor tissues were harvested and weighed, with results consistent with those described previously, demonstrating a favorable anti-neoplastic efficacy of LGG-MHS + US (Supplementary Fig. 12c, d). This against distal tumor mechanism can be attributed to the comprehensive activation of the immune system by the LGG-MHS + US therapeutic strategy, which was demonstrated by immunofluorescence to substantially increase the populations of infiltrating CD8⁺ and CD4⁺ T cells in distant tumor tissues (Fig. 8e and Supplementary Fig. 12e, f). This suggests that *IDO1* knockdown and US-induced ICD activation coupled with LGG-activated immune response pathways together mediated systemic anti-tumor immunity.

Lung metastasis from breast cancer is a major cause of death in patients[49], In order to investigate the anti-pulmonary metastatic effect produced by LGG-MHS + US treatment strategy, the model of lung metastasis were established. For prolong the survival of the tumor-bearing mice, the primary tumor was removed at the end of treatment, and 4T1 cells labeled with luciferase were injected into the tail vein of the mice 2 weeks later. The anti-lung metastasis effect of the different treatments was assessed at day 43 (Fig. 8f). The levels of central memory T cells (CD3⁺CD8⁺CD44⁺CD62L⁺, Tcm) and effector memory T cells (CD3⁺CD8⁺CD44⁺CD62L⁻, Tem) were measured by flow cytometry in the tumors of mice, which showed varying degrees of increase compared to the control group. A particularly pronounced increase was observed in the LGG-MHS + US group, with Tem numbers reaching an average of 18.7% and Tem numbers reaching an average of 38.7%. (Fig. 8g and Supplementary Fig. 12g, h). In addition, the pulmonary tissues were collected on day 42 and subjected to photography, bioluminescence imaging and HE staining, showing that the number of metastatic nodules in the lungs with LGG-MHS + US treated mice were significantly lower than those in the other groups (Fig. 8h–j and Supplementary Fig. 12i). Notably, MHS + US and LGG-MH + US, despite their powerful therapeutic effects in primary tumors, did not produce satisfactory systemic immune activation against distant tumors and lung metastases. These results confirm that the LGG-MHS + US therapeutic modality can facilitate the formation of T memory cells, leading to long-term anti-tumor effects to prevent lung metastasis.

## Discussion
Immunotherapy has become an effective therapeutic modality for tumors instead of surgery, radiotherapy, chemotherapy and targeted therapy through activation or modulation of the organism immune system. However, due to the existence of tumor immunosuppressive microenvironment (hypoxia, low pH, immunosuppressive cell infiltration, etc.) limits the effectiveness of immunotherapy. In particularly, IDO is a potential small molecule immune checkpoint which is overexpressed in a variety of tumor tissues and serves as an immunosuppressive factor to induce immune tolerance and immune escape in the organism's immune system. The Cancer Genome Atlas (TCGA) database analysis reveals that the expression level of *IDO1* is significantly upregulated in triple-negative breast cancer (TNBC) compared to normal

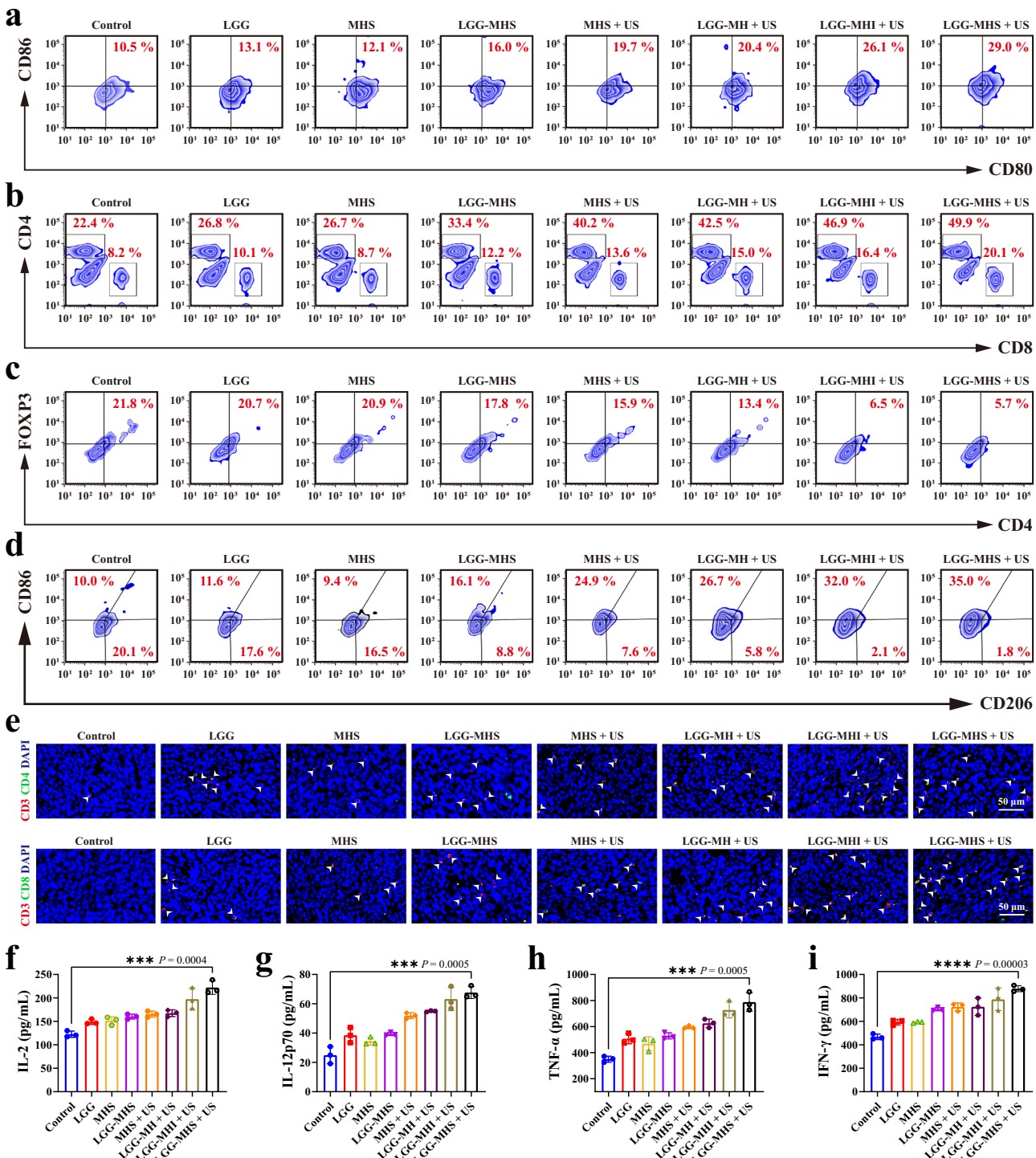

**Fig. 7 | Reprograming of the tumor immunosuppressive microenvironment by the self-driven LGG-MHS + US nanosystem. a** Typical flow cytometric of mature DCs in tumor tissue after 24 h after the first different treatments. **b** Typical flow cytometric of T cells of CD4⁺ and CD8⁺ T cells in the spleen after 24 h after the first different treatments. **c** Typical flow cytometric of Tregs in primary tumor tissue after 24 h after the first different treatments. **d** Representative flow cytometric of M2 macrophages in spleen after 24 h after the first different treatments. **e** Immunofluorescence images of helper T lymphocytes (CD3⁺CD4⁺) and

proliferated cytotoxic T lymphocytes (CD3⁺CD8⁺) in primary 4T1 tumor tissue slices. **f–i** Levels of the IL-2, IL-12p70, IFN-α, and TNF-γ in primary tumor tissues after 24 h after the first different treatments ($n = 3$ biologically independent samples). Representative plots or images of three biologically independent samples from each group is shown in **a**, **b**, **c**, **d**, and **e**. Statistical differences for **f**, **g**, **h**, and **i** were calculated using two-tailed unpaired Student's $t$-test, data were expressed as means ± SD. *$P < 0.05$, **$P < 0.01$, ***$P < 0.001$, ****$P < 0.0001$. Source data are provided as a Source Data file.

breast tissue. Then, to explore the correlation between the expression level of *IDO1* and the development of TNBC, we constructed stable overexpression of *IDO1* and stable interference with *IDO1* in 4T1 cell lines and constructed xenograft tumor models. The results indicate that

overexpression of *IDO1* significantly promotes the development of breast cancer. Therefore, reducing the expression level of *IDO1* contributed to inhibiting the proliferation of breast cancer. Accordingly, the CRISPR/Cas9 nanosystem developed in this research can efficiently

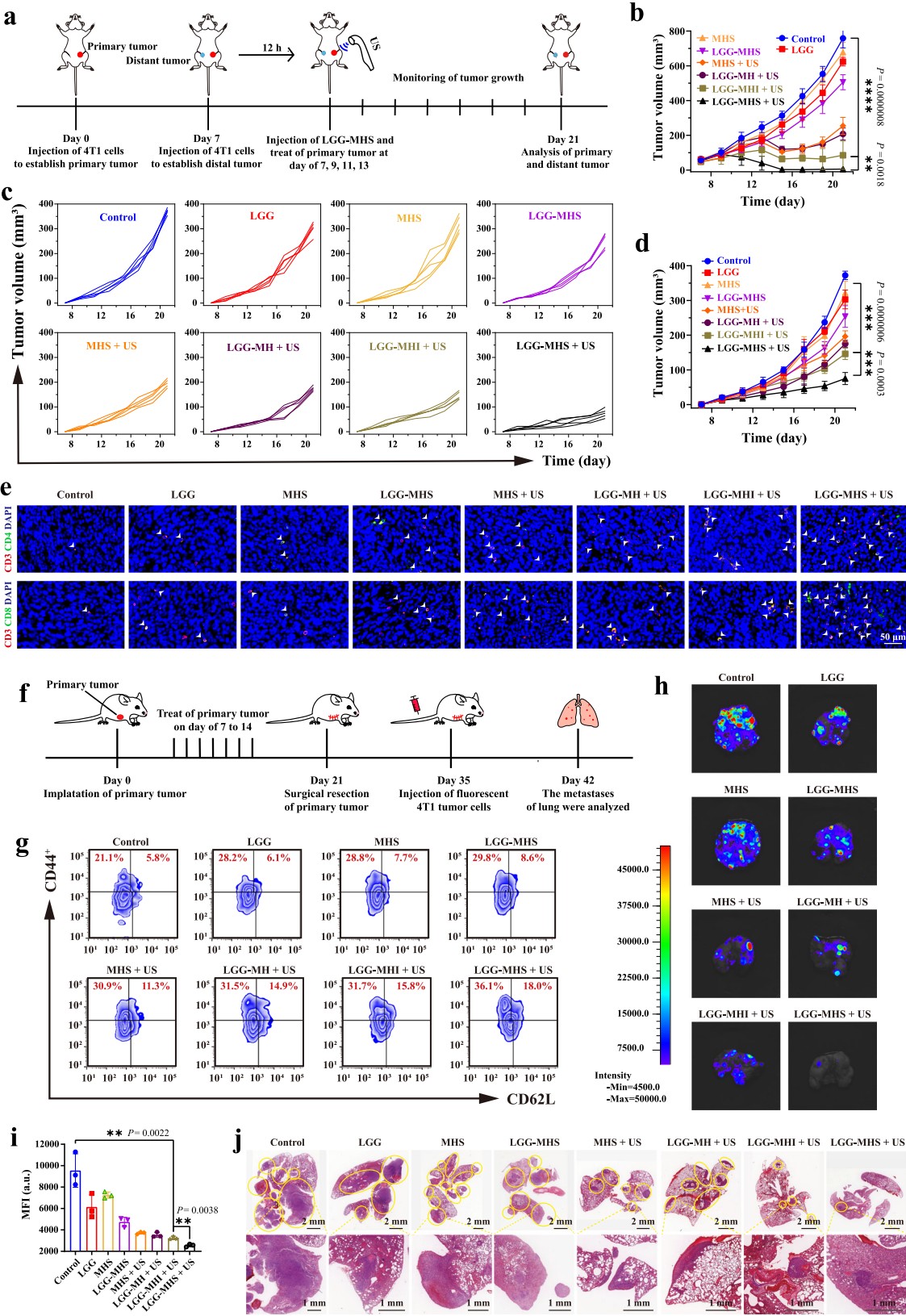

enrich the tumor region under probiotic drive and can precisely and controllably knock down *IDO1* under US irradiation, avoiding the lack of targeting and drug resistance of traditional inhibitors.

In addition, hypoxia plays a crucial role in the tumor immunosuppressive microenvironment and largely influences the outcome of treatment. Given the critical role of hypoxia in tumor progression and

its resistance to treatment, many efforts have been made to overcome the limitations associated with hypoxia regarding tumors. In contrast to traditional strategies of overcoming hypoxia, the present research exploited the hypoxic microenvironment of tumors and utilized the hypoxia-driven and colonization properties of LGG as a vector for delivery of the CRISPR/Cas9 nanosystem. After our study, we found

**Fig. 8 | Anti distal tumor effect and immunological memory of LGG-MHS + US in the 4T1 bearing mice model. a** Schematic diagram of the establishment of distal tumors model and the experimental procedure of treatment. **b** Average tumor growth curves of primary tumor in different groups (*n* = 5 mice per group). **c** Mean growth curves and **d** corresponding growth curves of distant tumors in different groups (*n* = 5 mice per group). **e** Immunofluorescence images of helper T lymphocytes (CD3⁺CD4⁺) and proliferated cytotoxic T lymphocytes (CD3⁺CD8⁺) in 4T1 tumor tissue slices of distal tumor. **f** Schematic diagram of the establishment and treatment process of mouse models of lung metastasis. **g** Typical flow cytometric of the effector memory T cells (CD3⁺CD8⁺CD44⁺CD62L⁻) (Tem) and (CD3⁺CD8⁺CD44⁺CD62L⁺) (Tcm) in the spleen after 24 h after the first different treatments. **h** Bioluminescence images and **i** corresponding fluorescence intensity quantification of lung metastatic nodules of the 4T1 tumors (*n* = 3 biologically independent samples). **j** HE staining of lung tissue from different groups of 4T1 tumor-bearing mice. The nodules with yellow circles in the section diagram indicate metastases in the lungs. Representative images or plots of three biologically independent samples from each group is shown in **e**, **g**, **h**, and **j**. Statistical differences for **b** and **d** were calculated using two-way ANOVA with the Geisser-Greehouse correction, match values are stacked into a subcolumn, and statistical differences in **i** were calculated using a two-tailed unpaired Student's *t* test. Data were expressed as means ± SD in **b**, **d**, and **i**. *$P < 0.05$, **$P < 0.01$, ***$P < 0.001$, ****$P < 0.0001$. Source data are provided as a Source Data file.

that LGG have an ability to target the hypoxic microenvironment of tumors. In vivo fluorescence images and semi-quantitative analysis indicate that the fluorescent intensity of Cy5.5 at the tumor site increased over time after intravenous injection of LGG-Cy.5.5 and LGG-MHS-Cy5.5, revealing relatively better tumor-targeting properties of the LGG-MHS complex. Meanwhile, it has been revealed that LGG is not only a vehicle but also a synergistic therapeutic adjuvant. LGG can inhibit tumor cell growth and metastasis by activating the immune response through certain specific pathways and increasing the infiltration of immune cells in the tumor microenvironment.

The system generates powerful immune responses that effectively attack tumor cells in mice, contributing to the inhibition of tumor metastasis in vivo. In addition, this strategy provides a powerful immunological memory effect which offers protection against tumor re-challenge after elimination. In summary, a self-driven probiotic delivery system for CRISPR/Cas9 was constructed in order to reprogram the TIME and then inhibit metastasis and recurrence of breast cancer. This system employs Lactobacillus rhamnosus as a carrier for the efficient delivery of the CRISPR/Cas9 nanosystem to knock down *IDO1*, reduce immunosuppressive cells infiltration, and activate intrinsic immunity by regulating signaling pathways associated with immune response and apoptosis. Meanwhile, the system is triggered by the US to improve gene editing efficiency and induce ICD, while the molecular damage-related proteins released during ICD are taken up by immature DCs as antigens to promote their maturation and thus upregulation of killer T cells. Immune cells are efficiently activated through this cocktail therapy to eliminate the primary tumor and inhibit its metastasis and recurrence. This research not only reprogram the TIME with multiple pathways to activate the immune system against tumors but also developed a synergistic gene editing therapeutic modality based on a unique CRISPR/Cas9 gene delivery technology, which is undoubtedly crucial for further clinical applications of gene editing technology in vivo.

## Methods

### Ethical issues
All animal experiments were performed according to protocols in accordance with policies of the National Ministry of Health and approved by the Laboratory Animal Center of Shanghai Tenth People's Hospital (SHDSYY-2021-6393). The maximum tumor volume allowed by the Experimental Animal Center of Shanghai Tenth People's Hospital is 1500 mm³, the maximum tumor diameter allowed by the NIH is no to exceed 2 cm (Guidelines for endpoints in animal study proposals). The tumor volume or diameter of the mice in the tumor-related animal experiments in this study did not exceed the allowed range described above.

### Materials
2-methylimidazole (C₄H₆N₂, 98%), and zinc nitrate hexahydrate (N₂O₆Zn·H₂O, AR, 99%) were purchased from Aladdin (Shanghai, China). Methanol was acquired from Shanghai Lingfeng Chemical Reagent Co., Ltd. Monomethorphyrin (HMME) was purchased from Xi'an Ruixi biological technology Co., Ltd. 1,3-diphenylisobenzofuran (DPBF) was gained from Sigma-Aldrich Co. Ltd., USA. SDS-PAGE Gel Rapid Preparation Kit, 2,7-dichlorodihydrofluorescein diacetate (DCFH-DA), QuickBlock™ blocking agent and transferred to polyvinylidene difluoride (PVDF) membrane were acquired from Beyotime Co. Ltd., China. 2× Rapid Taq Master Mix, ChamQ SYBR qPCR Master Mix, T7 Endonuclease I, Ultra GelRed nucleic acid stain (10,000×), BCA protein quantification kit, 100 bp DNA Ladder, RNA-easy isolation reagent, 2× Phanta Max Master Mix (Dye Plus), FastPure cell/tissue total RNA isolation kit V2 and HiScript II 1st strand cDNA synthesis kit were acquired from Vazyme Co., Ltd. Dimethyl sulfoxide (DMSO) was acquired from Macklin (Shanghai, China). Pierce™ ECL Western Blotting Substrate was purchase from Thermo Fisher Co., Ltd. Mouse IL-2, IL-12p70 ELISA kits were acquired from Multiscience, China.

### Characterization
The specific surface area and total pore volume of the composite nanoparticles were measured by nitrogen adsorption technique in a Micrometitics ASAP 2460 system at −195.850 °C. TEM and corresponding element mapping were adopted for the observation of the microstructure and component of nanosystem (FEI Tecnai G2 F20 operated at 200 kV and JEOL). Particle size and zeta potential were determined by Zetasizer Nanoseries (ZTS1240, Malvern Panalytical Ltd.). UV–vis spectra was acquired by UV-vis-NIR spectrophotometer (PE Lambda 950). Nikon bio-microscope (CI-L) was used to observe the fluorescent sections.

### Construction of sgRNA
The target of *IDO1* gene sequence (CGCCATGGTGATGTACCCCAGGG) was designed with the assistance of the online tool platform (http://chopchop.cbu.uib.no/). SgRNA (104618019) and CRISPR-Cas9 (Cat#: 1081059) system were obtained from Integrated DNA Technologies (IDT).

### Construction of *IDO1* gene overexpression recombinant plasmid and *IDO1* Interfering lentiviral plasmid construction
Overexpression recombinant plasmid ID: CMV-MCS-3XFlag-PGK-Puro. Oligomeric single-stranded DNA sequences 5′ to 3′:

Oligo P1 (XhoI): CTACCGGACTCAGATCTCGAGGCCACCATGGCACTCAGTAAAATATCTCC.

Oligo P2 (EcoRI): GTCATCCTTGTAATCGAATTCAGGCCAACTCAGAAGAGCTTTC

Interfering lentiviral plasmid ID: hU6-MCS-CMV-ZsGreen 1-PGK-Puro. Oligomeric single-stranded DNA sequences 5′ to 3′:

Top Strand A: GATCCGCCTCCTATTCTGTCTTATTTCAAGAGAATAAGACAGAATAGGAGGCTTTTTTG.

Bottom Strand A: AATTCAAAAAAGCCTCCTATTCTGTCTTATTCTCTTGAAATAAGACAGAATAGGAGGCG.

### Bacteria culture
*Lactobacillus rhamnosus GG* (LGG) was dipped and inoculated in culture tubes containing De Man Rogosa Sharpe (MRS) broth with an inoculation loop. The culture tubes were placed in oscillation incubator at 37 °C. LGG was collected by centrifugation when it grew to

logarithmic phase (4 °C, 2500 × $g$, 5 min). LGG was washed three times with PBS and resuspended to $1 \times 10^7$ CFU/mL for later use.

## Agarose gel electrophoresis test

The MHS composite nanosystem were prepared at MH/sgRNA mass ratio of 0:1, 2:1, 4:1, 8:1, 10:1 and 12:1 (MH: Cas9/sgRNA). Then, the MHS was adjusted to a mass ratio equal to the concentration, loaded into 2% (w/v) agarose gel in triacetate-EDTA (TAE) buffer and run 25 min at 80 V. The gels were visualized by UV illuminator with a Gel Doc system (Bio-Rad).

## Evaluation of Cas9/sgRNA release from MHS in vitro

Two dialysis bags were filled with MHS and MH nanoparticles (100 μg/mL), respectively, and the bags were placed in PBS (pH = 5.0 and 7.4). Then the MHS solution was incubated at 37 °C with US irradiation. After exposure to US irradiation (1.0 MHz, 1.0 W/cm², 50% duty cycle, 5 min), nucleic acid in the above solution was detected at different time points (0, 0.5, 1, 2, 4, 8, 12, and 24 h) via micro-Spectrophotometer (Nano-300, Hangzhou All-sheng Instrument Co., Ltd.).

## Synthesis of self-driven CRISPR/Cas9 nanosystem

2-Methylimidazole (1.910 g) and zinc nitrate solution (1.314 g) were dissolved in methanol (20 mL), respectively. Hematoporphyrin monomethyl ether (HMME, 200 μL, 2 mg/mL) was slowly added to 2-methylimidazole solution under mechanical stirring at room temperature, and after 10 min, zinc nitrate solution was added dropwise. The MH was obtained after stirring for 24 h at room temperature. Then, the MH and CRISPR/Cas9 system (mass ratio 4:1) were incubated at 37 °C according to the methodology instructions, finally, the integration of MHS nanosystem was constructed. The obtained product was gathered by centrifugation and washed with ddH₂O for three times to remove the residuum. MHS was further stirred with LGG (PBS = 1 mL, LGG = $1 \times 10^7$ CFU, MHS = 1 mg) in PBS for 24 h to arrangement LGG-MHS.

## Evaluation of biological activity of self-driven CRISPR/Cas9 nanosystem

Cas9/sgRNA (CRISPR/Cas9: sgRNA = 1:1, Molar ratio) and MHS (MH: Cas9/sgRNA = 4:1, mass ratio) was incubated respectively in PBS containing serum (10% v/v) for different durations (0 h, 6 h, 12 h, 24 h) to assess the stability of the CRISPR/Cas9 nanosystem, followed by agarose gel and SDS-PAGE electrophoresis, to examine the stability of sgRNA and Cas9 protein in MHS.

To investigate the effect of the loading process, Cas9/sgRNA were set into 5 groups dispersed in acidic PBS (pH = 5.0) and incubated for 24 h at 37 °C (Cas9/sgRNA, MHS, MHS + US, LGG-MHS, LGG-MHS + US). The same concentration of Cas9/sgRNA was taken for each group and incubated with *IDO1* DNA fragments, and finally agarose gel electrophoresis was performed.

MRS agar plates coating was employed to detect the effect of MHS loading on the biological activity of LGG. $1 \times 10^7$ CFU LGG in PBS without stirring was set as the control group, $1 \times 10^7$ LGG in PBS with different concentrations of MHS (0 mg/mL, 0.5 mg/mL, 1 mg/mL, 2 mg/mL, 4 mg/mL) and given mechanical stirring was set as the experimental group. After various times (0, 6, 12, 24 h) the groups were coated on MRS agar plates (100 μL taken after 100-fold dilution).

## Cell culture

The mouse-derived breast cancer cells 4T1 cell line was originally obtained from American Type Culture Collection (ATCC). Luc⁺ 4T1 cells were obtained from OBiO Technology (Shanghai) Corp.,Ltd. 4T1 and Luc⁺ 4T1 cells were cultured in Roswell Park Memorial Institute 1640 medium (RPMI 1640, Thermo Fisher Scientific) supplemented with 10% fetal bovine serum (United Bioresearch, Inc) and 1% penicillin-streptomycin at 37 °C in a humidity atmosphere containing 5% CO₂.

## Cellular uptake and intracellular ROS detection

4T1 cells were seeded into CLSM-specific culture dishes at a density of $1 \times 10^5$ and incubated for 24 h at 37 °C, followed by pre-treatments of MβCD, sucrose, and amiloride for 30 min, following the medium was replaced by Cy3-labeled MHS (MHS = 100 μg/mL), which was then co-incubated for 3 h. Then, the medium was washed with PBS for 3 times, followed by the cell nucleus was stained with DAPI for 20 min. To further observe the intracellular fluorescence intensity, and the fluorescence signals were measured.

The 4T1 cells were randomly divided into 8 groups: control, US only, MH (100 μg/mL), MH + US (100 μg/mL), ZIF-8 (100 μg/mL), ZIF-8+US (100 μg/mL), MHS (100 μg/mL) and MHS + US (100 μg/mL). Briefly, 4T1 cells were incubated with various samples at 37 °C for 6 h. Then, the medium was replaced with 2′, 7′-dichlorodihydrofluorescein diacetate (DCFH-DA), and US irradiation (1.0 MHz, 1.0 W/cm², 50% duty cycle, 5 min) was performed on US only and MHS + US groups. After continuing incubation for 30 min, the intracellular ROS generation in various treatment groups was assessed by CLSM.

## Evaluation of ¹O₂ generation in vitro

For evaluating the ability of MHS to generate ¹O₂ under US irradiation, 1, 3-diphenylisobenzofuran (DPBF), a ¹O₂ indicator, was also applied to assess the ¹O₂ generation of MHS in vitro. Briefly, 5 μL of DPBF in DMSO solution (5 mL) was added to the MHS solution (100 μg/mL). After irradiating the mixture with US (1.0 MHz, 1.0 W/cm², 50% duty cycle) for various durations (0, 1, 2, 3, 4, 5 min), the characteristic UV-Vis absorption spectra of DPBF were measured to assess the ¹O₂ generation.

## Analysis of cellular uptake and lysosomal escape of MHS nanoparticles

Lysosomal escape analysis of Cy5.5-labeled MHS by CLSM (ZEISS LSM900) was used to record the intracellular distribution of Cas9/sgRNA. 4T1 cells were pre-seeded into CLSM-specific culture dishes at density of $1 \times 10^5$ and incubated for 24 h to adhere to culture plates. Then, 4T1 cells were incubated with MHS nanosystem (100 μg/mL) with US irradiation (1.0 MHz, 1.0 W/cm², 50% duty cycle, 5 min) or without US irradiation for 1 or 3 h. DAPI (10 μg/mL) and LysoTracker (0.5 μg/mL) stained the nuclei blue and lysosome green, respectively.

## Evaluation of cell viability and cellular apoptosis

To detect the cytotoxicity MHS composite nanoparticles in vitro, 4T1 cells were inoculated in 96-well plates ($1 \times 10^4$) and allowed to attach for 24 h. Subsequently, various concentrations (0, 3.125, 6.25, 12.5, 25, 50, 75, and 100 μg/mL) of MHS nanosystem were co-incubated for 12 h. Cell activity was measured by standard CCK-8 assay.

4T1 cells were seeded in 96-well plates ($1 \times 10^4$) and attached to plates for 24 h to demonstrate the synergistic therapeutic effect in vitro. These cells were given various treatments, including control, US only, MH, MHS, MH + US and MHS + US (100 μg/mL, 1.0 MHz, 1.0 W/cm², 50% duty cycle, 5 min). The standard CCK-8 assay was used to determine cell viability.

Quantitative assessment of apoptosis was performed by flow cytometry. 4T1 cells were cultured in 6-well plates ($1 \times 10^5$) for 24 h and allowed for attachment to culture dishes with different treatments as described previously. Cells were collected after trypsin dissociation, centrifugation and washing with PBS three times. Finally, the cells were incubated for 20 min with a mixture of 5 μL of Propidium Iodide (PI) and 10 μL of fluorescein isothiocyanate (FITC). Flow cytometry was used to determine the level of apoptosis.

Meanwhile, 4T1 cells were seeded into CLSM-specific culture dishes at density of $1 \times 10^5$ and incubated for 24 h to attach culture plates. CLSM was used to examine 4T1 cells live and dead conditions after various treatments, where live and dead cells were stained with calcein-AM and PI, respectively.

## Detection of gene editing efficiency of in vitro

4T1 cells were cultured in 6-well plates for 24 h ($1 \times 10^5$ cells per well). The culture medium was replaced with new medium without serum. After 1 h, MH/MHS (100 g/mL) was added to the wells of the MH, MHS, MH + US, and MHS + US groups. Following 12 h incubation period, the US only group, MH + US group, and MHS + US group received US irradiation (1.0 MHz, 1.0 W/cm², 50% duty cycle, 5 min). The medium was replaced with 2 mL of fresh medium containing 10% FBS. After that, the *IDO1* mutation genome was analyzed after different treatments. Treated 4T1 cells were extracted genomic DNA for PCR analysis. Deep sequencing was performed after purification and collection of PCR products. (Primer sequences: *IDO1*-F- TTCTGTGTGGTTTCTCCACATC, *IDO1*-R- GTCTCCAAACCTTTCGAACATC).

## In vitro *IDO1* expression assay

4T1 cancer cells were seeded into CLSM-specific culture dishes at a density of $1 \times 10^5$ cells/dish and cultured overnight for cell attachment. The cells were stimulated with IFN-γ (1 ng/mL) for 12 h and these cells were subjected to different treatments including control, US only, MH, MHS, MH + US and MHS + US (100 µg/mL, 1.0 MHz, 1.0 W/cm², 50% duty cycle, 5 min). After washed three times with PBS, the cells were incubated with IDO Rabbit mAb (Cell Signaling Technology, Cat# 86630) for 1 h as well as staining with Alexa Fluor 555-goat anti-rabbit IgG (Abcam, Cat# ab150078) for 1 h (The antibody concentration was 1:1000 diluted with PBS). The cells were stained with DAPI for 20 min and the fluorescence images of cells were captured using a CLSM.

For the Western Blot detection of IDO, the above-treated cells in each well were washed twice with PBS, followed lysed on ice for 30 min and mixed gently upside down. After centrifugation ($13,680 \times g$), the total protein concentration of the supernatant cell lysate was measured with the BCA Protein Assay Kit. An equal amount of total protein (30 µg) was analyzed by 10% SDS-PAGE and transferred to PVDF membranes. The samples were incubated with blocking buffer for 1 h at room temperature. The membranes were immersed in a glyceraldehyde-3-phosphate solution (1:1000) containing β-actin Anti-Rabbit Antibody (Abcam, Cat# ab8227) and IDO Rabbit mAb, incubated for 12 h, and soaked with the conjugated secondary antibody (Beyotime) for 1 h at room temperature. Finally, PVDF membranes were washed three times (10 min each) with 0.02% Tris-buffered saline and Tween 20 buffer and Pierce™ ECL Western Blotting Substrate were used for visualization of the target protein.

## Surveyor assay for the detection of genomic modifications

After 3 days of different treatments, the genomic DNA of 4T1 cells was harvested using the FastPure Cell/Tissue DNA Isolation Mini Kit. DNA was amplified using $2 \times$ Fast Taq Master Mix to amplify the gene loci using forward primer TTCTGTGTGGTTTCTCCACATC and reverse primer GTCTCCAAACCTTTCGAACATC. To reduce nonspecific amplification, a touchdown PCR procedure ((95 °C for 15 s, 60 °C for 15 s, 72 °C for 15 s/kb) for 35 cycles, 72 °C for 5 min) was used. After purification by gel extraction, indole formation efficiency was assayed according to the T7 Endonuclease I kit. The digested DNA was analyzed by 2% agarose gel electrophoresis. Indole formation efficiency was calculated by ImageJ.

## Next-generation sequencing analysis of gene editing

The portion of gene editing was determined in 4T1 cells, as described below. Genomic DNA was extracted with QuickExtract DNA Extraction Solution and amplified using site-specific primers for Illumina sequencing.

## In vitro ICD expression assay

After 1 days of various treatment (control, US only, MH, MHS, MH + US and MHS + US), 4T1 cells were washed twice with PBS in 6-well plates ($5 \times 10^5$) before being lysed on ice for 30 min and gently mixed upside down. After centrifugation at $13,680 \times g$, the total protein concentration of the supernatant cell lysate was measured with the BCA Protein Assay Kit. An equal amount of total protein (30 µg) was analyzed by 10% SDS-PAGE and transferred to PVDF membranes. The samples were incubated with blocking buffer for 1 h at room temperature. The membranes were immersed in a glyceraldehyde-3-phosphate solution (1:1000) containing β-actin Anti- Rabbit Antibody (Abcam, Cat# ab8227), HMGB1 (Affinity Biosciences, Cat# AF7020), CRT (Affinity Biosciences, Cat# DF10202) and HSP70 (Affinity Biosciences, Cat# AF5466) rabbit polyclonal antibody, incubated for 12 h, and soaked with the conjugated secondary antibody (Beyotime) for 1 h at room temperature. Finally, PVDF membranes were washed three times (10 min each) with 0.02% Tris-buffered saline and Tween 20 buffer, and Pierce™ ECL Western Blotting Substrate were used for visualization of the target protein.

Detection of immunogenic cell death (ICD) in vitro. To determine CRISPR/Cas9 nanosystem-induced ICD of the tumor cells, surface expression of CRT, extracellular release of HMGB1, and HSP70 were examined by immunofluorescence in vitro. Typically, 4T1 cells ($1 \times 10^5$ cells per well) were seeded in CLSM-specific culture dishes. Then, the cells were incubated with PBS, US only, MH, MHS, MH + US, and MHS + US for 6 h, and then irradiated with or without the US irradiation (1.0 MHz, 1.0 W/cm², 50% duty cycle, 5 min). Following further incubation of 12 h, the cells were washed three times with PBS, incubated with HMGB1, HSP70, and CRT-rabbit monoclonal antibodies for 1 h as well as staining with Goat Anti-Rabbit IgG H&L (Alexa Fluor® 488) (Abcam, Cat# ab150077) or Goat Anti-Rabbit IgG H&L (Alexa Fluor® 555) (Abcam, Cat# ab150078) for 1 h. The cells were stained with DAPI for 20 min and observed using CLSM.

## DC maturation in vitro

BMDCs were extracted from the bone marrow of 5-week-old female Balb/c mice to study DC maturation in vitro. 4T1 cells were pretreated for 12 h with PBS, US only, MH, MHS, MH + US and MHS + US (MH/MHS = 100 µg/mL, US = 1.0 MHz, 1.0 W/cm², 50% duty cycle, 5 min). Afterwards, $1 \times 10^6$ immature DC cells were co-cultured with $1 \times 10^5$ pretreated 4T1 cells in a transwell system for 24 h. FCM was used to examine the maturation of DC cells after staining with APC anti-mouse CD80 Antibody (Biolegend, Cat# 104714, Clone No.16-10A1), PE anti-mouse CD86 Antibody (Biolegend, Cat# 105007, Clone No.GL-1), and FITC anti-mouse CD11c Antibody (Biolegend, Cat# 117306, Clone No. N418) antibodies (The antibody concentration was 1:100 diluted with PBS). Cell culture supernatants were also collected. And ELISA kits were used to measure the pro-inflammatory cytokines IL-2 and IL-12p70 secreted by DCs.

## Laboratory animals

Female Balb/c mice (5–6 weeks) were purchased from Shanghai Sakas Biotechnology Co., Ltd. All mice were housed in SPF-grade pathogen-free facilities, 12 light/dark cycle, 40–70% relative humidity, temperature ~25°, with free access to standard food and water.

## Transcriptome sequencing of injection of LGG

The study was conducted on 6-week-old female Balb/c mice ($n = 6$). To establish 4T1 tumor-bearing mouse models, Balb/c mice were subcutaneously implanted with 4T1 cells. After the tumor volume reached ~200 mm³, and then they were randomly divided into two groups ($n = 3$ per group), including the control and LGG groups (intravenous injection of $1 \times 10^7$ CFU LGG). At 24 h after the injection, tumor tissues were extracted, followed by nucleic acid extraction and full transcriptome sequencing.

## In vivo fluorescence imaging

Cy5.5-labeled MHS (200 µL, Cy5.5-MHS = 10 mg/kg, Cy5.5 = 10 µg/mL), Cy5.5-labeled LGG (200 µL, Cy5.5-LGG = $1 \times 10^7$ CFU, Cy5.5 = 10 µg/mL) and Cy5.5-labeled LGG-MHS (200 µL, LGG = $1 \times 10^7$ CFU, Cy5.5-MHS = 10 mg/kg, Cy5.5 = 10 µg/mL) were intravenously injected into mice when the tumors volume reached about 200 mm³. At various time points (0, 2, 4, 6, 8, 12, 48, and 72 h), mice were anesthetized and imaged by VISQUE imaging system.

## Bacterial colonization in vivo

After bacteria injection, the major organs (heart, liver, spleen, lung, kidney, and tumor) of mice were collected at the desired time points (0, 2, 6, 24, and 72 h). The major organs were extracted, weighed, and homogenized. The samples were plated on MRS plates after 100-fold dilution.

## In vivo safety evaluation

Fifteen 6-week-old female Balb/c mice were divided into five groups at random. Then blood was collected on control (without any treatment) and 1, 3, 7, and 30 days after tail vein injection of LGG-MHS (200 µL, LGG = $1 \times 10^7$ CFU, MHS = 10 mg/kg) for the determination of blood biochemical parameters and liver and kidney function parameters. The major organs (heart, liver, spleen, lung, and kidney) were collected for HE staining and MRS dish coating.

15 tumor-bearing 6-week-old female Balb/c mice were randomly divided into 5 groups, and different doses of LGG-MHS (control, 10 mL/kg, 20 mL/kg, 30 mL/kg, 40 mL/kg. A total of 1 mL LGG-MHS consisted of $1 \times 10^7$ LGG and 200 µg MHS) were injected when the tumor volume reached 60–80 mm³. US (1.0 W/cm², 50% duty cycle, 5 min) was performed on tumor the next day. Blood was then collected to determine blood biochemical parameters and liver and kidney function parameters. The major organs (heart, liver, spleen, lung, and kidney) were collected for HE staining.

## Effect and mechanism of In vivo synergistic multi-therapies

4T1 tumor cells ($1 \times 10^6$) were injected into the axillary of 6-week-old female Balb/c mice (~20 g) to establish a xenograft tumor model. These mice were divided at random into 8 groups (per group, $n = 5$): control (200 µL, PBS), LGG (200 µL, LGG = $1 \times 10^7$ CFU), MHS (200 µL, MHS = 10 mg/kg), LGG-MHS (200 µL, LGG = $1 \times 10^7$ CFU, MHS = 10 mg/kg), MHS + US (200 µL, MHS = 10 mg/kg, US = 1.0 MHz, 1.0 W/cm², 50% duty cycle, 5 min), LGG-MH + US (200 µL, LGG = $1 \times 10^7$ CFU, MH = 10 mg/kg, US = 1.0 MHz, 1.0 W/cm², 50% duty cycle, 5 min), LGG-MHI + US (200 µL, LGG = $1 \times 10^7$ CFU, MHI = 10 mg/kg, US = 1.0 MHz, 1.0 W/cm², 50% duty cycle, 5 min), LGG-MHS + US (200 µL, LGG = $1 \times 10^7$ CFU, MHS = 10 mg/kg, US = 1.0 MHz, 1.0 W/cm², 50% duty cycle, 5 min). The above drugs were injected on days 7, 9, 11, and 13, respectively, and the treatment groups with US application were irradiated with US on days 8, 10, 12, and 14, respectively. Tumor volume and body weight of mice were measured every 2 days during days 7–21. Calculate the tumor volume according to the formula (tumor length) × (tumor width)2/2. For ethical standards, mice were euthanized when they were moribund or when the tumor diameter (tumor volume not exceeding 1500 mm³ is a prerequisite) exceeded 2 cm in any dimension. Survival was assessed from the first day of 4T1 cells injection to day 60. After 24 h of the first treatment, the mice were euthanized (per group, $n = 3$) and the tumors were dissected and fixed in 10% formalin. For histological analysis, the tumor tissues were sectioned and stained with HE, TUNEL, and Ki-67.

To evaluation of IDO expression, Trp and Kyn content were tested in vivo. The mice in each group were euthanized (per group, $n = 3$) after 14 days of different treatments. The tumor tissues were collected for immunofluorescence staining (IDO). After that, a homogenizer was used to homogenize the tumor and TDLNs tissues, respectively. Half of the homogenate was incubated in 30% trichloroacetic acid at 50 °C for

30 min and the supernatant was collected for HPLC analysis to determine the level of Trp. The other half of the homogenate was centrifuged and the supernatant was collected for Elisa analysis to measure the content of Kyn.

## The immune response against primary tumor

CD3⁺CD4⁺ T cells, CD3⁺CD8⁺ T cells and CD3⁺CD4⁺FOXP3⁺ T cells, CD11c ⁺CD86⁺CD80⁺ dendritic cells as well as CD45⁺CD11b⁺F4/80⁺CD206⁺ M2 like macrophage and CD45⁺CD11b⁺F4/80⁺CD86⁺ M1 like macrophage in the tumor tissues or spleen were isolated and analyzed using flow cytometry. The antibodies involved in the experiment include FITC anti-mouse CD11c Antibody (Biolegend, Cat# 117306, Clone No. N418), PE anti-mouse CD86 Antibody (Biolegend, Cat# 105007, Clone No.GL-1), APC anti-mouse CD80 Antibody (Biolegend, Cat# 104714, Clone No.16-10A1), FITC anti-mouse CD3 (Biolegend, Cat# 100203, Clone No.17A2), PE anti-mouse CD4 (Biolegend, Cat# 100407, Clone No.GK1.5), APCanti-mouseCD8a (Biolegend, Cat# 100712, Clone No.53-6.7), AlexaFluor488 anti-mouse FOXP3 (Biolegend, Cat# 320012, Clone No.150D), FITC anti-mouse/human CD11b Antibody (Biolegend, Cat# 101205, Clone No. M1/70), APC anti-mouse CD45 (Biolegend, Cat# 103112, Clone No.30-F11), PerCP anti-mouse F4/80 Antibody (Biolegend, Cat# 123126, Clone No.BM8) and PE/Cyanine7 anti-mouse CD206 (Bioegend, Cat# 141720, Clone No.C068C2). The antibody concentration was 1:100 diluted with PBS.

IL-2, IL-12p70, TNF-α, and IFN-γ in primary tumors were also examined with ELISA kits. And tumors were stained for immunofluorescence of CD3⁺CD4⁺ and CD3⁺CD8⁺ proliferated CTLs in 4T1 tumor tissue slices of the primary tumor.

## The immune response against distant tumor

4T1 tumor cells ($1 \times 10^6$) were injected into the second left breast pad of the mice for 7 days as the primary tumor, and the second right breast pad of each mouse was injected as a distant tumor ($1 \times 10^6$ of 4T1 cells). The mice were randomly divided into 8 groups and the primary tumor was treated as described above, while the distant tumors did not accept any treatment (per group, $n = 5$). The bilateral tumor volumes of the mice were recorded every 2 days from 7 to 21 days. At 21 days after tumor cells injection, the bilaterally tumors of the rest mice were removed and photographed to document tumor size. In addition, to explore the immune activation effect of this system in distal tumors, different groups of mice were immunofluorescence stained (per group, $n = 3$) for CD3⁺CD4⁺ and CD3⁺CD8⁺ proliferating CTLs in distal tumor tissues after the first treatment for 24 h.

## Long-term immune effect to anti-lung metastasis

To establish anti-lung metastasis tumor models, $1 \times 10^6$ 4T1 cells was injected into right axillary of the mice. The mice were randomly divided into 8 groups (per group, $n = 3$) after 7 days, including control (200 µL, PBS), LGG (200 µL, LGG = $1 \times 10^7$ CFU), MHS (200 µL, MHS = 10 mg/kg), LGG-MHS (200 µL, LGG = $1 \times 10^7$ CFU, MHS = 10 mg/kg), MHS + US (200 µL, MHS = 10 mg/kg, US = 1.0 MHz, 1.0 W/cm², 50% duty cycle, 5 min), LGG-MH + US (200 µL, LGG = $1 \times 10^7$ CFU, MH = 10 mg/kg, US = 1.0 MHz, 1.0 W/cm², 50% duty cycle, 5 min), LGG-MHI + US (200 µL, LGG = $1 \times 10^7$ CFU, MHI = 10 mg/kg, US = 1.0 MHz, 1.0 W/cm², 50% duty cycle, 5 min), LGG-MHS + US (200 µL, LGG = $1 \times 10^7$ CFU, MHS = 10 mg/kg, US = 1.0 MHz, 1.0 W/cm², 50% duty cycle, 5 min). The tumors were surgically removed from the mice after 21 days. the mice were intravenously infused with $1 \times 10^5$ Luc-4T1 cells. The lung metastasis tumor was detected by in vivo luciferase-based noninvasive bioluminescence imaging system and record the number of nodules.

To study memory T cells, spleen tissues from different groups of mice were harvested and stained with APC anti-mouse CD8a (Biolegend, Cat# 100712, Clone No.53-6.7), FITC anti-mouse CD3 (Biolegend, Cat# 100203, Clone No.17A2), PE anti-mouse/human CD44 Antibody

(Bioegend, Cat# 103007, Clone No.IM7) and APC/Cyanine7 anti-mouse CD62L Antibody (Bioegend, Cat# 104428, Clone No.MEL-14) antibodies. Flow cytometry was used to isolate and analyze effector memory T cells ($CD3^+CD8^+CD44^+CD62L^-$, Tem and $CD3^+CD8^+CD44^+CD62L^+$, Tcm). The antibody concentration was 1:100 diluted with PBS. Finally, lung tissue tumor metastasis was visualized using HE staining.

## Statistics and reproducibility
GraphPad Prism (version 9.0.0, GraphPad Software, San Diego, California, USA) was employed to calculate all statistical analyses. All flow cytometry data were analyzed on FlowJo™ software package (version 10.5.2, BD Life Sciences, Ashland, Oregon, USA). Living imaging software CleVue (version 3.1.3.2054, Vieworks Co., Ltd, Anyang-si, Gyeonggi-do, Republic of KOREA) was used to analyse bioluminescent and fluorescent images. The $p$-value less than 0.05 was considered significant ($*P < 0.05$, $**P < 0.01$, $***P < 0.001$, $****P < 0.0001$).

## Reporting summary
Further information on research design is available in the Nature Portfolio Reporting Summary linked to this article.

## Data availability
The RNA-sequencing data generated in this study have been deposited in the Genome Sequence Archive (GSA) database under accession code CRA007764. The remaining data are available within the Article, Supplementary Information or Source Data file. Source data are provided with this paper.

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

## Acknowledgements

We greatly acknowledge the financial support from the National Natural Science Foundation of China grant to H.X.X. (82151318), L.P.S. (82171945) B.G.Z. (82202154), Y. F (82102050) and C.K.Z (82202174), Shanghai Science and Technology Committee grant to H.X.X. (19DZ2251100), Scientific Research and Development Fund of Zhongshan Hospital of Fudan University grant to H.X.X. (2022ZSQD07), Shanghai Municipal Health Commission grant to H.X.X. (2019LJ21 and SHSLCZDZK03502) and Shanghai Sailing Program to B.G.Z. (22YF1433700).

## Author contributions

H.H.Y and H.X.X. designed the research strategy and experiments; H.H.Y., J.F.Y., B.G.Z., S.Z., B.Y.Z., Y.K.S., X.L.L., D.D., Y.F., L.F.W., C.K.Z., and Y.Y.P. performed experiments and/or analyzed the data; H.H.Y., J.F.Y., S.Z., H.X.X., Y.Z., and L.P.S. wrote the paper; H.H.Y. and H.X.X. supervised the whole process.

## Competing interests

The authors declare no competing interests.
