## [Peer Review File · Nature Communications]

Design of a self-driven probiotic-CRISPR/Cas9 nanosystem for sono-immunometabolic cancer therapyREVIEWER COMMENTS

Reviewer #1 (Remarks to the Author): with expertise in bacteria engineering / cancer therapy

The manuscript presents a multimodal approach to reprogram tumor immunosuppressive microenvironment to improve immunotherapy efficacy. The authors had combined several technologies including tumor-colonizing bacteria, CRISPR system, and ultrasound irradiations to alter tumor microenvironment. While there has been increasing report using these systems individually to alter tumor immune microenvironment, combination of all technologies is novel. This system was tested in vitro under various conditions, and was shown to have significant efficacy in breast cancer animal models. While the study shows interesting combinations of multiple technologies to enhance cancer therapy efficacy, contributions from individual components as well as the rationale of the combinations were not clear. Furthermore, there remain key questions to be answered such as the inclusion of appropriate experimental controls, biological replicates, and statistical analysis to support claims in the manuscript.

The authors claim multiple mechanisms of their therapy contributing to the efficacy. However, individual contributions from each component and rationale for their combination are not clear. For example, there seems to be two mechanisms in which US works with this approach. On one hand the authors claims that the generation of ROS kills the tumor cells which shows apoptosis of most cells when treated in vitro. On the other hand, the authors also argue that US triggers release of CRISPR complex from the lysosomal compartments. It is not clear how CRISPR system can contribute to this therapy if the tumor cells are killed anyways. Deaths of the cells can simply result in the reduction of IDO levels. The mechanisms in which conjugated MHS gets released from LGG and get into cancer cells are also unclear. Another contradiction is the delivery of the MHS. It was not clear how MHS was delivered. Since authors showed MHS do not accumulate in tumors after systemic injections, how come MHS+US show strong efficacy?

Many important information such as controls is missing which makes it difficult to properly evaluate the data. First, several important data is missing biological replicates. For example, data supporting the CRISPR-mediated gene editing only shows single replicate or just the genetic sequence. The biodistribution of LGG in vivo were shown with single agar plate per group, but there are no quantifications or biological replicates. In Fig. S14, it seems like the bacteria level is decreasing from 24 to 72hrs in both liver and tumors, which is different from what the authors had claimed in the manuscript. Since the use of LGG for tumor targeting is new, this approach warrants more careful characterizations. Second, there are lack of description regarding the statistical analyses performed and some interpretations of the data are questionable. In Fig. 6, the figure caption describes the statistical test as t-test, but these data include multiple groups and timepoints which cannot be analyzed with the t-test. Third, some key controls are missing from several experiments. For example, the authors included some combination of their systems to compare their efficacy, but LGG+US is missing from the experimental groups. Is it possible that LGG+US is just as effective as LGG-MHS+UG? Lastly, some details on experimental settings are missing which makes it difficult to properly assess what was done. In the animal experiments, it is not clear how the treatment was performed. How were LGG-MHS administered and how many times? If the LGG-MHS was administered systemically, did they also get to the distal tumors in Fig. 8 experiments? The possibility of LGG-MHS colonization and its effect on distal tumors needs to be excluded before the authors claim the contribution of systemic immunity.

Overall, while the amount of data presented in the manuscript is impressive, the above points need to be addressed to properly assess the claims of the study.

Other points:

- There seems to be several typo, missing figure reference, and mislabeling that should be corrected
- Reference #30 do not include studies supporting LGG colonization in tumors
- Reference #31 did not show LGG tumor colonization and local remodeling of the microenvironment
- Line 132-134 is misleading. Fig. S3 shows some residual RNA by 3 hours. Where are the

replicates?

- Line 135-137: Is pH5 relevant to tumor microenvironment? How does this release relate in the in vivo conditions, since the CRISPR complexes needs to be delivered to intracellular regions? If it is released prematurely in the tumor microenvironment, doesn't this reduce the efficacy?
- Fig. 3d label should be edited. It isn't showing % viability
- Line 224-226: where is the data supporting this claim?
- Fig. 4f and g: why did the authors just decided to look into IL-12p70 and IL-2 among all other cytokines?
- Fig. 5a and c: Are LGG and MHS both labeled with Cy5? If so, shouldn't one expect much higher signal from LGG-MHS compared to LGG alone? It looks like they are at similar levels, possibly suggesting that MHS is not getting to tumors
- Since the authors had claimed the ability for MHS to remodel immunometabolism, RNA sequencing result on LGG-MHS may be helpful to decipher the contribution on TME remodeling from LGG alone vs LGG-MHS.
- Fig. S17: where are the data showing bacteria levels in tumors? This should also be quantified with biological replicates.
- Fig. S19 and S20: I don't see the control groups as claimed in the main text. The values seem to change over time – what statistics did the authors use to get the non-significance?
- Fig. 6: The authors indicated day 7, 9, 11, and 13 as treatment days. Is this LGG-MHS injections? Or is it US treatment? How are LGG-MHS administered?
- Fig. 8a: The inoculation of secondary tumors at mammary pad isn't strictly metastatic model. I suggest editing the main text

Reviewer #2 (Remarks to the Author): with expertise in cancer immunology, IDO

The manuscript by Yu et al describes use of a microbial vector (lactobacillus, LGG) and nanoparticle delivery system (MHS) activated by ultrasound (US) irradiation to target and manipulate the tumour microenvironment (TME). The authors used this approach to boost anti-tumour immunity in two ways by; (1) stimulating reactive oxygen species (ROS) production following US and (2) using LGG to deliver CRISPR/Cas9 gene editing functions to excise indoleamine 2,3 dioxygenase-1 (IDO1) genes, which mediate immune suppression via IDO enzyme activity. TME-targeting efficacy was evaluated in a murine breast cancer cell line (4T1) and the 4T1/BALB/c tumour model. Data reported largely support the authors' claims that the LGG/MHS delivery system is an effective method to incite protective anti-tumour immunity. The manuscript is generally well-written but would benefit from increased clarity and focus on key biological findings of potential clinical significance, and reduced emphasis on technical information such as nanoparticle synthesis and validation data (see below).

Major Points:

1. The Abstract does not clearly convey major findings from the study and would benefit from extensive rewriting to enhance clarity, emphasise significant findings and minimise technical information. In particular, more emphasis should be placed on describing outcomes from experiments conducted using the mouse tumour model, as these data are far more informative than studies performed on cell lines regarding future prospects for clinical translation of the results from this study.
2. The initial description of the nanoparticle delivery system in the Introduction is confusing (lines 86-103). In particular, the meaning of the acronym MHS needs clarifying, as does the purpose of using ZIF8 and HHME in the strategy used in this study. The graphic depicting study goals (Fig. 1) helps but is far too complicated. This graphic should be simplified to focus exclusively on key elements of the strategy employed in the study; in other words, make it into a graphic hypothesis.
3. The authors do not justify their choice of the 4T1 tumour model. Most importantly, is the 4T1 model dependent on IDO activity for optimal 4T1 tumour growth? If not, this undermines the strategy used and prompts the use of a tumour model known to be dependent on IDO for optimal growth (eg. the LLC tumour model). Linked to this key point, what is the authors' rationale for administering treatments when 4T1 tumours were 200mm³?
4. The authors must assess IDO enzyme activity by measuring kynurenine levels in the TME and

draining lymph nodes to evaluate if their treatment strategy reduces nominal levels of IDO enzyme activity that may promote immune suppression required for optimal tumour growth. Note that assessing (1) IDO1 protein expression or (2) Trp levels are not sufficient to measure IDO activity in the TME. Linked to this point, the authors should test if IDO inhibitors synergise with their nanoparticle approach to boost immune activation to assess if IDO inhibitors or LGG-CRISPR/cas9 gene editing is more effective in reducing IDO activity.

5. Data reported in Fig6 & Fig8 support the authors' conclusion that LGG-MHS+US treatments reduced primary and distal 4T1 tumour burdens at experimental endpoints (day 21). MHS+US treatments also reduced tumour burdens, though to a lesser extent. These outcomes suggest that combining LGG with MHS/US nanotherapy may fully protect against 4T1 tumour growth but more studies will be necessary to support this claim rigorously, in particular with regard to if IDO1 gene editing is critical to promote protective outcomes (see point 4). Accordingly, the authors should assess mouse survival over longer periods and test if LGG infection or IDO1 gene editing (or both) contribute to increased protection from 4T1 tumour growth, as well as evaluating IDO enzyme activity (see point 4).

6. The tumour re-challenge strategy depicted in Fig8h indicates that primary 4T1 tumours were surgically resected on day 21. It is not clear why tumours were resected. Tumour re-challenge should be conducted by injecting 4T1 tumour cells into mice that survive primary 4T1 tumour growth after therapy without resecting primary tumours prior to re-challenge to evaluate if therapy stimulates durable and stable anti-tumour immunity that clears both primary and secondary tumours.

7. The short Discussion (lines 513 – 525) does not adequately describe the relevance and significance of the study findings, or place them in the context of the current scientific literature. This section needs extensive rewriting to address these deficiencies.

Minor Point:

1. The large number of supplemental figures (33) make the manuscript difficult to read. The authors should consult with the editors to find ways to streamline this large set of supplemental figures.

Reviewer #3 (Remarks to the Author): with expertise in nanotechnology

The paper entitled “Self-driven Probiotic-CRISPR/Cas9 Nanosystem Reprogramming of Tumor Immunosuppressive Microenvironment to Enable Sono-immunometabolic Cancer Therapy” is reporting the use of a multifunctional immunotherapeutic system for solid tumor treatment. They loaded the sonosensitizer hematoporphyrin monomethyl ether (HMME) and CRISPR/CAS9 on ZIF-8 (MHS) and combined them with *Lactobacillus rhamnosus* GG (LGG) for enhancing immunotherapy efficacy. LGG bacteria was used as a carrier for in vivo study to increase the targetability of the system toward tumors. The system consisted of ZIF-8 which was used as a vector to protect Cas9/sgRNA, HMME was used to generate ROS under ultrasound irradiation (US) to induce lysosomal rupture and release Cas9/sgRNA which is intended to knock down the IDO1 gene and promote immunogenic cell death (ICD). They tested the efficacy of the system in both, in vitro and in vivo. It is evident that they tried to evaluate the efficiency of their system using different experimental approaches. While the in vivo results looked promising, they did not provide a clear conclusion about the advantage of each individual component of the system and its role in the success of the treatment. They lack many control experiments which made the data presented inexplicit. Therefore, acceptance can be recommended at this stage. The following comments need to be addressed to have a better understanding of their system.

1. For the construct assembly, it was not clear how HMME was loaded into ZIF-8. What type of interaction is happening? The same for Cas9/sgRNA, did it infiltrate ZIF-8 or did they form a complex?
2. The illustration and the terms “loading” and “encapsulation” are not very accurate. The author claimed the loading/ encapsulation of Cas9/sgRNA into ZIF-8, however, the reported pore size of ZIF-8 is very small for Cas9/sgRNA to internalize.
3. In figure 2C, how did ZIF-8 maintain its hexagonal structure after combining it with HMME and CRISPR/CAS9? and the size increase after complexation has to be justified.

4. The elemental mapping (EM) in figure 2i does not correspond to the TEM image of LGG-MHS in 2h. It is better to compare it to the elemental mapping of LGG alone and compare the EM of MHS to ZIF-8 alone using the same experimental settings.
5. In line 120, they mentioned "utilizing sodium dodecyl sulfate-polyacrylamide gel electrophoresis (SDS-PAGE)", however, figure S1 shows an agarose gel of the sgRNA only. Therefore, they need to show the loading of the different mass ratios of MH to Cas9 used in order to obtain the optimal loading concentration.
6. Figure S3, the MHS stability experiment has to be conducted after 12, 24hrs, since the system is incubated with the cells for 24 hrs. Also, running the same experiment on SDS PAGE with free Cas9/sgRNA would show the stability of Cas9 as well.
7. Figure 3a, the group measured the generated ROS after exposing MHS to US, but they did not report the effect of US radiation on ZIF-8 alone and MH, and hence, the reason for adding HMME would be justified.
8. In Figure S7, the author claims that Cy5.5-labeled Cas9/sgRNA system entered the nucleus, however, the Cy5 signal seems to follow the pattern of the lysotracker. In addition, the nucleus does not look intact. Z-stack is needed to show localization in the nucleus.
9. In the cytotoxicity experiment (Figure 3d), if the role of gene silencing is to improve the immune system mediated killing of the cells, why do we see improved efficacy when no immune cells are present in the model? Why is the toxicity MHS+US significantly higher than the MH+US system. Similar observation was seen with Fig.3e & S8 between MH+US and MHS+US group. Why the presence of Cas9/sgRNA increased the apoptosis in 4T1 cells?
10. In Figure 3h, the 12% difference in cleavage between the two groups is not reflected in agarose gel. Also, NGS and the Deep sequencing data for MHS only were not provided.
11. In figure 3f, in the MHS+US group, the reduced signal might be due to the cells being out of focus compared to the others. We suggest using the nucleus as a point of focus to make it easier to visualize and compare.
12. In fig. S9, the expression of IDO1 seems to be lower in the case of MHS compared to MHS+US which contradict the gene deletion rates mentioned in line 205 and 206.
13. In the experiment "In vitro exploration of ultrasonic-immunometabolic therapy" line 236-237, the correlation or the mechanism by which MHS + US triggered the ICD is not clear since some groups showed similar trends in the case of protein expression Ex. MHS group had similar protein expression for CRT and HSP70 to MHS +US group (Figure 4a).
14. In figure 5, was RNAseq-based KEGG analysis of differential gene expression profiles conducted for LGG-MHS+US treatment only? Again there are many controls missing
15. The biosafety of the LGG-MHS nanosystem on different organs was evaluated without applying the US which is the main activator of the system. It would be more reflective to show that after applying US.
16. For all in vivo experiments with LGG+MHS+US, a main control is missing. The role of gene knockdown of Cas9/gRNA will not be conveyed clearly if LGG-MH+US is not tested.
17. There are many grammatical mistakes that need to be corrected. Ex. Line 75 "is" not needed, line 77 "barrier", line 78 "it maintains", line 166 it improves" gene delivery, line 333 repetition of "that", figure 5e. "kidney".

Reviewer #4 (Remarks to the Author): with expertise in bacteria cancer therapy; nanotechnology

In this manuscript, the authors reported the synthesis of ZIF-8 for tumor targeted delivery of sonosensitizer HMME and CRISPR/Cas9 system by employing the intrinsic tumor hypoxia targeting ability of LGG. By downregulating the expression of IDO1, the obtained composites were shown to be able to effectively suppress tumor growth via the combined sonodynamic treatment and tumor immunosuppression reversion. However, similar topics have been widely reported in the past several years and this study did not provide enough attractive new results.

Specific comments:

1. Attributing to the intrinsic targeting ability of LGG, it is believed that HMME and CRISPR/Cas9 system loaded within the ZIF-8 nanoparticles would be primarily delivered to the hypoxic tumor

region. Therefore, I want to know if the hypoxic condition would diminish the sonosensitization efficacy of HMME under US exposure.

2. Actually, diverse small molecule IDO1 inhibitors have been developed to reverse tumor immunosuppression by restricting the production of Kyn. Therefore, I would like to suggest the authors to describe the advantages of the presented strategies.

3. Based on the results shown in Figure 2, the pore size of the obtained MH and MHS nanoparticles with typical ZIF-8 morphology is very small. Therefore, I want to know how CRISPR/Cas9 systems were loaded. Besides, would the loading process negatively impair the biological activity of loaded CRISPR/Cas9 system? Did the US irradiation promoted generation of ROS negatively the biological activity of CRISPR/Cas9 systems.

4. The authors are suggested to describe the methods used for the loading of MHS nanoparticle onto the surfaced of LGG. Besides, Did the MHS nanoparticles loading impact the colonization behaviors of LGG.

5. In Figure 3e, it was shown that the flow cytometric plot of MHS and US treated cells was distinct from the typical apoptotic cancer cells. Please double check. Maybe the combination treatment could not induce apoptosis since it has been well documented that apoptosis of cancer cells is not the immunogenic cell death because it could not promote the expression of CRT, release of HMGB1.

6. The authors are suggested to explain why the treatment of MHS plus US was more efficient than the treatment of MH plus US in promoting the immunogenic cell death of 4T1 cancer cells. Besides, the authors are suggested to explain the mechanism of such combination treatment in promoting the expression of HSP70.

7. In figure 4h, the flow cytometric patter of these maturated BMDCs is quite different from those published ones. Please double check.

8. In Figure 7c and S25, the gating strategy used for analyzing the percentages of CD4+Foxp3+ Tregs was not correct. Please reanalyze the results. Besides, it seems that the gate strategies shown in Figure S25 were not the standard ones.

9. The font size of Figure 6b was too small. Please reformat the figure.

Response to reviewer #1

*1. The manuscript presents a multimodal approach to reprogram tumor*
*immunosuppressive microenvironment to improve immunotherapy efficacy. The*
*authors had combined several technologies including tumor-colonizing bacteria,*
*CRISPR system, and ultrasound irradiations to alter tumor microenvironment. While*
*there has been increasing report using these systems individually to alter tumor*
*immune microenvironment, combination of all technologies is novel. This system was*
*tested in vitro under various conditions, and was shown to have significant efficacy in*
*breast cancer animal models. While the study shows interesting combinations of*
*multiple technologies to enhance cancer therapy efficacy, contributions from*
*individual components as well as the rationale of the combinations were not clear.*

**Response:** We appreciate very much for your constructive comments and kind
recommendations. The manuscript and supplementary data have been revised
accordingly. The LGG-MHS nanosystem is mainly composed of two parts, namely
LGG and MHS. And then the MHS is composed of three components, M (metal
organic framework, ZIF-8), H (sonosensitizer, HMME) and S (Cas9/sgRNA). The
contributions and rationality of individual components are herein clarified as follows:

(1) LGG: *Lactobacillus rhamnosus* GG (LGG) is a parthenogenic anaerobic probiotic
that, in our strategy, acts as a carrier for targeted delivery of the whole nanosystem to
the tumor site, and can also serve as a synergistic therapeutic adjuvant for immune
activation.

First, LGG serves as a delivery vehicle in the LGG-MHS nanosystem. The tumor
microenvironment is closely associated with heterogeneous tumor growth, metastasis,
and treatment resistance¹⁻³. Currently, the strategies to target the hypoxic
microenvironment of tumor are mainly categorized into exploiting hypoxia and
alleviating hypoxia. Compared to previous therapeutic strategies to alleviate hypoxia,
tumor-specific targeting is achieved by exploiting the characteristics of the tumor

hypoxic microenvironment, thus further improving the efficiency of drug delivery^{4, 5}.
Interestingly, it has been found in many studies that parthenogenic and specialized
anaerobic bacteria can selectively target tumors and partially colonize the tumor
region as the tumor establishes anaerobic conditions, provides abundant nutrients and
protects them from immune clearance⁶⁻¹⁰.

In our study, we found that LGG has an excellent ability to target the hypoxic
microenvironment of tumors. *In vivo* fluorescence images and semi-quantitative
analysis indicate that the fluorescent intensity of Cy5.5 at the tumor site increased
over time 24 h after intravenous injection of LGG-Cy5.5 and LGG-MHS-Cy5.5,
revealing superior tumor targeting properties of the LGG-MHS complex. Notably, the
CFU of LGG in the tumor was significantly higher than that in the liver at the 72 h
time point, and LGG in the tumor accumulated and was maintained for more than 72
40 hours, which further supports the superior tumor targeting and penetration ability of
41 LGG (Line 322-333, Page 10-11, Revised Manuscript).

Secondly, LGG serves as a synergistic therapeutic adjuvant. It has been found
that bacteria can be used as an immunotherapeutic adjuvant due to its unique immune
activating effects¹¹⁻¹³. Bacterial infection in tumors can lead to antitumor responses by
inducing the migration of innate immune cells such as DCs, neutrophils, macrophages
and neutrophils into colonized tumors and by enhancing the abundant expression of
tumor necrosis inflammatory cytokines, thus killing tumor cells and preventing
metastasis formation¹⁴⁻¹⁶. LGG has also been suggested to modulate the inflammatory
state during cancer development and transformation^{17, 18}. We then hypothesized that
*Lactobacillus rhamnosus*, a parthenogenic anaerobic (*Lactobacillus* spp.), also
possesses the ability to activate immunity to fight against tumor. Subsequently, we
sequenced the tumor-bearing mice injected with LGG alone, the results showed that
LGG stimulated multiple pro-inflammatory and anti-tumor signaling pathways in
mice. Analysis of these differential genes using gene ontology (GO) and Kyoto
Encyclopedia of Genes and Genomes (KEGG) reveals that they are associated with
multiple signaling pathways, including immune infiltration of the tumor

microenvironment and promotion of tumor cell apoptosis. In summary, LGG may
possess the ability to enhance the effect of immunotherapy for tumor. (Line290-293,
**Page 9, Revised Manuscript**)

(2) MHS: The MHS is composed of three components, M (metal organic framework,
ZIF-8), H (sonosensitizer, HMME) and S (Cas9/sgRNA). ZIF-8 delivers Cas9/sgRNA
and HMME to the tumor site, and upon entry into the cell, HMME generates ROS
upon US irradiation, inducing the release of tumor-associated antigens and
immunogenic cell death of tumor cells, leading to DCs maturation. In addition, ROS
effectively disrupts the structure of the endosomal/lysosomal membrane, allowing
Cas9/sgRNA to escape from the endosome/lysosome and transport to the nucleus for
effective *IDO1* knockdown, thereby reducing Treg cells aggregation in the tumor
microenvironment.

Zeolitic imidazolinium framework (ZIF-8) is a metal-organic framework with a
large specific surface area, tailored pore size, pre-designed morphology,
biocompatibility and controlled degradability that brings such materials closer to
pharmaceutical and medical translation, allowing them to be used as an excellent non-
viral CRISPR/Cas9 delivery system¹⁹⁻²². HMME and Cas9/sgRNA are delivered into
tumor cells *via* ZIF-8, and Cas9/sgRNA rapidly escapes from endosomes/lysosomes
*via* the proton-sponge effect, thus enabling effective gene editing^{23, 24}.

In this synergistic immunotherapy strategy, HMME was used as sonosensitizer to
generate abundant ROS to damage tumor cells upon US irradiation, while the
generated ROS induce endosomal/lysosomal rupture to release Cas9/sgRNA, setting
the stage for its next step of gene editing. HMME, an organic acoustic sensitizer,
which can lead to higher ROS level and therefore produces more adequate SDT
efficiency compared to inorganic acoustic sensitizers²⁵⁻²⁷. More importantly, HMME
has been approved by the FDA for clinical use because of its high safety profile as an
sonsensitizer²⁸.

Indoleamine-2,3-dioxygenase-1 (*IDO1*) is an endogenous immunosuppressive
mediator that can stimulate the accumulation of FOXP3⁺ Tregs and suppresses T-cell
activity by depleting Trp in the microenvironment^{29, 30}. Thus, *IDO1* is a potential
immunotherapeutic target to reprogram TIME by improving amino acid metabolism³¹.
Nevertheless, small molecule inhibitors generally do not provide durable responses
due to the presence of drug resistance^{32, 33}. Therefore, there is an urgent need for
alternative approaches to interfere with amino acid metabolism to reprogram the
TIME of cancer immunotherapy.

CRISPR/Cas9, as an emerging genome editing technology, has the advantages of
simple design, high specificity and high efficiency, which bringing a breakthrough in
the regulation and application of targeted genome modification and showing broad
application prospects in biomedicine³⁴. In this strategy, after MHS entered into tumor
cells, Cas9/sgRNA escapes from the endosome/lysosome under irradiation of US and
is translocated to the nucleus for efficient *IDO1* knockdown, thereby reducing the
aggregation of Treg cells in the TIME.

*2. Furthermore, there remain key questions to be answered such as the inclusion of*
*appropriate experimental controls, biological replicates, and statistical analysis to*
*support claims in the manuscript.*

**Response:** Thank you for your kind reminder, which is essential to improve the
quality of our research. According to the reviewer's suggestion, experimental controls
such as LGG-MH + US and LGG-MHI + US groups in animal models have been
added. Mice were randomly divided into 8 groups, including Control, LGG, MHS,
LGG-MHS, MHS + US, LGG-MH + US, LGG-MHI + US and LGG-MHS + US. As
a result, the LGG-MHS + US group showed excellent ability to inhibit tumor growth
compared to the other groups. The related data have been added in the Revised
Manuscript. (Figure 6, Page 38, Revised Manuscript).

**Fig. 6 LGG-MHS + US against 4T1 tumor *in vivo*.** (a) Schematic diagram of primary tumor
 treatment process *in vivo*. (b) Tumor growth curves of 4T1 after being treated by PBS, LGG, MHS,
 LGG-MHS, MHS + US, LGG-MH + US, LGG-MHI + US and LGG-MHS + US ($n = 5$). (c)
 Average tumor growth curves in different groups ($n = 5$). (d) HPLC assay of the Trp content in
 primary tumors and TDLNs of tumor-bearing mice after different treatments ($n = 3$). (e) Elisa
 assay Kyn content in primary tumors and TDLNs of tumor-bearing mice after different treatments
 ($n = 3$). (f) Antigen Ki-67 staining in tumor sections from each experiment group ($n = 3$). (g)
 Images and (h) corresponding fluorescence intensity of IDO immunofluorescence staining in
 primary tumors of 4T1 tumor-bearing mice after various treatments. DAPI was used to stain the
 nucleus of the cell (blue), and the IDO was stained with anti-IDO antibodies (red) ($n = 3$). (i)

Average tumor growth curves after being treated by re-challenge. ($n_{\text{LGG-MHI+US}} = 2$, $n_{\text{LGG-MHS+US}}$
$= 4$)

According to the reviewer's suggestion, biological replicates such as biological
replication of LGG-associated agar plate have been added. To explore the tumor
targeting ability of LGG, the tumor tissues and major organs of 4T1 tumor-bearing
mice were homogenized and coated on MRS agar plates at different time points (0, 2,
6, 24 and 72 h) after injection of the LGG. By counting the colony forming units
(CFU) in each plate, we found that the CFU amount of LGG in the tumor was
significantly higher than the other organs, which further supports the superior tumor
targeting and penetration ability of LGG (**Supplementary Fig. 6a, b**). The related
data and discussion have been added in the Revise Manuscript and Revised
Supplementary Information (**Figure 6, Page 13, Revised Supplementary**
**Information**).

**Supplementary Figure 6.** (a) Representative photographs of MRS agar plates and (b)
corresponding quantitative analysis of bacterial colonization in various organs and tumor of 4T1-
bearing mice in a different time (0, 2, 6, 24, and 72 h) ($n = 3$).

In addition, all data in the manuscript have been double-checked, and the
inappropriate statistical methods have been corrected. Based on this fact, we have
added the following brief description in the Revised Manuscript, which reads:
"GraphPad Prism (version 9.0.0, GraphPad Software, San Diego, California USA)
was employed to calculate all statistical analyses. Tumor growth curves were analyzed
using two-way ANOVA. Dunnett's multiple comparisons post test was utilized to
analyze hematological indexes. And for other comparisons, unpaired Student's t-test
was used when comparing two groups and one-way ANOVA with Holm Sidak
correction for multiple testing was used when comparing more than two groups. The
p-value less than 0.05 was considered significant (* $p < 0.05$, ** $p < 0.01$, *** $p <$
0.001 , **** $p < 0.0001$)." (Line 723-729, Page 24, Revised Manuscript)

*3. The authors claim multiple mechanisms of their therapy contributing to the efficacy.*
*However, individual contributions from each component and rationale for their*
*combination are not clear. For example, there seems to be two mechanisms in which*
*US works with this approach. On one hand the authors claims that the generation of*
*ROS kills the tumor cells which shows apoptosis of most cells when treated in vitro.*
*On the other hand, the authors also argue that US triggers release of CRISPR*
*complex from the lysosomal compartments. It is not clear how CRISPR system can*
*contribute to this therapy if the tumor cells are killed anyways. Deaths of the cells can*
*simply result in the reduction of IDO levels.*

**Response:** Thank you very much for your kind comments and questions. Each of the
components and their corresponding contributions have been described in detail above.
Treatment of tumors *in vivo*, which not only suffers from hypoxia but also from
immunosuppression and many other elements, it is obvious that ROS alone cannot
produce satisfactory therapeutic effects³⁵⁻³⁷.

We have added the following brief description in the Revised Manuscript, which

reads: “In addition, hypoxia plays a crucial role in the tumor immunosuppressive
microenvironment and largely influences the outcome of treatment. Given the critical
role of hypoxia in tumor progression and its resistance to treatment, many efforts have
been made to overcome the limitations associated with hypoxia regarding tumors.”
**(Line 500-503, Page 16, Revised Manuscript)**

In addition, the LGG-MH +US (without Cas9/sgRNA) group was included in
animal experimental models. The results showed that this strategy did not show a
satisfactory therapeutic effect either in the primary tumors or in the against re-
challenge and lung metastasis of tumors **(Fig. 6-8, Revised Manuscript)**.

In summary, we constructed a self-driven probiotic delivery CRISPR/Cas9
system, which utilizes *Lactobacillus* as a vector, realizing efficient delivery of
CRISPR/Cas9 system to knockdown *IDO1* to reduce immunosuppressive cells
(Tregs), while *Lactobacillus* activates multiple anti-tumor signaling pathways to
activate intrinsic immunity, in addition, the system can improve gene editing
efficiency and cause immunogenic cell death (ICD) when triggered by US irradiation,
this "cocktail therapy" can effectively activate immune cells to eliminate the primary
tumor and inhibit the lung metastasis and against re-challenge of tumors.

*4. The mechanisms in which conjugated MHS gets released from LGG and get into*
*cancer cells are also unclear. Another contradiction is the delivery of the MHS. It was*
*not clear how MHS was delivered. Since authors showed MHS do not accumulate in*
*tumors after systemic injections, how come MHS+US show strong efficacy?*

**Response:** Thank you very much for your kind comments and questions. In our self-
driven nanosystem therapeutic strategy, parthenogenic anaerobic LGG acts as a
hypoxia targeting vector to target the tumor hypoxic microenvironment by
electrostatic adsorption of loaded MHS. When LGG-MHS is enriched in the tumor
hypoxic microenvironment, the acidic nature of the tumor microenvironment reduces

the force between the drug molecules and the carrier material, facilitating the release
of the drug and thus improving the delivery efficiency of the MHS³⁸⁻⁴⁰. Nanodrugs
usually enter the cell by endocytosis. In this process, the membrane region in contact
or bound to the nanoparticle invaginates or folds, forming a vesicle pocket on the
cytoplasmic side, which in turn detaches from the plasma membrane to form a vesicle.
Endocytosis is divided into phagocytosis and cytokinesis, while cytokinesis is the
main way of internalizing nanoparticles in tumor cells and most somatic cells.
Depending on the types of proteins involved, cytosolic drinking is divided into lattice-
mediated endocytosis and small concave protein-mediated endocytosis. And ZIF-8 is
mainly internalized by fossa-mediated endocytosis⁴¹⁻⁴³.

We sincerely apologize for the misunderstanding of the reviewers as we may not
have been clear enough in the original manuscript. After tail vein administration, the
MHS penetrate into the tumor area mainly through passive targeting by enhanced
permeability and retention (EPR) effect whereas the lack of active targeting leads to
inefficient enrichment in the tumor. Although the enrichment efficiency of MHS into
tumor by passive targeting is not high, a certain amount of MHS is still enriched at the
tumor. Under the US irradiation, it will produce ROS to kill tumor cells and trigger
ICD. On the other hand, the generated ROS can promote the release of Cas9/sgRNA,
implement gene editing *in vivo* to knock down *IDO1*, which can block the body
immune tolerance caused by overexpression of IDO protein as an immunosuppressive
factor in 4T1 tumor cells, thereby promoting the disintegration of the tumor
immunosuppressive microenvironment. Therefore, MHS+US can show relatively
powerful therapeutic effects. It is also noteworthy that our study demonstrated that
although MHS + US displayed relatively powerful therapeutic effects in killing tumor
cells *in vitro* and treating primary tumors, it was unsatisfactory in combating tumor
metastasis (**Fig.8f-j, Revised Manuscript**). We have added the following brief
description in the Revised Manuscript, which reads: “Notably, MHS + US and LGG-
MH + US, despite their powerful therapeutic effects in primary tumors, did not
produce satisfactory systemic immune activation against distant tumors and lung

metastases.” (Line 476-478, Page 15, Revised Manuscript)

5. Many important information such as controls is missing which makes it difficult to
properly evaluate the data. First, several important data is missing biological
replicates. For example, data supporting the CRISPR-mediated gene editing only
shows single replicate or just the genetic sequence.

**Response:** Thanks very much for your question. We totally understand the reviewer’s
concern, which is highly appreciated. According to the reviewer’s suggestion, the
CRISPR-mediated gene editing has been repeated three times. To investigate the gene
editing efficacy of the MHS nanosystem under US irradiation, Cas9/sgRNA-mediated
*IDO1* degradation was examined in 4T1 cells by T7 endonuclease I. As the results
reveal that the MHS + US group produced more cleavage products relative to the
MHS group (Fig. 3i and Supplementary Fig. 3h). This result proves that the MHS
nanosystem under US irradiation efficiently deliver the CRISPR/Cas9 system and
perform target gene loci knockdown for the gene editing purposes. The related data
and discussion have been shown in the Revised Manuscript and Revised
Supplementary Information (Line 216-220, Page 7, Revised Manuscript).

**Fig. 3 Evaluation of US-associated *IDO1* genome editing *in vitro*.** (i) T7EI cleavage analysis
after 4T1 cells with different treatments, including control, US only, MH, MH + US, MHS and
MHS + US ($n = 3$).

**Supplementary Figure 3.** (h) Corresponding quantitative analysis of T7E I cleavage after 4T1
 cells with different treatments, including control, US only, MH, MH + US, MHS, and MHS + US.

*6. The biodistribution of LGG in vivo were shown with single agar plate per group,*
 *but there are no quantifications or biological replicates. In Fig. S14, it seems like the*
 *bacteria level is decreasing from 24 to 72hrs in both liver and tumors, which is*
 *different from what the authors had claimed in the manuscript. Since the use of LGG*
 *for tumor targeting is new, this approach warrants more careful characterizations.*

**Response:** Thanks very much for pointing this issue out. According to the reviewer's
 suggestion, the agar plate replicates and quantification of LGG have been performed.
 To explore the tumor targeting ability of LGG, the tumor tissues and major organs of
 4T1 tumor-bearing mice were homogenized and smeared at different time points. By
 counting the colony forming units (CFU) in each plate, the results show that the
 amount of LGG has a trend to decrease after 24 h in both tumor and liver. However,
 what is even more remarkable than that is the decreased trend is relatively slight in
 tumor compared to the liver. In addition, the CFU amount of LGG in the tumor was
 significantly higher than the liver at 72 h. which further supports the superior tumor
 targeting and penetration ability of LGG (**Supplementary Fig. 6a, b**). The related
 data and discussion have been added in the Revised Manuscript (**Line 273-282 ,Page**
 **9, Revised Manuscript**).

Furthermore, we are extremely regretful for the error in the description and
typography of S14 in the manuscript, which in the original manuscript was about the
exploration of LGG alone tumor targeting and was mistyped as “LGG-MHS” in the
manuscript. The modified data are shown in **Supplementary Fig. 6b**.

**Supplementary Figure 6.** (a) Representative photographs of MRS agar plates and (b)
corresponding quantitative analysis of bacterial colonization in various organs and tumor of 4T1-
bearing mice in a different time (0, 2, 6, 24, and 72 h) ($n = 3$).

7. Second, there are lack of description regarding the statistical analyses performed
and some interpretations of the data are questionable. In Fig. 6, the figure caption
describes the statistical test as *t*-test, but these data include multiple groups and
timepoints which cannot be analyzed with the *t*-test.

**Response:** Thanks very much for pointing this issue out. We apologize for the
inappropriate analysis methods used in the data counts. We have re-run the statistical
analysis using appropriate statistical methods for all data. We have added the
following brief description in the Revised Manuscript which reads: “GraphPad Prism

(version 9.0.0, GraphPad Software, San Diego, California USA) was employed to
calculate all statistical analyses. Tumor growth curves were analyzed using two-way
ANOVA. Dunnett's multiple comparisons post test was utilized to analyze
hematological indexes. And for other comparisons, unpaired Student's t-test was used
when comparing two groups and one-way ANOVA with Holm Sidak correction for
multiple testing was used when comparing more than two groups. The p-value less
than 0.05 was considered significant (* $p < 0.05$, ** $p < 0.01$, *** $p < 0.001$, **** $p <$
0.0001)." (Line 723-729, Page 24, Revised Manuscript)

*8. Third, some key controls are missing from several experiments. For example, the*
*authors included some combination of their systems to compare their efficacy, but*
*LGG+US is missing from the experimental groups. Is it possible that LGG+US is just*
*as effective as LGG-MHS+UG?*

**Response:** Thank you for your kind comments. In order to better represent the
efficacy of each component in tumor treatment, we added two group animal models,
which named LGG-MH+US (without CRISPR/Cas9 system) and LGG-MHI+US (the
I in MHI is the IDO small molecule inhibitor NLG919). Mice were randomly divided
into 8 groups, which including Control, LGG, MHS, LGG-MHS, MHS + US, LGG-
MH + US, LGG-MHI + US and LGG-MHS + US. As a result, the LGG-MHS + US
group showed excellent ability to inhibit tumor growth and against lung metastasis
compared to other groups. (Figure 6-8, Revised Manuscript) The related data and
discussion have been added in the Revised Manuscript. (Line 356-371, Page 11-12,
Revised Manuscript)

**Fig. 6 LGG-MHS + US against 4T1 tumor *in vivo*.** (a) Schematic diagram of primary tumor
 treatment process *in vivo*. (b) Tumor growth curves of 4T1 after being treated by PBS, LGG, MHS,
 LGG-MHS, MHS + US, LGG-MH + US, LGG-MHI + US and LGG-MHS + US ($n = 5$). (c)
 Average tumor growth curves in different groups ($n = 5$). (d) HPLC assay of the Trp content in
 primary tumors and TDLNs of tumor-bearing mice after different treatments ($n = 3$). (e) Elisa
 assay Kyn content in primary tumors and TDLNs of tumor-bearing mice after different treatments
 ($n = 3$). (f) Antigen Ki-67 staining in tumor sections from each experiment group ($n = 3$). (g)
 Images and (h) corresponding fluorescence intensity of IDO immunofluorescence staining in
 primary tumors of 4T1 tumor-bearing mice after various treatments. DAPI was used to stain the

nucleus of the cell (blue), and the IDO was stained with anti-IDO antibodies (red) ($n = 3$). (i)
 Average tumor growth curves after being treated by re-challenge. ($n_{LGG-MHI + US} = 2, n_{LGG-MHS + US}$
 $= 4$)

**Fig. 7. Reprogramming of the tumor immunosuppressive microenvironment by the self-driven**
 **LGG-MHS + US nanosystem.** (a) Typical flow cytometric of mature DCs in tumor tissue after 24
 322 h after the first different treatments ($n = 3$). (b) Typical flow cytometric of T cells of CD4⁺ and
 323 CD8⁺ T cells in the spleen after 24 h after the first different treatments ($n = 3$). (c) Typical flow
 cytometric of Tregs in primary tumor tissue after 24 h after the first different treatments ($n = 3$). (d)
 Representative flow cytometric of M2 macrophages in spleen after 24 h after the first different
 treatments ($n = 3$). (e) Immunofluorescence images of helper T lymphocytes (CD3⁺CD4⁺) and

proliferated cytotoxic T lymphocytes (CD3⁺CD8⁺) in primary 4T1 tumor tissue slices ($n = 3$). (f-i)
 Levels of the IL-2, IL-12p70, IFN- α , and TNF- γ in primary tumor tissues after 24 h after the first
 different treatments ($n = 3$).

**Fig. 8 Anti distal tumor effect and immunological memory of LGG-MHS + US in the 4T1**
 **bearing mice model.** (a) Schematic diagram of the establishment of distal tumors model and the
 experimental procedure of treatment. (b) Average tumor growth curves of primary tumor in
 different groups ($n = 5$). * $P < 0.05$, ** $P < 0.01$, *** $P < 0.001$, **** $P < 0.0001$. (c) Mean growth
 curves and (d) corresponding growth curves of distant tumors in different groups ($n = 5$). * $P <$
 0.05 , ** $P < 0.01$, *** $P < 0.001$, **** $P < 0.0001$. (e) Immunofluorescence images of helper T
 lymphocytes (CD3⁺CD4⁺) and proliferated cytotoxic T lymphocytes (CD3⁺CD8⁺) in 4T1 tumor
 tissue slices of distal tumor ($n = 3$). (f) Schematic diagram of the establishment and treatment
 process of mouse models of lung metastasis. (g) Typical flow cytometric of the effector memory T

cells (CD3⁺CD8⁺CD44⁺CD62L⁻) (Tem) and (CD3⁺CD8⁺CD44⁺CD62L⁺) (Tcm) in the spleen
after 24 h after the first different treatments ($n = 3$). (h) Bioluminescence images and (i)
corresponding fluorescence intensity quantification of lung metastatic nodules of the 4T1 tumors
($n = 3$). (j) HE staining of lung tissue from different groups of 4T1 tumor-bearing mice. The
nodules with yellow circles in the section diagram indicate metastases in the lungs.

*9. Lastly, some details on experimental settings are missing which makes it difficult to*
*properly assess what was done. In the animal experiments, it is not clear how the*
*treatment was performed. How were LGG-MHS administered and how many times? If*
*the LGG-MHS was administered systemically, did they also get to the distal tumors in*
*Fig. 8 experiments? The possibility of LGG-MHS colonization and its effect on distal*
*tumors needs to be excluded before the authors claim the contribution of systemic*
*immunity.*

**Response:** Thank you very much for pointing this issue out. We administered LGG-
MHS by tail vein injection. Following injection of tumor cells into the right axilla of
the mice on day 0, LGG-MHS was injected on days 7, 9, 11 and 13. The mice were
treated with US irradiation in several groups on the 8th, 10th, 12th and 14th days. The
experimental details have been provided in the Revised Manuscript according to the
reviewer's kind suggestions, which reads: "4T1 tumor cells (1×10^6) were injected
into the axillary of female Balb/c mice (~20 g) to establish a xenograft tumor model.
These mice were divided at random into 8 groups (*per* group, $n = 5$): control (200 μ L,
PBS), LGG (200 μ L, LGG = 1×10^7 CFU), MHS (200 μ L, MHS = 10 mg/kg), LGG-
MHS (200 μ L, LGG = 1×10^7 CFU, MHS = 10 mg/kg), MHS + US (200 μ L, MHS =
10 mg/kg, US = 1.0 MHz, 1.0 W/cm², 50% duty cycle, 5 min), LGG-MH + US (200
μ L, LGG = 1×10^7 CFU, MH = 10 mg/kg, US = 1.0 MHz, 1.0 W/cm², 50% duty
cycle, 5 min), LGG-MHI + US (200 μ L, LGG = 1×10^7 CFU, MHI = 10 mg/kg, US =
1.0 MHz, 1.0 W/cm², 50% duty cycle, 5 min), LGG-MHS + US (200 μ L, LGG = $1 \times$
10^7 CFU, MHS = 10 mg/kg, US = 1.0 MHz, 1.0 W/cm², 50% duty cycle, 5 min). The
above drugs were injected on days 7, 9, 11, and 13, respectively, and the treatment
groups with US application were irradiated with US on days 8, 10, 12, and 14,

respectively. Tumor volume and body weight of mice were measured every 2 days
during days 7-21. Calculate the tumor volume according to the formula (tumor length)
373 \times (tumor width)²/2.” (Line 662-674, Page 22, Revised Manuscript).

Since the LGG-MHS nanosystem is administered systemically *via* tail vein
injection, some of the LGG-MHS is bound to enter the distal tumor site as well due to
hypoxia targeting properties of LGG. However, since the control variable is the
imposition of US, the distal tumors cannot produce ROS to trigger DAMPs to
promote immunotherapy, resulting in the distal tumors in the LGG-MHS group of
mice are not eliminated. (Fig 6, Page 37, Revised Manuscript).

*10. overall, while the amount of data presented in the manuscript is impressive, the*
*above points need to be addressed to properly assess the claims of the study.*

**Response:** All reviewers' concerns have been addressed. Finally, we are very grateful
for your comments and suggestions on our ideas and work, which are very important
for us to improve and revise the manuscript.

*Other points:*

*1.- There seems to be several typo, missing figure reference, and mislabeling that*
*should be corrected*

**Response:** Thank you very much for pointing this issue out. We have carefully
checked and corrected misspellings in the manuscript.

*2.- Reference #30 do not include studies supporting LGG colonization in tumors*

**Response:** Thank you very much for pointing this issue out. We have replaced

reference #30, citing the reported article on tumor-targeted therapy with Lactobacillus
as a reference to support our intent to apply LGG⁴⁴. (Line 89, Page 3, Revised
Manuscript).

3.- Reference #31 did not show LGG tumor colonization and local remodeling of the
microenvironment

**Response:** Thank you very much for pointing this issue out. Reference #31 in the
original manuscript has been removed.

4.- Line 132-134 is misleading. Fig. S3 shows some residual RNA by 3 hours. Where
are the replicates?

**Response:** Thank you very much for pointing this issue out. In order to clearly convey
what we were trying to express and to address the reviewers' concerns about
reproducibility, we improved the experiment by increasing the loading volume (300
409 ng) and extending the incubation time to 24 h, the experiments were performed three
410 times and quantified. The results of sgRNA stability are shown in Supplementary
Figure 2d-e. The sgRNA with MH remained stable after 12 h. On the contrary, the
free sgRNA was almost completely degraded, which further indicates that
Cas9/sgRNA can minimize degradation after being loaded by MH. The related data
and discussion have been added in the Revised Manuscript and Revised
Supplementary Information (Line 144-151, Page 5, Revised Manuscript;
Supplementary Figure 2d, e, Page 8, Revised Supplementary Information).

**Supplementary Figure 2.** (d) Agarose gel electrophoresis and (e) corresponding quantitative
analysis to evaluate the serum stability of naked Cas9/sgRNA and Cas9/sgRNA reconstituted from
MHS ($n = 3$).

*5.- Line 135-137: Is pH5 relevant to tumor microenvironment? How does this release*
*relate in the in vivo conditions, since the CRISPR complexes needs to be delivered to*
*intracellular regions? If it is released prematurely in the tumor microenvironment,*
*doesn't this reduce the efficacy?*

**Response:** Thank you very much for the kind question. It is shown that the tumor
microenvironment is slightly more acidic with a weak acidity of pH 6 to 7 relative to
normal tissue pH due to poor vascular perfusion, regional hypoxia and fermentative
glycolysis^{45, 46}. The intracellular pH of tumor cells can be even as low as 4 to 6⁴⁷⁻⁴⁹.
Therefore, we used pH=5 to simulate the acidic environment of intracellular
lysosomes in tumor cells to verify that the acidic microenvironment of lysosomes can
promote the release of ZIF-8-loaded CRIPR/Cas9⁵⁰. It was proven that the CRISPR
complex would be released in trace amounts in the weakly acidic tumor
microenvironment. The loss of trace amounts of CRISPR complexes due to premature
release was compensated by increasing the number of doses. Therefore, the reduction
in efficacy is negligible.

*6.- Fig. 3d label should be edited. It isn't showing % viability*

**Response:** Thank you very much for pointing this issue out. We have carefully
reviewed and edited the label. (**Fig. 3d, Page 32, Revised Manuscript**)

*7.- Line 224-226: where is the data supporting this claim?*

**Response:** Thank you very much for the kind question. After double-checking and

refining the content of the manuscript with regard to the reviewer's concerns, we have
confirmed that line 224-226 are notes to the original manuscript, Figure 3h, 3i. In the
Revised Manuscript, which reads: "(i) T7EI cleavage analysis after 4T1 cells with
different treatments, including control, US only, MH, MH + US, MHS and MHS +
US ($n = 3$). (j-k) Deep sequencing analysis of gene editing in 4T1 cells in the presence
of MHS and MHS+US." (Line 884-886, Page 33, Revised Manuscript)

8.- Fig. 4f and g: why did the authors just decided to look into IL-12p70 and IL-2
among all other cytokines?

**Response:** Thank you very much for the kind question. IL-2 is a pleiotropic cytokine
produced by T-cell antigen activation, also known as T-cell growth factor. It has been
shown that IL-2 mediates a range of immune effects by binding to IL-2 receptors on
the surface of lymphocytes, and that cellular responses in vivo are regulated by the
amount of IL-2 produced in response to antigens. The production of IL-2 receptors on
the surface of immune cells is stimulated and acts by autocrine or paracrine means.
The result is the expansion and activation of macrophages, natural killer cells, B
lymphocytes, *etc*⁵¹⁻⁵³.

IL-12p40 (p40) is known to be a subunit of the IL-12 cytokine family, which
binds to the p35 subunit to form IL-12p70 (IL-12). IL-12 has excellent antitumor
effects, for example, shifting CD4⁺ Th0 cells to a Th1 phenotype⁵⁴⁻⁵⁶, increasing
activated NK cells, the proliferation, survival and/or cytotoxic capacity of CD8⁺ and
CD4⁺ T cells⁵⁷ and programming T cells for optimal progression to effector memory T
cells⁵⁸, among others.

Therefore, we chose to study IL-2 and IL-12p70 among numerous cytokines to
validate the antitumor effects of MHS nanosystem.

9.- Fig. 5a and c: Are LGG and MHS both labeled with Cy5? If so, shouldn't one
expect much higher signal from LGG-MHS compared to LGG alone? It looks like they
are at similar levels, possibly suggesting that MHS is not getting to tumors

**Response:** Thank you very much for the kind question. We apologize for the lack of a
clear description of the experimental steps in the manuscript, which led to
misunderstanding by the reviewers. When using VISQUE imaging system to explore
the *in vivo* hypoxic targeting of LGG, we labeled MHS only with cy5.5 in the LGG-
MHS group, LGG was not labeled, and the amount of cy5.5 used in each group was
equal (10 μ g/mL). Therefore, there was no significant difference between the signal
of LGG-MHS compared with LGG alone. The related experimental details have been
provided in the Revised Manuscript according to the reviewer's kind question, which
reads: "Cy5.5-labeled MHS (200 μ L, Cy5.5-MHS = 10 mg/kg, Cy5.5 = 10 μ g/mL),
Cy5.5-labeled LGG (200 μ L, Cy5.5-LGG = 1×10^7 CFU, Cy5.5 = 10 μ g/mL) and
Cy5.5-labeled LGG-MHS (200 μ L, LGG = 1×10^7 CFU, Cy5.5-MHS = 10 mg/kg,
Cy5.5 = 10 μ g/mL) were intravenously injected into mice when the tumors volume
reached about 200 mm³. At various time points (0, 2, 4, 6, 8, 12, 48 and 72 h), mice
were anesthetized and imaged by VISQUE imaging system." (Line 638-642, Page 21,
**Revised Manuscript**)

10.- Since the authors had claimed the ability for MHS to remodel immunometabolism,
RNA sequencing result on LGG-MHS may be helpful to decipher the contribution on
TME remodeling from LGG alone vs LGG-MHS.

**Response:** Thank you very much for the kind comments and suggestions. We
apologize for the errors in the description and layout of the paper that caused some
confusion to the reviewers. Firstly, we have demonstrated the targeting of LGG and
then further explored the effect of LGG on the tumor microenvironment. Therefore,
the main purpose of conducting RNA sequencing was to investigate the potential

mechanism of LGG on tumor therapy. Subsequently we also demonstrated that LGG
after loading the MHS system, it still has favorable biological activity. Therefore,
LGG in the LGG-MHS system not only acts as a vector but also has a role in
activating the immune system. Additionally, we agree that the RNA sequencing
results of LGG-MHS could help decipher the contribution of LGG alone versus LGG-
MHS to TME remodeling, but the severe COVID-19 pandemic conditions in
Shanghai led to laboratory closing and prevented further access to in-depth studies.

In addition, the remodeling effect of MHS on tumor microenvironment was fully
investigated *in vitro*, and the general process is that the entry of MHS into tumor cells,
under US irradiation, triggers molecular damage related patterns, which in turn
promotes the infiltration of immune cells at the tumor site. Moreover, in the
subsequent animal experiments, we also included MHS as a separate group, so as to
investigate the contribution of MHS to tumor immunity *in vivo*.

*11.- Fig. S17: where are the data showing bacteria levels in tumors? This should also*
*be quantified with biological replicates.*

**Response:** Thank you very much for the kind questions and suggestions. Firstly, we
repeated the biological safety of LGG-MHS. The tumor tissues and major organs of
4T1 tumor-bearing mice were homogenized and smeared at different time points after
injection of the LGG-MHS (1, 3, 7, 30 d). Afterwards, we quantified LGG based on
the number of colonies in MRS agar plates and the weight of the tissue before
homogenization. The results showed that the heart, spleen, lung and kidney were free
of LGG growth except for minor LGG residues in the liver after 30 days of LGG-
MHS nanosystem injection (**Supplementary Fig. 7a, b**). The related experimental
results have been provided in the Revised Supplementary Information according to
the reviewer's kind suggestions. (**Supplementary Figure 7, Page 14, Revised**
**Supplementary Information**)

Supplementary Figure 7. (a) Representative photographs and (b) corresponding CFU count
 analysis of MRS agar plates of bacterial colonization in various organs of healthy mice in a month
 (1, 3, 7 and 30 days) ($n = 3$), Control *i.e.* without any treatment.

12.- Fig. S19 and S20: I don't see the control groups as claimed in the main text. The
 values seem to change over time – what statistics did the authors use to get the non-
 significance?

**Response:** Thank you very much for the kind questions and suggestions. We
 apologize for the difficulty in understanding the inappropriate description. We defined
 the 0 d as the control group (without any treatment), which serves as a reference value
 for comparison with other experimental groups (hematological indicators of mice on
 536 days 1, 3, 7, and 30 after LGG-MHS injection). Finally, Dunnett's multiple
 comparisons post test was used to test for significance between groups. The analysis
 shows that these values were not statistically significant. We have corrected the labels
 of the diagrams as detailed in Revised Supplementary Information. (**Supplementary**
 **Figure 7, Page 14, Revised Supplementary Information**)

**Supplementary Figure 7.** (d) *In vivo* hematological indices. Hematological assays of mice at 1, 3,
 7 and 30 days after LGG-MHS injection. Control *i.e.* without any treatment. ($n = 3$). (e) *In vivo*
 liver and kidney function index. Hematological assays of mice at 1, 3, 7 and 30 days after LGG-
 MHS injection ($n = 3$). Control *i.e.* without any treatment. * $P < 0.05$, ** $P < 0.01$, *** $P < 0.001$,
 **** $P < 0.0001$.

13.- Fig. 6: The authors indicated day 7, 9, 11, and 13 as treatment days. Is this LGG-
 MHS injections? Or is it US treatment? How are LGG-MHS administered?

**Response:** Thank you very much for the kind questions. Days 7, 9, 11, and 13 are the
 time points for administering LGG-MHS, while US irradiation is performed on days 8,
 10, 12, and 14. The details of LGG-MHS injection and application of US have added
 in the Revised Manuscript which reads: “4T1 tumor cells (1×10^6) were injected into
 the axillary of female Balb/c mice (~ 20 g) to establish a xenograft tumor model.

These mice were divided at random into 8 groups ($n = 15$): control (200 μ L, PBS),
LGG (200 μ L, LGG = 1×10^7 CFU), MHS (200 μ L, MHS = 10 mg/kg), LGG-MHS
(200 μ L, LGG = 1×10^7 CFU, MHS = 10 mg/kg), MHS + US (200 μ L, MHS = 10
558 mg/kg, US = 1.0 MHz, 1.0 W/cm², 50% duty cycle, 5 min), LGG-MH + US (200 μ L,
LGG = 1×10^7 CFU, MH = 10 mg/kg, US = 1.0 MHz, 1.0 W/cm², 50% duty cycle, 5
560 min), LGG-MHI + US (200 μ L, LGG = 1×10^7 CFU, MHI = 10 mg/kg, US = 1.0
561 MHz, 1.0 W/cm², 50% duty cycle, 5 min), LGG-MHS + US (200 μ L, LGG = 1×10^7
CFU, MHS = 10 mg/kg, US = 1.0 MHz, 1.0 W/cm², 50% duty cycle, 5 min). The
above drugs were injected on days 7, 9, 11, and 13, respectively, and the treatment
groups with US application were irradiated with US on days 8, 10, 12, and 14,
respectively. Tumor volume and body weight of mice were measured every 2 days
during days 7-21. Calculate the tumor volume according to the formula (tumor length)
567 \times (tumor width)²/2.” (Line 662-674, Page 22, Revised Manuscript).

*14- Fig. 8a: The inoculation of secondary tumors at mammary pad isn't strictly*
*metastatic model. I suggest editing the main text.*

**Response:** Thank you very much for pointing this issue out. we have added the
following brief description in the Revised Manuscript which reads: “**The immune**
**response against distant tumor.** 4T1 tumor cells (1×10^6) were injected into the
second left breast pad of the mice for 7 days as the primary tumor, and the second
right breast pad of each mouse was injected as a distant tumor (1×10^6 of 4T1 cells).”
(Line 693-696, Page 23, Revised Manuscript)

Response to reviewer #2

*The manuscript by Yu et al describes use of a microbial vector (lactobacillus,*
*LGG) and nanoparticle delivery system (MHS) activated by ultrasound (US)*
*irradiation to target and manipulate the tumour microenvironment (TME). The*
*authors used this approach to boost anti-tumour immunity in two ways by; (1)*
*stimulating reactive oxygen species (ROS) production following US and (2) using*
*LGG to deliver CRISPR/Cas9 gene editing functions to excise indoleamine 2,3*
*dioxygenase-1 (IDO1) genes, which mediate immune suppression via IDO enzyme*
*activity. TME-targeting efficacy was evaluated in a murine breast cancer cell line*
*(4T1) and the 4T1/BALB/c tumour model. Data reported largely support the authors'*
*claims that the LGG/MHS delivery system is an effective method to incite protective*
*anti-tumour immunity. The manuscript is generally well-written but would benefit*
*from increased clarity and focus on key biological findings of potential clinical*
*significance, and reduced emphasis on technical information such as nanoparticle*
*synthesis and validation data (see below).*

**Response:** Thank you very much for the positive comment and recommendation.
Please find the following detailed responses to your comments and suggestions.

Major Points:

*1. The Abstract does not clearly convey major findings from the study and would*
*benefit from extensive rewriting to enhance clarity, emphasise significant findings and*
*minimize technical information. In particular, more emphasis should be placed on*
*describing outcomes from experiments conducted using the mouse tumour model, as*
*these data are far more informative than studies performed on cell lines regarding*
*future prospects for clinical translation of the results from this study.*

**Response:** Thank you for your kind comments. According to the suggestions, we have

rewritten the abstract, which reads “Reprogramming the tumor immunosuppressive
microenvironment is a promising strategy for improving tumor immunotherapy
efficacy. The clustered regularly interspaced short palindromic repeat
(CRISPR)/CRISPR-associated protein 9 system is used to knockdown tumor
immunosuppression-related genes. Therefore, a self-driven multifunctional delivery
vector was constructed to efficiently deliver the CRISPR-Cas9 nanosystem for
indoleamine 2,3-dioxygenase-1 (*IDO1*) knockdown in order to amplify immunogenic
cell death (ICD) and then reverse tumor immunosuppression. *Lactobacillus*
*rhamnosus GG* (LGG) is a self-driven safety probiotic that can penetrate the hypoxic
tumor center, allowing efficient delivery of the CRISPR/Cas9 system to the tumor
region. While LGG efficiently colonizes the tumor area, and it also stimulates the
organism to activate the immune system. The CRISPR/Cas9 nanosystem can generate
abundant reactive oxygen species (ROS) under the ultrasound irradiation, resulting in
ICD, while the produced ROS can induce endosomal/lysosomal rupture and then
releasing Cas9/sgRNA to knock down the *IDO1* gene to lift immunosuppression. The
system generates powerful immune responses that effectively attack tumor cells in
mice, contributing to the inhibition of tumor metastasis *in vivo*. In addition, this
strategy provides a powerful immunological memory effect which offers protection
against tumor re-challenge after elimination.” (Line 26-41, Page 1-2, Revised
**Manuscript**)

2. *The initial description of the nanoparticle delivery system in the Introduction is*
*confusing (lines 86-103). In particular, the meaning of the acronym MHS needs*
*clarifying, as does the purpose of using ZIF8 and HMME in the strategy used in this*
*study. The graphic depicting study goals (Fig. 1) helps but is far too complicated. This*
*graphic should be simplified to focus exclusively on key elements of the strategy*
*employed in the study; in other words, make it into a graphic hypothesis.*

**Response:** Thank you very much for your kind comments and constructive

suggestions. Thank you very much for your kind comments and constructive
suggestions. The MHS consists of M (ZIF-8), H (sonosensitizer, HMME) and S
(Cas9/sgRNA), and ZIF-8 and HMME serve the following purposes:

(1) ZIF-8: Zeolitic imidazolinium framework (ZIF-8) is a metal-organic
framework with a large specific surface area, tailored pore size, pre-designed
morphology, biocompatibility and controlled degradability that brings such materials
closer to pharmaceutical and medical translation, allowing them to be used as an
excellent non-viral CRISPR/Cas9 delivery system.¹⁹⁻²² HMME and Cas9/sgRNA are
delivered into tumor cells *via* ZIF-8, and Cas9/sgRNA rapidly escapes from
endosomes/exosomes *via* the proton-sponge effect, thus enabling effective gene
editing.^{23,24}

(2) HMME: HMME was used as sonosensitizer to generates abundant ROS to
damage cancer cells upon US irradiation, while the generated ROS induce lysosomal
rupture to release Cas9/sgRNA, setting the stage for its next step of gene editing.
HMME, an organic acoustic sensitizer, leads to higher ROS and therefore produces a
more adequate SDT efficiency compared to inorganic acoustic sensitizers.²⁵⁻²⁷ More
importantly, HMME has been approved by the FDA for clinical use because of its
high safety profile as an sensitizer.²⁸ A more important point is that in our strategy
of synergistic immunotherapy strategy, HMME generates abundant ROS to damage
cancer cells upon US irradiation, while the generated ROS induce
endosomal/lysosomal rupture to release Cas9/sgRNA, disrupting oxidative stress
defense and facilitating the release of Cas9/sgRNA into the cytoplasm, setting the
stage for its next step of gene editing.

We have re-edited part of the introduction to explain the role of each component
separately to make it easier for the reader to understand our study, which reads
“Zeolitic imidazolinium framework (ZIF-8) is a metal-organic framework (MOF)
with a large specific surface area, tailored pore size, pre-designed morphology,
biocompatibility and controlled degradability that bring such materials closer to

pharmaceutical and medical translation³⁶. Hence, ZIF-8 (M) was used as an excellent
non-viral CRISPR/Cas9 delivery vehicle for delivery of the sonosensitizer
hematoporphyrin monomethyl ether (H) and CRISPR/Cas9 system (S), which named
as MHS.” (Line 98-103, Page 3-4, Revised Manuscript).

In addition, we have simplified Figure 1 to make it easier for the reader to
understand. (Page 29, Revised Manuscript)

**Fig. 1. Schematic of the LGG-MHS nanosystem delivery of CRISPR/Cas9 system for**
**reprogrammed the TIME *via* activation of immune response.** The use of a US-triggered
Cas9/sgRNA delivery system improved the efficiency of delivering Cas9/sgRNA to the nucleus of
tumor cells for gene editing.

3. The authors do not justify their choice of the 4T1 tumor model. Most importantly, is
the 4T1 model dependent on IDO activity for optimal 4T1 tumor growth? If not, this
undermines the strategy used and prompts the use of a tumor model known to be
dependent on IDO for optimal growth (eg. the LLC tumor model). Linked to this key
point, what is the authors' rationale for administering treatments when 4T1 tumours
were 200mm³?

**Response:** Thank you very much for your kind comments and question. Researches
have shown that *IDO1/TDO2* expression in the Cancer Genome Atlas (TCGA)
database is upregulated in TNBC compared to normal breast and skin tissue⁵⁹. It have
reported that inhibition of IDO function or reduction of Kyn production in 4T1 tumor-
bearing mice can effectively inhibit the 4T1 tumor growth⁶⁰⁻⁶². Therefore, it can be
concluded that *IDO1* is indeed overexpressed and closely associated with
tumorigenesis/progression in 4T1. IDO reduction and inhibition enhances
immunotherapy efficacy^{63, 64}.

In addition, we injected *IDO1* knockdown/overexpressing 4T1 cells into mouse
mammary pads to construct an *IDO1* knockdown/overexpressing Balb/c mouse
models, and monitored the tumor size from day 7-21 after injection. At the end of
monitoring, mice were euthanized and tumor tissue was collected for IDO protein
fluorescence staining and fluorescence quantification. The results show that the
results indicate that overexpression of *IDO1* significantly promotes the development
of the breast cancer (**Supplementary Fig. 1**). Therefore, *IDO1* is a potential target for
4T1 tumor therapy, which is promising to inhibit the growth in 4T1 by
downregulating *IDO1* levels. The related data and discussion have been added in the
Revised Supporting Information. (**Supplementary Figure 1, Page 7, Revised**
**Supplementary Information**).

Taken together, *IDO1* is a potential therapeutic target for 4T1 tumor. In this

regard, we have added the following brief explanation in the Revised Manuscript,
which reads “The Cancer Genome Atlas (TCGA) database analysis reveals that the
expression level of *IDO1* is significantly upregulated in triple-negative breast cancer
(TNBC) compared to normal breast tissue. Then, to explore the correlation between
the expression level of *IDO1* and the development of TNBC, we constructed stable
overexpression of *IDO1* and stable interference with *IDO1* in 4T1 cell lines and
constructed xenograft tumor models. The results indicate that overexpression of *IDO1*
significantly promotes the development of breast cancer (**Supplementary Figure 1**).
Therefore, reducing the expression level of *IDO1* contributed to inhibit the
proliferation of breast cancer. Accordingly, the CRISPR/Cas9 nanosystem developed
in this research can efficiently enrich the tumor region under probiotic drive and can
precisely and controllably knock down *IDO1* under ultrasound, avoiding the lack of
targeting and drug resistance of traditional inhibitors.” (**Line 490-499, Page15-16,**
**Revised Manuscript**)

We apologize for the typographical error in the manuscript, as shown in Figure 6
and Figure 8, we started treatment of the mice on day 7 after 4T1 tumor cell injection
with the tumor volume was approximately 60-80 mm³. We made the following
corrections to the manuscript, which reads “Mice were randomly divided into 8
groups once the tumor volume reached an approximate size of 60~80mm³, which
including Control, LGG, MHS, LGG-MHS, MHS + US, LGG-MH + US, LGG-MHI
+ US and LGG-MHS + US, and were treated on days 7-14.” (**Line 359-361, Page 11,**
**Revised Manuscript**)

**Supplementary Figure 1.** (a) Separate and integrated tumor growth curves ($n = 5$) and (b) Images
 of IDO immunofluorescence staining and corresponding mean fluorescence intensity of 4T1
 tumor-bearing mouse after being treated Control (without treating), OE (*IDO1* over expression
 plasmid), OE-Control (Untreated plasmid for OE), KD (*IDO1* knock down plasmid), KD-Control
 (Untreated plasmid for KD). DAPI was used to stain the nucleus of the cell (blue), and the IDO
 was stained with anti-IDO antibodies (red). ($n = 3$) * $P < 0.05$, ** $P < 0.01$, *** $P < 0.001$, **** $P <$
 0.0001 .

4. The authors must assess IDO enzyme activity by measuring kynurenine levels in the
 TME and draining lymph nodes to evaluate if their treatment strategy reduces
 nominal levels of IDO enzyme activity that may promote immune suppression required
 for optimal tumour growth. Note that assessing (1) *IDO1* protein expression or (2)
 Trp levels are not sufficient to measure IDO activity in the TME. Linked to this point,
 the authors should test if IDO inhibitors synergise with their nanoparticle approach to
 boost immune activation to assess if IDO inhibitors or LGG-CRISPR/cas9 gene
 editing is more effective in reducing IDO activity.

**Response:** Thank you for your constructive suggestions, which will help improve the
rigor of our study. We added LGG in combination with MH and IDO inhibitor (LGG-
MHI + US) groups to the animal model grouping to compare whether CRIPR is more
effective than IDO inhibitors in inhibiting tumor growth. Further, we examined the
levels of Trp and Kyn in the primary tumor and tumor draining lymph nodes to assess
the activity of IDO. The related data have been supplemented in the Revised
Manuscript, as shown in Revised Manuscript Figure 6d-e and g-h, we detected similar
levels of Trp and Kyn, and slightly different IDO fluorescence intensities in the LGG-
MHI+US and LGG-MHS+US groups, indicating that IDO inhibitors are similar to
CRISPR/Cas9 in inhibiting the activity of IDO proteins in primary tumors within a
short period of time. However, combined with our monitoring of tumor size and study
of tumors (**Figures 6b, c**), we found that the nanoplatform combined with IDO
inhibitors was therapeutically effective in eliminating primary tumor growth to some
extent (2/5), but its efficacy was inferior to that of CRISPR/Cas9 (4/5). Furthermore,
mice were re-challenged on day 60 by subcutaneous implantation of 3× 4T1 cells into
the left axilla (**Figure 6a**). For surviving mice that had been treated with LGG-MHS +
US, the second tumor challenge was rejected at a rate of 100%. Although mice treated
with LGG-MHI + US initially showed a 2/5 survival rate, tumor progression was
observed after tumor re-challenge, indicating inefficient development of adaptive
immune responses against 4T1 cells (**Figure 6i**). These results suggest that while the
IDO inhibitor combination LGG exhibited anti-tumor activity under US exposure, it
was less effective than CRISPR/Cas9 in triggering durable immunity. In addition,
CRISPR showed superior tumor suppression compared to IDO inhibitors in
suppressing distal and pulmonary metastases (**Figures 8**). This may be due to the
resistance of the organism to small molecule inhibitors.⁵⁹

**Fig. 6 LGG-MHS + US against 4T1 tumor *in vivo*.** (a) Schematic diagram of primary tumor
 treatment process *in vivo*. (b) Tumor growth curves of 4T1 after being treated by PBS, LGG, MHS,
 LGG-MHS, MHS + US, LGG-MH + US, LGG-MHI + US and LGG-MHS + US ($n = 5$). (c)
 Average tumor growth curves in different groups ($n = 5$). (d) HPLC assay of the Trp content in
 primary tumors and TDLNs of tumor-bearing mice after different treatments ($n = 3$). (e) Elisa
 assay Kyn content in primary tumors and TDLNs of tumor-bearing mice after different treatments
 ($n = 3$). (f) Antigen Ki-67 staining in tumor sections from each experiment group ($n = 3$). (g)
 Images and (h) corresponding fluorescence intensity of IDO immunofluorescence staining in
 primary tumors of 4T1 tumor-bearing mice after various treatments. DAPI was used to stain the
 nucleus of the cell (blue), and the IDO was stained with anti-IDO antibodies (red) ($n = 3$). (i)
 Average tumor growth curves after being treated by re-challenge. ($n_{LGG-MHI+US} = 2, n_{LGG-MHS+US}$
 $= 4$)

**Fig. 8 Anti distal tumor effect and immunological memory of LGG-MHS + US in the 4T1**
 **bearing mice model.** (a) Schematic diagram of the establishment of distal tumors model and the
 experimental procedure of treatment. (b) Average tumor growth curves of primary tumor in
 different groups ($n = 5$). $*P < 0.05$, $**P < 0.01$, $***P < 0.001$, $****P < 0.0001$. (c) Mean growth
 curves and (d) corresponding growth curves of distant tumors in different groups ($n = 5$). $*P <$
 0.05 , $**P < 0.01$, $***P < 0.001$, $****P < 0.0001$. (e) Immunofluorescence images of helper T
 lymphocytes ($CD3^+CD4^+$) and proliferated cytotoxic T lymphocytes ($CD3^+CD8^+$) in 4T1 tumor
 tissue slices of distal tumor ($n = 3$). (f) Schematic diagram of the establishment and treatment
 process of mouse models of lung metastasis. (g) Typical flow cytometric of the effector memory T
 cells ($CD3^+CD8^+CD44^+CD62L^-$) (Tem) and ($CD3^+CD8^+CD44^+CD62L^+$) (Tcm) in the spleen after
 24 h after the first different treatments ($n = 3$). (h) Bioluminescence images and (i) corresponding
 fluorescence intensity quantification of lung metastatic nodules of the 4T1 tumors ($n = 3$). (j) HE
 staining of lung tissue from different groups of 4T1 tumor-bearing mice. The nodules with yellow
 circles in the section diagram indicate metastases in the lungs.

*5. Data reported in Fig6 & Fig8 support the authors' conclusion that LGG-MHS+US*
*treatments reduced primary and distal 4T1 tumor burdens at experimental endpoints*
*(day 21). MHS+US treatments also reduced tumor burdens, though to a lesser extent.*
*These outcomes suggest that combining LGG with MHS/US nanotherapy may fully*
*protect against 4T1 tumor growth but more studies will be necessary to support this*
*claim rigorously, in particular with regard to if IDO1 gene editing is critical to*
*promote protective outcomes (see point 4). Accordingly, the authors should assess*
*mouse survival over longer periods and test if LGG infection or IDO1 gene editing*
*(or both) contribute to increased protection from 4T1 tumor growth, as well as*
*evaluating IDO enzyme activity (see point 4).*

**Response:** Thank you for your constructive suggestions, which will help to improve
the rigor of our research. According to reviewer's suggestion, we added the LGG-MH
+ US group to the animal models to explore the contribution made by *IDO1*
knockdown to inhibit tumor growth, and the related data have been supplemented in
the Revised Manuscript and Revised Supplementary Information, as shown in
Revised Manuscript Figure 6, the tumors of mice in the LGG-MH + US group did not
differ much from MHS+US, and simply inhibited tumor growth more mildly.

In addition, after referring to the extensive literature, we extended the survival
assessment of surviving mice in the LGG-MHI + US group and LGG-MHS + US
group to 60 days and reinoculated $3 \times 4T1$ cells into the left axilla on day 60 to verify
the ability of surviving mice to reject re-challenge. The related data have been
supplemented in the Revised Supplementary Information and Revised Manuscript,
which reads: "For the survivors that had been treated with LGG-MHS + US, the
second tumor challenge was rejected at a 100% rate. Though animals treated with the
LGG-MHI + US initially demonstrated 2/5 survival rate, all with tumor progression
observed after the tumor re-challenge, indicating inefficient development of an
adaptive immune response against 4T1 cells. These results show that, while IDO

inhibitor combination LGG exhibits antitumor activity under the US exposure, it is
 not as efficient as the CRISPR/Cas9 at eliciting long-lasting immunity.” (Line 397-
 403, Page 12-13, Revised Manuscript and Supplementary Figure 9b, Page16,
 Revised Supplementary Information)

**Supplementary Figure 9.** (b) survival curves of 4T1-tumor-bearing mice with different treatment
 (control, US only, MH, MH + US, MHS, and MHS + US) ($n = 5$). $*P < 0.05$, $**P < 0.01$, $***P <$
 0.001 , $****P < 0.0001$.

**Fig. 6 LGG-MHS + US against 4T1 tumor *in vivo*.** (i) Average tumor growth curves after being
 treated by re-challenge. ($n_{\text{LGG-MHI + US}} = 2$, $n_{\text{LGG-MHS + US}} = 4$)

6. The tumor re-challenge strategy depicted in Fig8h indicates that primary 4T1

*tumors were surgically resected on day 21. It is not clear why tumors were resected.*
*Tumor re-challenge should be conducted by injecting 4T1 tumor cells into mice that*
*survive primary 4T1 tumor growth after therapy without resecting primary tumors*
*prior to re-challenge to evaluate if therapy stimulates durable and stable anti-tumor*
*immunity that clears both primary and secondary tumors.*

**Response:** Thank you very much for the constructive suggestions, which are highly
appreciated. Our results above showed that most groups of tumor-bearing mice
survived less than 60 days with different treatments. In order to explore the long-term
immunological memory effect of treated mice, we had to extend the survival period of
the mice. Therefore, at the termination of treatment on day 21, we performed tumor
resection on all mice with tumors still present.

According to reviewer's suggestion, we have improved the experimental
protocol of primary tumor model. Survival assessment of surviving mice were
extended the in the LGG-MHI and LGG-MHS groups and reinoculated 4T1 cells two
847 weeks after the end of treatment (day 60) to exploring the ability to anti-rechallenge
of treated mice. Though animals treated with the LGG-MHI + US initially
demonstrated 2/5 survival rate, all with tumor progression observed after the tumor
re-challenge, indicating inefficient development of an adaptive immune response
against 4T1 cells. These results show that, while IDO inhibitor combination LGG
exhibits antitumor activity under the US exposure, it is not as efficient as the
CRISPR/Cas9 at eliciting long-lasting immunity. The related data have been
supplemented in the Revised Manuscript. (**Figure 6, Page38-39, Revised**
**Manuscript**)

**Fig. 6 LGG-MHS + US against 4T1 tumor *in vivo*.** (i) Average tumor growth curves after being
 treated by re-challenge. ($n_{\text{LGG-MHI+US}} = 2$, $n_{\text{LGG-MHS+US}} = 4$)

7. The short Discussion (lines 513 – 525) does not adequately describe the relevance
 and significance of the study findings, or place them in the context of the current
 scientific literature. This section needs extensive rewriting to address these
 deficiencies.

**Response:** Thank you very much for the constructive suggestions. We have added
 related description and discussion in the Revised Manuscript, which reads:
 “Immunotherapy has become an effective therapeutic modality for tumors instead of
 surgery, radiotherapy, chemotherapy and targeted therapy through activation or
 modulation of the organism immune system. However, due to the existence of tumor
 immunosuppressive microenvironment (hypoxia, low pH, immunosuppressive cell
 infiltration, *etc.*) limits the effectiveness of immunotherapy. In particular, IDO is a
 potential small molecule immune checkpoint which is overexpressed in a variety of
 tumor tissues and serves as an immunosuppressive factor to induce immune tolerance
 and immune escape in the organism's immune system. The Cancer Genome Atlas

(TCGA) database analysis reveals that the expression level of *IDO1* is significantly
upregulated in triple-negative breast cancer (TNBC) compared to normal breast tissue.
Then, to explore the correlation between the expression level of *IDO1* and the
development of TNBC, we constructed stable overexpression of *IDO1* and stable
interference with *IDO1* in 4T1 cell lines and constructed xenograft tumor models. The
results indicate that overexpression of *IDO1* significantly promotes the development
of breast cancer. Therefore, reducing the expression level of *IDO1* contributed to
inhibit the proliferation of breast cancer. Accordingly, the CRISPR/Cas9 nanosystem
developed in this research can efficiently enrich the tumor region under probiotic
drive and can precisely and controllably knock down *IDO1* under US irradiation,
avoiding the lack of targeting and drug resistance of traditional inhibitors.

In addition, hypoxia plays a crucial role in the tumor immunosuppressive
microenvironment and largely influences the outcome of treatment. Given the critical
role of hypoxia in tumor progression and its resistance to treatment, many efforts have
been made to overcome the limitations associated with hypoxia regarding tumors. In
contrast to traditional strategies of overcoming hypoxia, the present research exploited
the hypoxic microenvironment of tumors and utilized the hypoxia-driven and
colonization properties of LGG as a vector for delivery of the CRISPR/Cas9
nanosystem. After our study, we found that LGG does have an excellent ability to
target the hypoxic microenvironment of tumors. In vivo fluorescence images and
semi-quantitative analysis indicate that the fluorescent intensity of Cy5.5 at the tumor
site increased over time after intravenous injection of LGG-Cy5.5 and LGG-MHS-
Cy5.5, revealing superior tumor targeting properties of the LGG-MHS complex.
Meanwhile, it has been revealed that LGG is not only a vehicle but also a synergistic
therapeutic adjuvant. LGG can inhibit tumor cell growth and metastasis by activating
the immune response through certain specific pathways and increasing the infiltration
of immune cells in the tumor microenvironment.

The system generates powerful immune responses that effectively attack tumor
cells in mice, contributing to the inhibition of tumor metastasis *in vivo*. In addition,

this strategy provides a powerful immunological memory effect which offers
protection against tumor re-challenge after elimination. In summary, a self-driven
probiotic delivery system for CRISPR/Cas9 was constructed in order to reprogram the
TIME and then inhibit metastasis and recurrence of breast cancer. This system
employs *Lactobacillus rhamnosus* as a carrier for the efficient delivery of the
CRISPR/Cas9 nanosystem to knock down *IDO1*, reduce immunosuppressive cells
infiltration, and activate intrinsic immunity by regulating signaling pathways
associated with immune response and apoptosis. Meanwhile, the system is triggered
by US to improve gene editing efficiency and induce ICD, while the molecular
damage-related proteins released during ICD are taken up by immature DCs as
antigens to promote their maturation and thus upregulation of killer T cells. Immune
cells are efficiently activated through this cocktail therapy to eliminate the primary
tumor and inhibit its metastasis and recurrence. This research not only reprogram the
TIME with multiple pathways to activate the immune system against tumors, but also
developed a synergistic gene editing therapeutic modality based on a unique
CRISPR/Cas9 gene delivery technology, which is undoubtedly crucial for further
clinical applications of gene editing technology *in vivo*.” (Line 483-530, Page 15-17,
**Revised Manuscript**)

**Minor Point:**

*1. The large number of supplemental figures (33) make the manuscript difficult to*
*read. The authors should consult with the editors to find ways to streamline this large*
*set of supplemental figures.*

**Response:** Thank you very much for pointing this issue out. We have rearranged the
supplementary figures (12) to make them easier to read and understand.

Response to reviewer #3

*The paper entitled “Self-driven Probiotic-CRISPR/Cas9 Nanosystem Reprogramming*
*of Tumor Immunosuppressive Microenvironment to Enable Sono-immunometabolic*
*Cancer Therapy” is reporting the use of a multifunctional immunotherapeutic system*
*for solid tumor treatment. They loaded the sonosensitizer hematoporphyrin*
*monomethyl ether (HMME) and CRISPR/CAS9 on ZIF-8 (MHS) and combined them*
*with Lactobacillus rhamnosus GG (LGG) for enhancing immunotherapy efficacy.*
*LGG bacteria was used as a carrier for in vivo study to increase the targetability of*
*the system toward tumors. The system consisted of ZIF-8 which was used as a vector*
*to protect Cas9/sgRNA, HMME was used to generate ROS under ultrasound*
*irradiation (US) to induce lysosomal rupture and release Cas9/sgRNA which is*
*intended to knock down the IDO1 gene and promote immunogenic cell death (ICD).*
*They tested the efficacy of the system in both, in vitro and in vivo. It is evident that*
*they tried to evaluate the efficiency of their system using different experimental*
*approaches. While the in vivo results looked promising, they did not provide a clear*
*conclusion about the advantage of each individual component of the system and its*
*role in the success of the treatment. They lack many control experiments which made*
*the data presented inexplicit. Therefore, acceptance can be recommended at this stage.*
*The following comments need to be addressed to have a better understanding of their*
*system.*

**Response:** Thank you very much for your kind comments. We have clarified in the
discussion section of the Revised Manuscript about the advantages of each component
of our LGG-MHS nanosystem, which reads: “Immunotherapy has become an
effective therapeutic modality for tumors instead of surgery, radiotherapy,
chemotherapy and targeted therapy through activation or modulation of the organism
immune system. However, due to the existence of tumor immunosuppressive
microenvironment (hypoxia, low pH, immunosuppressive cell infiltration, etc.) limits
the effectiveness of immunotherapy. In particularly, IDO is a potential small molecule

immune checkpoint which is overexpressed in a variety of tumor tissues and serves as
an immunosuppressive factor to induce immune tolerance and immune escape in the
organism's immune system. The Cancer Genome Atlas (TCGA) database analysis
reveals that the expression level of *IDO1* is significantly upregulated in triple-
negative breast cancer (TNBC) compared to normal breast tissue. Then, to explore the
correlation between the expression level of *IDO1* and the development of TNBC, we
constructed stable overexpression of *IDO1* and stable interference with *IDO1* in 4T1
cell lines and constructed xenograft tumor models. The results indicate that
overexpression of *IDO1* significantly promotes the development of breast cancer.
Therefore, reducing the expression level of *IDO1* contributed to inhibit the
proliferation of breast cancer. Accordingly, the CRISPR/Cas9 nanosystem developed
in this research can efficiently enrich the tumor region under probiotic drive and can
precisely and controllably knock down *IDO1* under US irradiation, avoiding the lack
of targeting and drug resistance of traditional inhibitors.

In addition, hypoxia plays a crucial role in the tumor immunosuppressive
microenvironment and largely influences the outcome of treatment. Given the critical
role of hypoxia in tumor progression and its resistance to treatment, many efforts have
been made to overcome the limitations associated with hypoxia regarding tumors. In
contrast to traditional strategies of overcoming hypoxia, the present research exploited
the hypoxic microenvironment of tumors and utilized the hypoxia-driven and
colonization properties of LGG as a vector for delivery of the CRISPR/Cas9
nanosystem. After our study, we found that LGG have an excellent ability to target
the hypoxic microenvironment of tumors. In vivo fluorescence images and semi-
quantitative analysis indicate that the fluorescent intensity of Cy5.5 at the tumor site
increased over time after intravenous injection of LGG-Cy5.5 and LGG-MHS-Cy5.5,
revealing superior tumor targeting properties of the LGG-MHS complex. Meanwhile,
it has been revealed that LGG is not only a vehicle but also a synergistic therapeutic
adjuvant. LGG can inhibit tumor cell growth and metastasis by activating the immune

response through certain specific pathways and increasing the infiltration of immune
cells in the tumor microenvironment.

The system generates powerful immune responses that effectively attack tumor
cells in mice, contributing to the inhibition of tumor metastasis *in vivo*. In addition,
this strategy provides a powerful immunological memory effect which offers
protection against tumor re-challenge after elimination. In summary, a self-driven
probiotic delivery system for CRISPR/Cas9 was constructed in order to reprogram the
TIME and then inhibit metastasis and recurrence of breast cancer. This system
employs *Lactobacillus rhamnosus* as a carrier for the efficient delivery of the
CRISPR/Cas9 nanosystem to knock down *IDO1*, reduce immunosuppressive cells
infiltration, and activate intrinsic immunity by regulating signaling pathways
associated with immune response and apoptosis. Meanwhile, the system is triggered
by US to improve gene editing efficiency and induce ICD, while the molecular
damage-related proteins released during ICD are taken up by immature DCs as
antigens to promote their maturation and thus upregulation of killer T cells. Immune
cells are efficiently activated through this cocktail therapy to eliminate the primary
tumor and inhibit its metastasis and recurrence. This research not only reprogram the
TIME with multiple pathways to activate the immune system against tumors, but also
developed a synergistic gene editing therapeutic modality based on a unique
CRISPR/Cas9 gene delivery technology, which is undoubtedly crucial for further
clinical applications of gene editing technology *in vivo*.” (Line 483-530, Page 15-17,
**Revised Manuscript**)

We apologize for the absence of many control experiments, and we have added
the appropriate control experiments for your concerns, please find the following
detailed responses.

1. For the construct assembly, it was not clear how HMME was loaded into ZIF-8.

*What type of interaction is happening? The same for Cas9/sgRNA, did it infiltrate*
*ZIF-8 or did they form a complex?*

**Response:** Thanks very much for your kind question. Metal-organic frameworks
(MOFs), consisting of metal or cluster nodes linked by organic ligands, have emerged
as a promising platform for biomedical applications due to their highly porous
structure, friendliness to various functionalization methods, and excellent
biocompatibility and biodegradability^{60, 61}.

There are mainly three methods for various drugs/large/small molecules binding
to MOFs: grafting, permeation and encapsulation²⁰. It has been reported that
biomacromolecules such as enzymes may be encapsulated within MOFs *via* two
general strategies: by assembling the MOF around the enzyme (which term *de novo*
encapsulation) or by introducing the enzyme into the pre-existing MOF (which term
*post-synthetic* encapsulation). Zinc 2-methylimidazole (ZIF-8), a nanoscale metal –
organic framework with excellent biocompatibility, has unique features in
biomacromolecules condensing and chemical drug-loading efficiency due to its
positive charge and high surface ratio. More importantly, the acidic environment of
endosomes and/or lysosomes can trigger the degradation of ZIF-8 hosts, which can
facilitate cargo escape from endosomes and/or lysosomes to the cytosol^{61, 62}.

Our strategy firstly employs one-step encapsulation approach to encapsulate
HMME into the interior of ZIF-8. The HMME was dropwise into the
dimethylimidazole solution stirred for 10 min before the addition of zinc nitrate
hexahydrate. The material after encapsulating HMME with ZIF-8 (MOF) is named
MH. Second, MH was incubated with Cas9/sgRNA to form MHS. The detailed MHS
experimental procedure and results been provided in the Revised Manuscript
according to the reviewer's kind question, which reads: "Hematoporphyrin
monomethyl ether (HMME, 200 μ L, 2 mg/mL) was slowly added to 2-
methylimidazole solution under mechanical stirring at room temperature, and after 10
1039 min, zinc nitrate solution was added dropwise. The MH was obtained after stirring for

24 h at room temperature. Then, the MH and CRISPR/Cas9 system (mass ratio 4:1)
were incubated at 37 ° C according to the methodology instructions, finally, the
integration of MHS nanosystem was constructed.” (Line 569-574, Page 19, Revised
**Manuscript**) In summary, HMME is encapsulated into the interior of ZIF-8 during
the synthesis process. In contrast, Cas9/sgRNA is partially internalized into the
interior of MH and partially grafted onto the surface of MH after incubation with MH,
resulting in MHS.

2. *The illustration and the terms “loading” and “encapsulation” are not very*
*accurate. The author claimed the loading/ encapsulation of Cas9/sgRNA into ZIF-8,*
*however, the reported pore size of ZIF-8 is very small for Cas9/sgRNA to internalize.*

**Response:** Thank you very much for pointing this issue out. Illustrations and term
have been corrected. It has been reported that biomacromolecules such as enzymes
may be encapsulated within MOFs *via* two general strategies by assembling the MOF
around the enzyme (which term *de novo* encapsulation) or by introducing the enzyme
into the pre-existing MOF (which term *post-synthetic* encapsulation). Zinc 2-
methylimidazole (ZIF-8), a nanoscale metal - organic framework with excellent
biocompatibility, has unique features in biomacromolecules condensing and chemical
drug-loading efficiency due to its positive charge and high surface ratio.. Thus, our
strategy firstly employs one-step encapsulation approach to encapsulate HMME into
the interior of ZIF-8. The material after encapsulating HMME with ZIF-8 (MOF) is
named MH. Second, MH was incubated with Cas9/sgRNA to form MHS. Revised
Manuscript Figure 2b and Supplementary Figure 2c show that the average pore size of
MHS decreased relative to ZIF-8, demonstrating that some Cas9/sgRNA penetrated
into the interior of MH. Revised Manuscript Figure 2e shows that the particle size of
MHS slightly increases compared to MH, which proves that some Cas9/sgRNA is
also grafted on the surface of MH. Finally, Revised Manuscript Figure 2d
demonstrates that the elemental mapping of MHS corresponds to a more dense P-

element compared to ZIF-8 and MH, which further suggests MH was successfully
loaded with Cas9/sgRNA. Therefore, the final MH and Cas9/sgRNA formed the MHS
complex.

*3. In figure 2C, how did ZIF-8 maintain its hexagonal structure after combining it*
*with HMME and CRISPR/CAS9? and the size increase after complexation has to be*
*justified.*

**Response:** Thank you very much for the kind question and constructive suggestion.
Our strategy firstly employs one-step encapsulation approach to encapsulate HMME
into the interior of ZIF-8. The HMME was dropwise into the dimethylimidazole
solution stirred for 10 min before the addition of zinc nitrate hexahydrate. The
material after encapsulating HMME with ZIF-8 (MOF) is named MH. Second, MH
was incubated with Cas9/sgRNA to form MHS. So that the encapsulated HMME still
maintain their hexagonal structure.

Related studies have shown that the crystalline growth process of ZIF-8 crystals
includes four processes: nucleation, crystallization, growth, and stabilization⁶³. excess
2-methylimidazole deprotonates and zinc ions coordinate to form nuclei, then the
nuclei grow rapidly to form ZIF-8 nanocrystal particles, and finally neutral 2
methylimidazole combined with positively charged ZIF-8 to terminate the reaction.⁶⁴
It has been shown that the particle size of ZIF-8 increases with the increase of the
amount of encapsulated material^{50, 62}. Because we added MHHE to the
dimethylimidazole solution before adding zinc nitrate hexahydrate and stirred for 10
1091 min to prepare MH, the larger nuclei would result in a particle size of MH larger than
1092 ZIF-8.

4. The elemental mapping (EM) in figure 2i does not correspond to the TEM image of
LGG-MHS in 2h. It is better to compare it to the elemental mapping of LGG alone
and compare the EM of MHS to ZIF-8 alone using the same experimental settings.

**Response:** Thank you very much for the kind question and constructive suggestion.
According to the reviewer's suggestion, we have revalidated TEM characteristics
under the same conditions. As shown in Fig. 2c, there is no changes in nanoparticles
morphology of ZIF-8, MH and MHS except for the slightly increase in particle size of
MH and MHS compared to ZIF-8. The elemental profile corresponds to a denser P
element within MHS than ZIF-8 and MH, which further suggests that Cas9/sgRNA
was successfully loaded into MH. (**Fig. 2d**)." In addition, TEM results show that the
LGG surface was not smooth with numerous nanoparticles attached after
compounding. The corresponding elemental mapping reveals the presence of more Zn
elements on the surface of LGG, which further implies that LGG was successfully
compounded with the MHS nanosystem (**Fig. 2h**). The related data and results have
been added in the Revised Manuscript. (**Line 162-165, Page 5, Revised Manuscript**)

**Fig. 2 Synthesis and structural characterization of ZIF-8, MH, MHS, LGG and LGG-MHS.**
(d) Elemental mappings of ZIF-8, MH and MHS. (h) Transmission electron microscopic (TEM)
and corresponding elemental mappings of LGG and LGG-MHS.

*5. In line 120, they mentioned “utilizing sodium dodecyl sulfate-polyacrylamide gel*
*electrophoresis (SDS-PAGE)”*, however, figure S1 shows an agarose gel of the sgRNA
*only. Therefore, they need to show the loading of the different mass ratios of MH to*
*Cas9 used in order to obtain the optimal loading concentration.*

**Response:** Thank you very much for your kind comments. We sincerely apologize for
the error in our wording in line 122 of the manuscript, we did use agarose gel
electrophoresis to explore the optimal mass ratio for MH loading Cas9/sgRNA. Due
to the ratio of Cas9 to sgRNA being fixed, we preincubated CRISPR-Cas9 system to
sgRNA in a 1:1 molar ratio to synthesize RNP according to the product specification
(Cat# 1081058, IDT). And then, different mass ratios of MH to Cas9/sgRNA (MH:
sgRNA of 0, 2, 4, 6, 8, 10, 12) were used to prepare MHS in order to achieve optimal
Cas9/sgRNA loading efficiency. The outcome shows that a ratio of 4 for MH:
Cas9/sgRNA result in the optimal loading efficiency of Cas9/sgRNA.

We apologize for the unclear description in the manuscript, and it has been
corrected in the Revised Manuscript, which reads: “Different mass ratios of MH to
Cas9/sgRNA were used to prepare MHS in order to achieve optimal Cas9/sgRNA
loading efficiency, and the amount of sgRNA in the nanosystem was determined
utilizing agarose gel electrophoresis (AGE) (**Supplementary Fig. 2a**). The outcome
shows that a ratio of 4 for MH: Cas9/sgRNA results in the optimal loading efficiency
of Cas9/sgRNA.” (**Line 127-131, Page 4, Revised Manuscript**)

*6. Figure S3, the MHS stability experiment has to be conducted after 12, 24hrs, since*
*the system is incubated with the cells for 24 hrs. Also, running the same experiment*

on SDS PAGE with free Cas9/sgRNA would show the stability of Cas9 as well.

**Response:** Thank you very much for your constructive suggestions. According to the
reviewer's suggestion, we have improved the experimental method by incubating
Cas9/sgRNA and MHS in 10% serum for 0 h, 6 h, 12 h and 24 h before performing
agarose gel electrophoresis. In addition, we also performed electrophoresis on SDS-
PAGE for the Cas9/sgRNA and MHS after the same treatment to explore the stability
of Cas9. The results of sgRNA stability are shown in Supplementary Figure 2d-e. The
sgRNA with MH remained stable after 12 h. On the contrary, the free sgRNA was
almost completely degraded, which further indicates that Cas9/sgRNA can minimize
degradation after being loaded by MH. And the stability of Cas9 protein was not
affected by either naked Cas9/sgRNA or MHS (**Supplementary Fig. 2f, g**). The
related data have been added in the Revised Supplementary Information.
(**Supplementary Figure 2, Page 8, Revised Supplementary Information**)

**Supplementary Figure 2.** (d) Agarose gel electrophoresis and (e) corresponding quantitative
analysis to evaluate the serum stability of Cas9/sgRNA and Cas9/sgRNA reconstituted from MHS
($n = 3$). (f) SDS-PAGE and (g) corresponding quantitative analysis to evaluate the serum stability
of Cas9/sgRNA and Cas9/sgRNA reconstituted from MHS ($n = 3$).

7. Figure 3a, the group measured the generated ROS after exposing MHS to US, but
they did not report the effect of US radiation on ZIF-8 alone and MH, and hence, the
reason for adding HMME would be justified.

**Response:** Thank you very much for your constructive suggestions. We refined the
experimental groups (including Control, US only, ZIF-8, ZIF-8 + US, MH, MH + US,
MHS and MHS + US) to investigate whether ZIF-8 and MH had an effect on ROS
production in the presence of US, respectively. The confocal laser scanning
microscopy (CLMS) images show that the MH + US group and MHS + US group
produce a large amount of ROS compared to other groups, demonstrating that the
presence of HMME is one of the necessary components for ROS production (**Fig. 3a**
**and Supplementary Fig. 3a**). The related data have been added in the Revised
Manuscript and Revised Supplementary Information. (**Line179-184, Page 6, Revised**
**Manuscript and Supplementary Figure 3a, Page 10, Revised Supplementary**
**Information**)

**Fig. 3 Evaluation of US-associated *IDO1* genome editing *in vitro*.** (a) CLSM images of 4T1
cells with different treatments (including Control, US only, ZIF-8, ZIF-8 + US, MH, MH + US,
MHS and MHS + US). Concentration = 100 $\mu\text{g}/\text{mL}$. Incubation time = 12 h. ($n = 3$)

**Supplementary Figure 3.** (a) Fluorescence intensity of CLSM images of 4T1 cells with different
 treatments (including Control, US only, ZIF-8, ZIF-8 + US, MH, MH + US, MHS and MHS + US)
 ($n = 3$). * $P < 0.05$, ** $P < 0.01$, *** $P < 0.001$, **** $P < 0.0001$.

8. In Figure S7, the author claims that Cy5.5-labeled Cas9/sgRNA system entered the
 nucleus, however, the Cy5 signal seems to follow the pattern of the lysotracker. In
 addition, the nucleus does not look intact. Z-stack is needed to show localization in
 the nucleus.

**Response:** Thank you very much for your constructive suggestions. According to the
 reviewer's suggestion, confocal laser scanning microscopy (Z-stack model) have been
 conducted, As shown in Supplementary Fig. 3b, under US irradiation, the Cy5.5-
 labeled red fluorescence signal was separated from the green fluorescence signal of
 lysosomes, while Cy5.5-labeled red fluorescence was detected in the nucleus,
 indicating that US irradiation is required for Cas9/sgRNA endosomal/lysosomal
 escape. The related data have been updated in the Revised Information (**Line 187-190,**
 **Page 6, Revised Information**).

**supplementary Figure 3.** (b) Z-stack CLSM images of 4T1 cells cultured with Cy5.5-labeled
 MHS nanosystem upon US irradiation for 1 and 3 h at 37 °C. The cell nuclei were stained with
 DAPI (blue), endo/lysosomes were stained with LysoTracker Green (green), and MHS was
 labeled with Cy5.5 (red). ($n = 3$)

*9. In the cytotoxicity experiment (Figure 3d), if the role of gene silencing is to improve*
 *the immune system mediated killing of the cells, why do we see improved efficacy*
 *when no immune cells are present in the model? Why is the toxicity MHS+US*
 *significantly higher than the MH+US system. Similar observation was seen with*
 *Fig.3e &S8 between MH+US and MHS+US group. Why the presence of Cas9/sgRNA*
 *increased the apoptosis in 4T1 cells?*

**Response:** Thank you very much for your kind question. IDO inhibition results not
 only in enhanced immune aspects, but also in other aspects that inhibit tumor cell
 proliferation and promote apoptosis. The current studies on the effect of *IDO1* gene
 silencing are mainly focused on the immune aspect⁶⁵⁻⁶⁸. IDO acts as an endogenous
 immunosuppressive mediator, stimulating the accumulation of FOXP3⁺ Tregs and
 suppressing T cell activity by depleting Trp in the microenvironment^{29, 30}. However,
 the presence of IDO as a rate-limiting step enzyme of the kynurenine pathway (KP)
 can have a fundamental impact on cell function and survival⁶⁹. Tryptophan is the

rarest essential amino acid in food and is used not only for tissue protein synthesis but
also as a precursor for a range of biologically active metabolites. Although a small
fraction of free Trp is used for protein synthesis and the production of
neurotransmitters such as 5-hydroxytryptamine and neuromodulators such as
tryptamine, more than 95% of free Trp is a substrate of the KP pathway, which
produces several metabolites with unique biological activity in immune responses and
neurotransmission^{70, 71}. Representative of these is NAD(P)H and KP is a major source
of ab initio NAD synthesis, with studies showed that abnormalities in the KP pathway
lead to rapid depletion of NAD by PARP, which results in apoptosis of lung cancer
cells mediated by NAD(P)H quinone dehydrogenase 1 (NQO1)⁷²⁻⁷⁴. It was shown that
IDO metabolizes TRP to generate kyn and kyn, which are further metabolized to
3HK (3-hydroxy-kynurenine) and HAA (3-hydroxyanthranilic acid), two downstream
metabolites with a strong ability to scavenge ROS⁷⁵, which would affect the efficiency
of SDT and thus reduce the killing effect on 4T1 cells *in vitro*. In addition, another
downstream metabolite of TRP, indole-3-pyruvate, was reported to have strong anti-
iron death activity not long ago⁷⁶. Alternatively, it has been shown that tumors display
enhanced IDO expression and that downstream metabolites (*e.g.*, Kyn) can activate β -
linked protein signaling, leading to increased proliferation of colon cancer in mice.⁷⁷

Therefore, IDO inhibition results not only in enhanced immune aspects, but also
in other aspects that inhibit tumor cell proliferation and promote apoptosis, so that in
vitro also results in superior therapeutic effects compared to other groups.

*10. In Figure 3h, the 12% difference in cleavage between the two groups is not*
*reflected in agarose gel. Also, NGS and the Deep sequencing data for MHS only were*
*not provided.*

**Response:** Thank you very much for your kind comments. Agarose gel
electrophoresis was used to re-probe the gene editing efficiency of Cas9/sgRNA on

4T1 cells for 3 times. Grayscale analysis for the target bands showed that the MHS +
 US group produced more cleavage products relative to the MHS group (Fig. 3i,
 Supplementary Fig. 3h). In addition, NGS and deep sequencing for other groups
 have been provided in Revised Manuscript and Revised Supplementary Information.
 (Fig.3j, k, Page32-33, Revised Manuscript; Supplementary Fig. 3-4, Page10-11,
 Revised Supplementary Information) The results indicate that US-generated ROS
 disruption of the lysosomal membrane could significantly improve genome editing
 efficiency.

**Fig. 3 Evaluation of US-associated *IDO1* genome editing *in vitro*.** (a) CLSM images of 4T1
cells with different treatments (including Control, US only, ZIF-8, ZIF-8 + US, MH, MH + US,
MHS and MHS + US). Concentration = 100 $\mu\text{g}/\text{mL}$. Incubation time = 12 h. ($n = 3$) (b) Illustration
of transfection process of 4T1 cells by MHS upon US. (c) Toxicity evaluation in 4T1 after
incubated with different concentrations of MHS, cell viability was analyzed by 24 h after the
treatment. ($n = 5$) (d) Cell viability of 4T1 cells after various treatments for 24 h. ($n = 5$) $*P < 0.05$,
$**P < 0.01$, $***P < 0.001$, $****P < 0.0001$. (e) Flow cytometry analysis of apoptosis of 4T1 cells
with various treatments, including control, US only, MH, MH + US, MHS, and MHS + US. (f)
CLSM images and (g) corresponding mean fluorescence intensity of 4T1 cells treated with various
treatments after IFN γ -stimulation, including control, US only, MH, MH + US, MHS and MHS +
US, followed by staining with fluorescent anti-IDO antibody (red). DAPI was used to stain the
nucleus of the cell (blue) ($n = 3$). (h) *In vitro* DNA sequencing of *IDO1* in 4T1 cells after
treatment with MHS and MHS + US. (i) T7EI cleavage analysis after 4T1 cells with different
treatments, including control, US only, MH, MH + US, MHS and MHS + US ($n = 3$). (j-k) Deep
sequencing analysis of gene editing in 4T1 cells in the presence of MHS and MHS + US.

**Supplementary Figure 3.** (h) Corresponding quantitative analysis of T7E I cleavage after 4T1
cells with different treatments, including control, US only, MH, MH + US, MHS, and MHS + US.

**Supplementary Figure 4.** (a) Deep sequencing for targeted disruption of *IDO1* locus in control,
 US only, MH, MH + US, MHS and MHS + US. (b) Nucleotide deletion and insert distribution
 around the cut site of *IDO1* locus in control, US only, MH, MH + US, MHS and MHS + US.

*11.* In figure 3f, in the MHS+US group, the reduced signal might be due to the cells
 being out of focus compared to the others. We suggest using the nucleus as a point of

*focus to make it easier to visualize and compare.*

**Response:** Thank you very much for your kind comments. The image of the MHS +
US group in Fig. 3f has been replaced. The replacement image is from a previous
repeat experiment of the MHS + US group taken under the same experimental
conditions, with its focus on the nucleus, making its experimental results convincing
in comparison with those of the other groups. (Fig 3f, g, Page 32-33, Revised
Manuscript)

**Fig. 3 Evaluation of US-associated IDO1 genome editing *in vitro*.** (f) CLSM images and (g)
corresponding mean fluorescence intensity of 4T1 cells treated with various treatments after IFN γ -
stimulation, including control, US only, MH, MH + US, MHS and MHS + US, followed by
staining with fluorescent anti-IDO antibody (red). DAPI was used to stain the nucleus of the cell
(blue) ($n = 3$).

*12. In fig. S9, the expression of IDO1 seems to be lower in the case of MHS compared*
*to MHS+US which contradict the gene deletion rates mentioned in line 205 and 206.*

**Response:** Thank you very much for your kind comments. To investigate the gene
editing efficacy of the MHS nanosystem under US irradiation, Cas9/sgRNA-mediated
IDO1 degradation was examined in 4T1 cells by employing immunofluorescence
staining and Western blotting. Four replicates of WB were performed for IDO protein
expression. We then performed a quantification analysis of the results. The average
IDO/ β -Actin value in the MHS group was 0.46, whereas the average IDO/ β -Actin
value in the MHS+US group was significantly lower compared to the MHS group,

with an average value of 0.30. These results indicate that Cas9/sgRNA effectively
mediated the *IDO1* knockdown. The related data have been added in the Revised
Supplementary Information (Supplementary Figure 3e, f, Page 10, Revised
Supplementary Information).

**Supplementary Figure 3.** (e) Western Blot and (f) corresponding quantitative analysis of IFN γ -
stimulated 4T1 cells treated with various treatments, including control, US only, MH, MH + US,
MHS, and MHS + US ($n = 4$). * $P < 0.05$, ** $P < 0.01$, *** $P < 0.001$, **** $P < 0.0001$.

13. In the experiment “In vitro exploration of ultrasonic-immunometabolic therapy”
line 236-237, the correlation or the mechanism by which MHS + US triggered the
ICD is not clear since some groups showed similar trends in the case of protein
expression Ex. MHS group had similar protein expression for CRT and HSP70 to
MHS +US group (Figure 4a).

**Response:** Thank you very much for your kind comments. There is growing evidence
that ultrasound-activated sonosensitizers can cause apoptosis/necrosis of tumor cells,
which then elicit some degree of immune response by generating tumor-associated

neoantigens⁷⁸⁻⁸⁰. It has also been shown that when cells are subjected to
microenvironmental stimuli or dysregulation of the antioxidant system to generate an
excess of ROS, the production of intracellular ROS can disrupt the integrity of
macromolecular biology, cause cellular damage, generate oxidative stress, have
damaging effects on intracellular mitochondrial DNA and induce apoptosis^{81, 82}.
Therefore, our strategy is to use the irradiation of MHS nanosystem US to generate
ROS, which induces ICD,*i.e.*, triggers ER stress response, and dying tumor cells
release tumor antigens and present them to DCs, while releasing DAMPs from
intracellular cells to promote maturation of immature DCs and enhance the ability of
DCs to recognize the presented antigens. When ICD occurs, dying tumor cells release
immune signaling molecules, collectively known as DAMPs, which include CRT
exposed on the cell surface and high mobility group protein 1 (HMGB1) released
outside the cell nucleus.

In addition, we also performed protein extraction and WB replicate experiments
on cells after different treatments (control, US only, MH, MHS, MH + US, MHS +
US). The protein bands as well as the grey scale analysis showed that the protein
expression of the groups without US irradiation was significantly different from that
of the groups with US irradiation. Co-incubation of 4T1 cells with MH + US or
MHS+ US caused a decrease of HMGB1 band intensity and an increase of CRT and
HSP70 band intensity. (Fig. 4a and Supplementary Fig. 5a-c). In addition, CLSM was
also used to detect the expression of protein amounts after different treatments. As a
result of fluorescence quantification, it showed the similar tendency as WB. It
indicates that HMME induced by US caused the production of ROS inside the cells,
which triggered ICD in tumor cells. The related data and discussion have been added
in the Revised Manuscript and Revised Supplementary Information (**Fig. 4a-d, Page**
**34-35, Revised Manuscript; Supplementary Figure. 5, Page 12, Revised**
**Supplementary Information**).

**Fig. 4 ICD facilitates antitumor immunity against 4T1 cells *in vitro*.** (a) Western blot analysis
 of specific proteins expression after DAMPs (HMGB1, CRT and HSP70). 4T1 cells were left
 untreated, treated with US only, co-incubated with MH, MHS, MH + US and MHS + US.
 Concentration = 100 μ g/mL. Incubation time = 12 h ($n = 4$). (b-d) Immunofluorescence analysis of
 specific proteins expression after DAMPs, including HMGB1 (red), CRT (red) and HSP70 (green).
 4T1 cells were left untreated, treated with US only, co-incubated with MH, MHS, MH + US and
 MHS + US. DAPI was used to stain the nucleus of the cell (blue) ($n = 3$)

**Supplementary Figure 5.** (a-c) The quantitative analysis of HMGB1, CRT and HSP70 on
 Western Blot. ($n = 4$). * $P < 0.05$, ** $P < 0.01$, *** $P < 0.001$, **** $P < 0.0001$. (d-f) Fluorescence
 intensity of HMGB1, CRT and HSP70 on CLSM ($n = 3$). * $P < 0.05$, ** $P < 0.01$, *** $P < 0.001$,
 **** $P < 0.0001$.

*14. In figure 5, was RNAseq-based KEGG analysis of differential gene expression*
*profiles conducted for LGG-MHS+US treatment only? Again there are many controls*
*missing.*

**Response:** Thank you very much for your kind comments. We apologize for the errors
in the description and layout of the paper that caused some confusion to the reviewers.
The sequencing in Figure 5 explores the mechanism by which LGG alone promotes
tumor therapy, and our statement in the label in Figure 5g and line 308-314 of the
original manuscript is correct. We apologize for the misspelling of "LGG" as "LGG-
MHS+US" in the figure caption to Figure 5. we have made corrections in the Revised
Manuscript. (Fig.5, Page 36, Revised Manuscript).

**Fig. 5 Bacterial hypoxia targeting characterization and bacterial sequencing.** (a) Volcano map
 and (b) Heatmap of genes alteration with or without LGG treatment ($P < 0.05$, $|\text{fold change}| \geq 2$).
 (c) RNAseq-based KEGG analysis of differential gene expression profiles after LGG treatment. (d)
 *In vivo* imaging and (g) corresponding fluorescence intensity of Cy5.5-labeled MHS, Cy5.5-
 labeled LGG and Cy5.5-labeled LGG-MHS in mice, respectively. (5×10^6 CFU per mouse, $n = 3$).

* $P < 0.05$, ** $P < 0.01$, *** $P < 0.001$, **** $P < 0.0001$. (e) Accumulation and (f) corresponding
mean fluorescence intensity of Cy5.5-labeled MHS, Cy5.5-labeled LGG and Cy5.5-labeled LGG-
MHS in major organs (1. Heart, 2. Liver, 3. Spleen, 4. Lung, 5. Kidney, 6. Tumor. $n = 3$). (h)
Photographs of bacterial colonization in various organs harvested from 4T1-bearing mice at
various time points after injection of MHS, LGG, and LGG-MHS on solid MRS agar plates ($n =$
3).

*15. The biosafety of the LGG-MHS nanosystem on different organs was evaluated*
*without applying the US which is the main activator of the system. It would be more*
*reflective to show that after applying US.*

**Response:** Thank you for your constructive suggestions. Based on your suggestion,
we explored the safety of different doses of LGG-MHS under US irradiation (control,
10 mL/kg, 20 mL/kg, 30 mL/kg, 40 mL/kg. 1mL LGG-MHS including 1×10^7 LGG
and 200 μ g MHS). Mice were injected with different doses of LGG-MHS 7 days after
tumor cell injection and US was applied to the tumor site the day after LGG-MHS
injection. Statistical analysis of the data samples for safety evaluation was performed
using Dunnett's multiple comparisons post test. It was found that mice injected with
2-fold the therapeutic dose showed no abnormalities in haematological parameters
and organ HE sections compared to untreated mice, demonstrating the excellent
biosafety of the LGG-MHS nanosystem under US irradiation (**Supplementary Fig. 8**).
The related data and discussion have been added in the Revised Supplementary
Information (**Supplementary Figure 8, Page 15, Revised Supplementary**
**Information**).

**Supplementary Figure 8.** (a) HE staining of histological sections of healthy mice treated with
 different doses of LGG-MHS (PBS, 10 mL/kg, 20 mL/kg, 30 mL/kg, 40 mL/kg. 1 mL LGG-MHS = 1
 $\times 10^7$ LGG, 1 mg MHS) and subjected to US irradiation of each organ. ($n = 3$) (b) In vivo
 hematological indices. Hematological assays of healthy mice treated with different doses of LGG-
 MHS (PBS, 10 mL/kg, 20 mL/kg, 30 mL/kg, 40 mL/kg. 1 mL LGG-MHS = 1 $\times 10^7$ LGG, 1 mg
 MHS). (c) *In vivo* liver and kidney function index. Hematological assays of mice healthy mice
 treated with different doses of LGG-MHS (PBS, 10 mL/kg, 20 mL/kg, 30 mL/kg, 40 mL/kg. 1 mL
 LGG-MHS = 1 $\times 10^7$ LGG, 1 mg MHS). ($n = 3$) * $P < 0.05$, ** $P < 0.01$, *** $P < 0.001$, **** $P <$
 0.0001 .

*16. For all in vivo experiments with LGG+MHS+US, a main control is missing. The*
*role of gene knockdown of Cas9/gRNA will not be conveyed clearly if LGG-MH+US*
*is not tested.*

**Response:** Thank you for your constructive suggestions. According to the reviewer's
suggestion, the corresponding experimental controls such as LGG-MH + US (without
CRISPR/Cas9 system) group in animal models have been added to explore the
contribution of IDO decrease to tumor growth inhibition. As a result, compared to the
control group although LGG-MH exhibited some inhibition of tumor growth under
irradiation with US, it failed to achieve the elimination of the primary tumor.
Attributed to IDO immunotherapeutic target inhibition, the LGG-MHS+US group
exhibited a superior ability to inhibit tumor growth with a tumor elimination rate of
4/5 (**Figure 6b, c**). Despite the relatively strong inhibitory effect of LGG-MH + US
on primary tumor growth, the results of its survival analysis (**Supplementary Figure**
**9b**), inhibition of distal tumors, and against lung metastases (**Figure 8**) were not
satisfying. The relevant details have been provided in the Revised Manuscript as
suggested by the reviewers. (**Line 476-480, Page 15, Revised Manuscript and**
**Supplementary Figure 9, Page 16, Revised Supplementary Information**)

**Fig. 6** LGG-MHS + US against 4T1 tumor *in vivo*. (a) Schematic diagram of primary tumor
 treatment process *in vivo*. (b) Tumor growth curves of 4T1 after being treated by PBS, LGG, MHS,
 LGG-MHS, MHS + US, LGG-MH + US, LGG-MHI + US and LGG-MHS + US ($n = 5$). (c)
 Average tumor growth curves in different groups ($n = 5$). (d) HPLC assay of the Trp content in
 primary tumors and TDLNs of tumor-bearing mice after different treatments ($n = 3$). (e) Elisa
 assay Kyn content in primary tumors and TDLNs of tumor-bearing mice after different treatments
 ($n = 3$). (f) Antigen Ki-67 staining in tumor sections from each experiment group ($n = 3$). (g)

Images and (h) corresponding fluorescence intensity of IDO immunofluorescence staining in
 primary tumors of 4T1 tumor-bearing mice after various treatments. DAPI was used to stain the
 nucleus of the cell (blue), and the *IDO1* was stained with anti-*IDO* antibodies (red) ($n = 3$). (i)
 Average tumor growth curves after being treated by re-challenge. ($n_{LGG-MHI + US} = 2$, $n_{LGG-MHS + US}$
 $= 4$)

**Fig. 8 Anti distal tumor effect and immunological memory of LGG-MHS + US in the 4T1**
 **bearing mice model.** (a) Schematic diagram of the establishment of distal tumors model and the
 experimental procedure of treatment. (b) Average tumor growth curves of primary tumor in
 different groups ($n = 5$). * $P < 0.05$, ** $P < 0.01$, *** $P < 0.001$, **** $P < 0.0001$. (c) Mean growth
 curves and (d) corresponding growth curves of distant tumors in different groups ($n = 5$). * $P <$
 0.05 , ** $P < 0.01$, *** $P < 0.001$, **** $P < 0.0001$. (e) Immunofluorescence images of helper T
 lymphocytes (CD3⁺CD4⁺) and proliferated cytotoxic T lymphocytes (CD3⁺CD8⁺) in 4T1 tumor
 tissue slices of distal tumor ($n = 3$). (f) Schematic diagram of the establishment and treatment
 process of mouse models of lung metastasis. (g) Typical flow cytometric of the effector memory T

cells (CD3⁺CD8⁺CD44⁺CD62L⁻) (Tem) and (CD3⁺CD8⁺CD44⁺CD62L⁺) (Tcm) in the spleen after
24 h after the first different treatments ($n = 3$). (h) Bioluminescence images and (i) corresponding
fluorescence intensity quantification of lung metastatic nodules of the 4T1 tumors ($n = 3$). (j) HE
staining of lung tissue from different groups of 4T1 tumor-bearing mice. The nodules with yellow
circles in the section diagram indicate metastases in the lungs.

*17. There are many grammatical mistakes that need to be corrected. Ex. Line 75 “is”*
*not needed, line 77 “barrier”, line 78 “it maintains”, line 166 it improves gene*
*delivery, line 333 repetition of “that”, figure 5e. “kidney”.*

**Response:** Thank you very much for pointing this issue out. We have carefully
checked and corrected the spelling and grammatical errors throughout the whole
manuscript.

Finally, we would like to thank you very much for your comments and
suggestions of our idea and work, which are very important for us to improve and
revise our manuscript.

Response to reviewer #4

*In this manuscript, the authors reported the synthesis of ZIF-8 for tumor targeted*
*delivery of sonosensitizer HMME and CRISPR/Cas9 system by employing the*
*intrinsic tumor hypoxia targeting ability of LGG. By downregulating the expression of*
*IDO1, the obtained composites were shown to be able to effectively suppress tumor*
*growth via the combined sonodynamic treatment and tumor immunosuppression*
*reversion. However, similar topics have been widely reported in the past several years*
*and this study did not provide enough attractive new results.*

**Response:** We appreciate very much for your constructive comments and kind
recommendation. The specific originality and novelty of this work are herein clarified
as follows:

(1) **First paradigm of microbial biomimetic CRISPR/Cas9 nanosystem.**
Although CRISPR/Cas9-mediated gene editing has shown promising results in
clinical studies. However, how to achieve efficient delivery and controlled release of
protein/nucleic acid complexes in the *in vivo* environment, thereby reducing off-target
rates and enabling effective and precise cancer therapy, is an important scientific
question to be addressed by the CRISPR/Cas9 delivery system. In the present study,
anaerobic bacteria were combined with CRISPR/Cas9 nanosystem to form a self-
driven CRISPR/Cas9 nanosystem. The hypoxia-targeting property of LGG provides
them with the ability to carry CRISPR/Cas9 nanosystem to actively target and
colonize the tumor. The designed self-driven CRISPR/Cas9 nanosystem provides a
novel microbial vector for CRISPR/Cas9 delivery, which dramatically decreases the
off-target rate of gene editing and significantly improves the possibility of further
clinical application of gene editing technology *in vivo*. Importantly, LGG has
promising applications in tumor therapy not only as a carrier for nanomedicine
delivery, but also for regulating tumor microenvironment to activate the immune
system.

**(2) Pioneering utilization of ultrasound for dual modulation of gene editing**
**system and immune system.** For the first time, we have established a platform that
allows gene knockdown under US irradiation while reprogramming the tumor
immunosuppressive microenvironment. The CRISPR/Cas9 gene editing system can
generate ROS by US triggered. ROS effectively disrupts the structure of the
lysosomal membrane and promotes the CRISPR/Cas9 nanosystem release, enabling
gene knockdown. Meanwhile, abundant ROS generated by US can induce ICD.
Molecular damage-related proteins released by ICD are absorbed by immature DC as
antigens, promoting their maturation, thereby upregulating killer T cells and
enhancing immunotherapy.

**(3) Comprehensive activation of the immune system by multiple pathways.**
The self-driven system efficiently delivers the CRISPR/Cas9 system to knock down
*IDO1* to reduce immunosuppressive cells (Tregs), while LGG activates multiple
signaling pathways to enhance intrinsic immunity. In addition, the system can
increase the efficiency of gene editing and cause ICD under US irradiation. This
“cocktail therapy” can effectively activate immune cells to eliminate the primary
tumor and inhibit tumor metastasis and recurrence.

*Specific comments:*

*1. Attributing to the intrinsic targeting ability of LGG, it is believed that HMME and*
*CRISPR/Cas9 system loaded within the ZIF-8 nanoparticles would be primarily*
*delivered to the hypoxic tumor region. Therefore, I want to know if the hypoxic*
*condition would diminish the sonosensitization efficacy of HMME under US exposure.*

**Response:** Thanks very much for your kind question. Oxygen insufficiency, known as
hypoxia, is a unique and intrinsic feature of most malignancies caused by aggressive

cell proliferation and dysfunctional angiogenesis. Hypoxia plays a crucial role in
hostile tumor microenvironment (TME) and greatly influences the therapeutic
outcome of treatments in which oxygen is a key factor in killing tumors. Given the
critical role of hypoxia in tumor progression and its resistance to treatment, many
efforts have been made to overcome the limitations associated with hypoxia regarding
tumors. These approaches can be roughly classified into three categories: ⁸³ (a) The
use of oxygen supplementation strategies to alleviate tumor hypoxia by improving
intratumoral blood flow, utilizing hostile TME at the molecular level, generating
oxygen in situ, delivering exogenous oxygen to the tumor, or reducing oxygen
consumption during treatment⁸⁴⁻⁸⁷, (b) The development of some innovative oxygen
reduction dependent therapeutic modalities or combining one or more of these
approaches with some other non-oxygen dependent cancer therapies⁸⁸⁻⁹⁰, and (c)
exploiting inherent tumor hypoxia and post-treatment amplified hypoxia, which is
then combined with some hypoxia-activated bioreduction therapies, hypoxia-sensitive
molecules in nanoscale carriers, or cancer starvation therapies⁹¹⁻⁹³. Hypoxic
conditions certainly reduce the efficacy of acoustic sensitization of HMME under US
irradiation.

Our strategy, however, is to use LGG as a hypoxia-responsive component,
leading to tumor accumulation of LGG and thus to massive enrichment of MHS in
tumors, compensating at the quantitative level for the lack of efficiency of ROS
production due to tumor hypoxic microenvironment. In addition, since our drug
administration and US application are performed on alternate days, the exacerbation
of hypoxia due to ROS production will inevitably lead to LGG enrichment, ultimately
achieving high specificity as well as synergistic anti-cancer efficiency of the LGG-
MHS nanosystem. Therefore, the LGG-MHS nanosystem could be considered as a
comprehensive self-feedback therapeutic process, resulting in integrated anticancer
efficacy as well as higher therapeutic efficacy.

2. Actually, diverse small molecule *IDO1* inhibitors have been developed to reverse
tumor immunosuppression by restricting the production of Kyn. Therefore, I would
like to suggest the authors to describe the advantages of the presented strategies.

**Response:** Thank you very much for the kind comments. Indoleamine-2,3-
dioxygenase-1 (*IDO1*) is an endogenous immunosuppressive mediator that stimulates
the accumulation of FOXP3⁺ Tregs and suppresses T-cell activity by depleting Trp in
the microenvironment. Thus, *IDO1* is a potential immunotherapeutic target to
reprogram TIME by improving amino acid metabolism. Nevertheless, small molecule
inhibitors generally do not provide durable responses due to the presence of drug
resistance²⁹⁻³³. A number of compounds have been reported in the relevant patent
literature, but no inhibitors have been marketed. The promising efficacy in animal
models has also greatly contributed to the advancement of clinical trials of *IDO*
inhibitors, but the clinical performance of *IDO* inhibitors has fallen short of
expectations⁹⁴. Therefore, there is an urgent need for alternative approaches to
interfere with amino acid metabolism to reprogram the TIME of cancer
immunotherapy.

The evolution of gene editing technologies for (CRISPR)/CRISPR-associated
protein 9 (Cas9) is seen as an innovative approach to solve a variety of intractable
biomedical problems, ushering in a promising new era in biology and medicine.⁹⁵⁻⁹⁸
CRISPR/Cas9 gene editing systems show great potential in biomedical fields,
including disease model construction, disease therapy, and gene function research⁹⁹⁻¹⁰².

CRISPR/Cas9, as an emerging genome editing technology, has the advantages of
simple design, high specificity and high efficiency, bringing a breakthrough in the
regulation and application of targeted genome modification and showing broad
application prospects in biomedicine³⁴. In our strategy, after the entry of MHS into
tumor cells, Cas9/sgRNA escapes from the lysosome under irradiation of US and is
translocated to the nucleus for efficient *IDO1* knockdown, inhibiting the expression of
*IDO* protein from the source, eliminating the defects such as drug resistance that

exists in small molecule inhibitors, thereby reducing the aggregation of Treg cells in
the tumor microenvironment.

According to the reviewer's comment, we have added the following brief
description of the current status of IDO small molecule inhibitors in the Revised
Manuscript to justify this approach, which reads: "Thus, *IDO1* is a potential
immunotherapeutic target to reprogram the TIME by improving amino acid
metabolism. Nevertheless, small molecule inhibitors generally cannot provide durable
responses due to the presence of drug resistance, and a phase III clinical trial of IDO
inhibitor combination therapy was declared a failure." (Line 64-68, Page 2, Revised
**Manuscript**)

*3. Based on the results shown in Figure 2, the pore size of the obtained MH and MHS*
*nanoparticles with typical ZIF-8 morphology is very small. Therefore, I want to know*
*how CRISPR/Cas9 systems were loaded. Besides, would the loading process*
*negatively impair the biological activity of loaded CRISPR/Cas9 system? Did the US*
*irradiation promoted generation of ROS negatively the biological activity of*
*CRISPR/Cas9 systems.*

**Response:** Thank you very much for the kind questions and comments. Metal-organic
frameworks (MOFs), consisting of metal or cluster nodes linked by organic ligands,
have emerged as a promising platform for biomedical applications due to their highly
porous structure, friendliness to various functionalization methods, and excellent
biocompatibility and biodegradability^{60, 61}.

Related studies have shown that due to the open porous structure, available metal
or organic active sites, and good thermal and chemical stability of MOFs, various
drugs/large/small molecules are mainly three methods of binding to MOFs: grafting,
permeation and encapsulation²⁰. It has been reported that biomacromolecules such as
enzymes may be encapsulated within MOFs *via* two general strategies: by assembling

the MOF around the enzyme (which term de novo encapsulation) or by introducing
the enzyme into the pre-existing MOF (which term post-synthetic encapsulation).
(Enzyme encapsulation in metal – organic frameworks for applications in catalysis).
Zinc 2-methylimidazole (ZIF-8), a nanoscale metal – organic framework with
excellent biocompatibility, has unique features in biomacromolecules condensing and
chemical drug-loading efficiency due to its positive charge and high surface ratio.
More importantly, the acidic environment of endosomes/lysosomes can trigger the
degradation of ZIF-8 hosts, which can facilitate cargo escape from
endosomes/lysosomes to the cytosol^{61, 62}.

Our strategy firstly employs one-step encapsulation approach to encapsulate
HMME into the interior of ZIF-8. The HMME was dropwise into the
dimethylimidazole solution stirred for 10 min before the addition of zinc nitrate
hexahydrate. The material after encapsulating HMME with ZIF-8 (MOF) is named
MH. Second, MH was incubated with Cas9/sgRNA to form MHS. In summary,
HMME is encapsulated into the interior of ZIF-8 during the synthesis process. In
contrast, Cas9/sgRNA is partially internalized into the interior of MH and partially
grafted onto the surface of MH after incubation with MH, resulting in MHS. Revised
Manuscript Figure 2b and Supplementary Figure 2c show that the average pore size of
MHS decreased relative to ZIF-8, demonstrating that some Cas9/sgRNA penetrated
into the interior of MH. Revised Manuscript Figure 2e shows that the particle size of
MHS slightly increases compared to MH, which proves that some Cas9/sgRNA is
also grafted on the surface of MH. Finally, Revised Manuscript Figure 2d
demonstrates that the elemental mapping of MHS corresponds to a more dense P-
element compared to ZIF-8 and MH, which further suggests that MH was
successfully loaded with Cas9/sgRNA. Therefore, the final MH and Cas9/sgRNA
formed the MHS complex.

The detailed MHS experimental procedure been provided in the revised
manuscript according to the reviewer’s kind question, which reads “2-
Methylimidazole (1.910 g) and zinc nitrate solution (1.314 g) were dissolved in

methanol (20 mL), respectively. Hematoporphyrin monomethyl ether (HMME, 200
μL , 2 mg/mL) was slowly added to 2-methylimidazole solution under mechanical
stirring at room temperature, and after 10 min, zinc nitrate solution was added
dropwise. The MH was obtained after stirring for 24 h at room temperature. Then, the
MH and CRISPR/Cas9 system (mass ratio 4:1) were incubated at 37 ° C according to
the methodology instructions, finally, the integration of MHS nanosystem was
constructed. The obtained product was gathered by centrifugation and washed with
ddH₂O for three times to remove the residuum.” (Line 568-575, Page 19, Revised
Manuscript).

According to the reviewer’s suggestion, the more detailed distributions of the
effect of loading process on the activity of CRISPR/Cas9 nanosystem have been
further recorded and the data have been supplemented in the revised manuscript.
(Line 583-586, Page 19, Revised Manuscript).

To investigate the effect of the loading process on the activity of the
CRISPR/Cas9 nanosystem. Different states of Cas9/sgRNA (including Cas9/sgRNA
Only, MHS, MHS + US, LGG-MHS, LGG-MHS + US) were incubated in acidic PBS
(pH = 5) for 6 h, and then incubated by quantitative extraction of equal amounts of
Cas9/sgRNA with target DNA fragments, and finally agarose gel electrophoresis was
performed. The results are shown in Supplementary Figure 2k, l. Quantitative analysis
of the cut bands indicates that with the loading process or the application of US, the
activity of Cas9/sgRNA is maintained at a high level, although a slight decrease
occurs.

**Supplementary Figure 2.** (k) Agarose gel electrophoresis and (l) corresponding quantitative
analysis of the activity of CRISPR/Cas9 nanosystem under different states, including I (DNA
Only), II (Cas9/sgRNA + DNA), III (MHS + DNA), IV (MHS + US + DNA), V (LGG-MHS +
DNA), VI (LGG-MHS + US + DNA). ($n = 3$) * $P < 0.05$, ** $P < 0.01$, *** $P < 0.001$, **** $P <$
0.0001 .

*4. The authors are suggested to describe the methods used for the loading of MHS*
*nanoparticle onto the surfaced of LGG. Besides, Did the MHS nanoparticles loading*
*impact the colonization behaviors of LGG.*

**Response:** Thank you for your constructive comments. According to the reviewer's
suggestion, the more detailed methods used for the loading of MHS nanoparticle onto
the surface of LGG have been supplemented in the revised manuscript, which reads
"The obtained product was gathered by centrifugation and washed with ddH₂O for
three times to remove the residuum. MHS was further stirred with LGG (PBS = 1mL,
LGG = 1×10^7 CFU, MHS = 1 mg) in PBS for 24 h to arrangement LGG-MHS.
**(Line 574-576, Page 19, Revised Manuscript).**

In addition, according to the reviewer's suggestion, we investigated the activity
of LGG loaded with different concentrations of MHS. The related experimental
procedures and data have been supplemented in the revised manuscript, which reads:
" 1×10^7 CFU LGG in PBS without stirring was set as the control group, 1×10^7 LGG
in PBS with different concentrations of MHS (0 mg/mL, 0.5 mg/mL, 1 mg/mL, 2
1675 mg/mL, 4 mg/mL) and given mechanical stirring was set as the experimental group.
After various times (0, 6, 12, 24 h) the groups were coated on MRS agar plates (100
μ L taken after 100-fold dilution)" As shown in Supplement Materials Figure 2m-n,
the effect on LGG activity was not statistically significant when the concentration of
MHS was 2 mg/mL, whereas the CFU decreased substantially when the concentration
of MHS reached 4 mg/mL. The results indicate that the concentration of LGG loaded
MHS (1 mg/mL) in our strategy does not negatively affect the activity of LGG. **(Line**
**587-592, Page 19, Revised Manuscript)**

 **Supplementary Figure 2.** (m) Representative photographs and (n) corresponding CFU
 quantitative of MRS agar plates of bacterial activity with various concentrations of MHS in a
 different time (0, 2, 6, 12 and 24 h) ($n = 3$).

5. In Figure 3e, it was shown that the flow cytometric plot of MHS and US treated
 cells was distinct from the typical apoptotic cancer cells. Please double check. Maybe
 the combination treatment could not induce apoptosis since it has been well
 documented that apoptosis of cancer cells is not the immunogenic cell death because
 it could not promote the expression of CRT, release of HMGB1.

**Response:** Thank you for your constructive comments. We have reanalyzed the flow
 cytometric data from the original Figure 3e, and the related data have been updated in
 the Revised Manuscript Fig. 3e¹⁰³⁻¹⁰⁵.

**Fig. 3** (e) Flow cytometry analysis of apoptosis of 4T1 cells with various treatments, including
control, US only, MH, MH + US, MHS, and MHS + US.

Apoptosis of normal cancer cells is certainly not all about immunogenic cell
death, so those cells that die non-immunogenically do not promote the release of CRT
and HMGB1. There is growing evidence that ultrasound-activated sonosensitizers can
cause apoptosis/necrosis of tumor cells, which then elicit some degree of immune
response by generating tumor-associated neoantigens.⁷⁸⁻⁸⁰ It has also been shown that
when cells are subjected to microenvironmental stimuli or dysregulation of the
antioxidant system to generate an excess of ROS, the production of intracellular ROS
can disrupt the integrity of macromolecular biology, cause cellular damage, generate
oxidative stress, have damaging effects on intracellular mitochondrial DNA and
induce apoptosis^{81, 82}.

Therefore, our strategy is to use the MHS nanosystem upon US irradiation to
generate ROS, which induces ICD, *i.e.*, triggers ER stress response, and dying tumor
cells release tumor antigens and present them to DCs, while releasing DAMPs from
intracellular cells to promote maturation of immature DCs and enhance the ability of
DCs to recognize the presented antigens. The mature DCs enter the lymph nodes,
present tumor antigens to T lymphocytes and activate T cells, which become effector
T cells (*e.g.* CD4⁺ T cells, CD8⁺ T cells). The dying tumor cells release tumor
antigens and present them to the DCs, while releasing DAMPs from the cells, which
promote the maturation of immature DCs and enhance the ability of DCs to recognize
the presented antigens. When ICD occurs, dying tumor cells release immune signaling
molecules, collectively known as DAMPs, which include CRT exposed on the cell
surface and high mobility group protein 1 (HMGB1) released outside the cell nucleus.

*6. The authors are suggested to explain why the treatment of MHS plus US was more*
*efficient than the treatment of MH plus US in promoting the immunogenic cell death*
*of 4T1 cancer cells. Besides, the authors are suggested to explain the mechanism of*

such combination treatment in promoting the expression of HSP70.

**Response:** Thank you for your constructive comments. According to the
reviewer's suggestion, the quantification of the WB bands and statistical analysis of
the CLSM fluorescence intensity quantification have been performed. It was found
that there was no significant difference in the ability of MHS + US and MH + US to
trigger the ICD. The related experimental details have been provided in the revised
manuscript according to the reviewer's kind suggestions. (Fig. 4a-d, Page 34,
Revised Manuscript and Supplementary Figure 5, Page 12, Revised
Supplementary Information)

**Fig. 4 ICD facilitates antitumor immunity against 4T1 cells *in vitro*.** (a) Western blot analysis of
specific proteins expression after DAMPs (HMGB1, CRT and HSP70). 4T1 cells were left
untreated, treated with US only, co-incubated with MH, MHS, MH + US and MHS + US.
Concentration = 100 μ g/mL. Incubation time = 12 h ($n = 4$). (b-d) Immunofluorescence analysis of
specific proteins expression after DAMPs, including HMGB1 (red), CRT (red) and HSP70 (green).
4T1 cells were left untreated, treated with US only, co-incubated with MH, MHS, MH + US and
MHS + US. DAPI was used to stain the nucleus of the cell (blue) ($n = 3$).

**Supplementary Figure 5.** (a-c) The quantitative analysis of HMGB1, CRT and HSP70 on
 Western Blot. (n = 4). * $P < 0.05$, ** $P < 0.01$, *** $P < 0.001$, **** $P < 0.0001$. (d-f) Fluorescence
 intensity of HMGB1, CRT and HSP70 on CLSM (n = 3). * $P < 0.05$, ** $P < 0.01$, *** $P < 0.001$,
 **** $P < 0.0001$.

Related researches have shown that Heat shock proteins (HSPs) are a conserved
 family of chaperone proteins that function under physiological and environmental
 stress. HSP70 is involved in the regulation of essential cellular processes such as
 signal transduction, cell cycle regulation, apoptosis and innate immunity¹⁰⁶⁻¹⁰⁸. One
 mechanism of cellular protection from the adverse consequences of ROS action is
 provided by highly conserved heat shock proteins (HSPs), which are ubiquitously
 expressed intracellular stress proteins^{109, 110}. These molecular chaperones are involved
 in proper protein folding and utilization, preventing protein aggregation and providing
 cellular resistance to stress. It was shown that the JAK/STAT pathway mediates H₂O₂-
 induced HSP70 expression, which contributes to cellular adaptation to oxidative
 stress^{111, 112}. In our strategy, HMME in the MHS were irradiated with US to produce
 an abundance of ROS (Figure 3a, Page 33 Revised Manuscript), which in turn
 caused tumors to develop ICDs, and the secreted DAMPs included the
 aforementioned HSP70.

**Fig. 3 Evaluation of US-associated *IDO1* genome editing *in vitro*.** (a) CLSM images of 4T1
 cells with different treatments (including Control, US only, ZIF-8, ZIF-8 + US, MH, MH + US,
 MHS and MHS + US). Concentration = 100 $\mu\text{g/mL}$. Incubation time = 12 h. ($n = 3$)

7. In figure 4h, the flow cytometric patten of these maturated BMDCs is quite different
 from those published ones. Please double check.

**Response:** Thank you very much for the kind reminding, which is highly appreciated.
 After careful examination of the flow cytometry for these mature BMDCs and
 reanalysis of the data based on the reviewers' suggestions, we have provided the
 relevant results in the Figure 4g of Revised Manuscript, as described below^{103, 113, 114}:

**Fig. 4 ICD facilitates antitumor immunity against 4T1 cells *in vitro*.** (g) Representative flow
 cytometry plots and statistical data of maturated BMDCs ($\text{CD80}^+\text{CD86}^+\text{CD11c}^+$) after various
 treatments, including control, US only, MH, MH + US, MHS and MHS + US. ($n = 3$).

8. In Figure 7c and S25, the gating strategy used for analyzing the percentages of
CD4⁺Foxp3⁺ Tregs was not correct. Please reanalyze the results. Besides, it seems
that the gate strategies shown in Figure S25 were not the standard ones.

**Response:** Thank you very much for the kind reminding, which is highly appreciated.
After carefully checking the gating strategy used for analyzing the percentages of
CD4⁺Foxp3⁺ Tregs, we have found that the methods and results are inappropriate.
According to the reviewer's questions, appropriate gate strategies was performed to
investigate the population of Tregs (CD3⁺CD4⁺Foxp3⁺)^{111, 115}. The rest of the flow
cytometry in the original manuscript Figure 7c and S25 was reanalyzed as well^{114, 116-}
¹²⁰. The related result and gating strategy is as follows:

**Fig. 7. Reprogramming of the tumor immunosuppressive microenvironment by the self-driven**
**LGG-MHS+US nanosystem.** (a) Typical flow cytometric of mature DCs in tumor tissue after 24
1792 h after the first different treatments ($n = 3$). (b) Typical flow cytometric of T cells of CD4⁺ and
1793 CD8⁺ T cells in the spleen after 24 h after the first different treatments ($n = 3$). (c) Typical flow
cytometric of Tregs in primary tumor tissue after 24 h after the first different treatments ($n = 3$). (d)
Representative flow cytometric of M2 macrophages in spleen after 24 h after the first different

treatments ($n = 3$).

**Supplementary Figure 10.** (e) Gating strategies for isolating CD80⁺CD86⁺ mature DCs from
tumor tissue. (f) Gating strategies for isolating CD4⁺ and CD8⁺ T cells from spleen tissue. (g)
Gating strategies for isolating Tregs from tumor tissue. (h) Gating strategies for isolating M2
macrophages from spleen tissue.

9. The font size of Figure 6b was too small. Please reformat the figure.

**Response:** Thank you very much for pointing this issue out. We have carefully
reformatted the size figure to make them look comfortable.

**Fig. 6 LGG-MHS + US against 4T1 tumor *in vivo*.** (b) Tumor growth curves of 4T1 after being
treated by PBS, LGG, MHS, LGG-MHS, MHS + US, LGG-MH + US, LGG-MHI + US and
LGG-MHS + US ($n = 5$).

Finally, we greatly appreciate and thank the reviewers' kind, professional and
constructive reminding, comments and suggestions for this manuscript. We have tried
our best to address all these issues as possible as we can. We sincerely hope that the
revised manuscript has addressed all the comments and suggestions as kindly raised
by the reviewers and meet the publication standard of *Nature Communications*.
Thank you very much.

**References**

- 1. Joyce JA, *et al.* Microenvironmental regulation of metastasis. *Nat Rev Cancer*
**9**, 239-252 (2009).
- 2. Correia AL, *et al.* The tumor microenvironment is a dominant force in
multidrug resistance. *Drug Resistance Updates : Reviews and Commentaries*
*In Antimicrobial and Anticancer Chemotherapy* **15**, 39-49 (2012).
- 3. Bissell MJ, *et al.* The organizing principle: Microenvironmental influences in
the normal and malignant breast. *Differentiation; Research In Biological*
*Diversity* **70**, 537-546 (2002).
- 4. Kulkarni P, *et al.* Hypoxia responsive, tumor penetrating lipid nanoparticles
for delivery of chemotherapeutics to pancreatic cancer cell spheroids.
*Bioconjug Chem* **27**, 1830-1838 (2016).
- 5. Kulkarni P, *et al.* Hypoxia-responsive polymersomes for drug delivery to
hypoxic pancreatic cancer cells. *Biomacromolecules* **17**, 2507-2513 (2016).
- 6. Forbes NS. Engineering the perfect (bacterial) cancer therapy. *Nat Rev Cancer*
**10**, 785-794 (2010).
- 7. Minton NP. Clostridia in cancer therapy. *Nat Rev Microbiol* **1**, 237-242 (2003).

- 8. Van Mellaert L, *et al.* Clostridium spores as anti-tumour agents. *Trends*
*Microbiol* **14**, 190-196 (2006).
- 9. Hoffman RM. Tumor-seeking salmonella amino acid auxotrophs. *Curr Opin*
*Biotechnol* **22**, 917-923 (2011).
- 10. Taniguchi S, *et al.* Targeting solid tumors with non-pathogenic obligate
anaerobic bacteria. *Cancer Sci* **101**, 1925-1932 (2010).
- 11. Triantafilou M, *et al.* Lipopolysaccharide recognition: Cd14, tlrs and the lps-
activation cluster. *Trends Immunol* **23**, 301-304 (2002).
- 12. Dasari S, *et al.* Surfacing role of probiotics in cancer prophylaxis and therapy:
A systematic review. *Clin Nutr* **36**, 1465-1472 (2017).
- 13. Nishikawa H, *et al.* In vivo antigen delivery by a salmonella typhimurium type
iii secretion system for therapeutic cancer vaccines. *J Clin Invest* **116**, 1946-
1954 (2006).
- 14. Kim JE, *et al.* Salmonella typhimurium suppresses tumor growth via the pro-
inflammatory cytokine interleukin-1beta. *Theranostics* **5**, 1328-1342 (2015).
- 15. Hu Q, *et al.* Engineering nanoparticle-coated bacteria as oral DNA vaccines
for cancer immunotherapy. *Nano Lett* **15**, 2732-2739 (2015).
- 16. Chandra D, *et al.* Myeloid-derived suppressor cells have a central role in
attenuated listeria monocytogenes-based immunotherapy against metastatic
breast cancer in young and old mice. *Br J Cancer* **108**, 2281-2290 (2013).
- 17. Wang Y, *et al.* An lgg-derived protein promotes iga production through
upregulation of april expression in intestinal epithelial cells. *Mucosal Immunol*
**10**, 373-384 (2017).
- 18. Seow SW, *et al.* Lactobacillus rhamnosus gg induces tumor regression in mice
bearing orthotopic bladder tumors. *Cancer Sci* **101**, 751-758 (2010).
- 19. McKinlay AC, *et al.* Biomofs: Metal-organic frameworks for biological and
medical applications. *Angew Chem Int Ed Engl* **49**, 6260-6266 (2010).
- 20. Doonan C, *et al.* Metal-organic frameworks at the biointerface: Synthetic
strategies and applications. *Acc Chem Res* **50**, 1423-1432 (2017).

- 21. Alsaiari SK, *et al.* Endosomal escape and delivery of crispr/cas9 genome
editing machinery enabled by nanoscale zeolitic imidazolate framework. *J Am*
*Chem Soc* **140**, 143-146 (2018).
- 22. Guo X, *et al.* Multifunctional nanoplatfoms for subcellular delivery of drugs
in cancer therapy. *Progress in Materials Science* **107**, (2020).
- 23. Guo L, *et al.* Radicals scavenging mofs enabling targeting delivery of sirna for
rheumatoid arthritis therapy. *Small* **18**, e2202604 (2022).
- 24. Chen J, *et al.* Metal-phenolic coatings as a platform to trigger endosomal
escape of nanoparticles. *ACS Nano* **13**, 11653-11664 (2019).
- 25. Rabiee N, *et al.* Recent advances in porphyrin-based nanocomposites for
effective targeted imaging and therapy. *Biomaterials* **232**, 119707 (2020).
- 26. Deepagan VG, *et al.* Long-circulating au-tio2 nanocomposite as a
sonosensitizer for ros-mediated eradication of cancer. *Nano Lett* **16**, 6257-
6264 (2016).
- 27. Yu T, *et al.* Anticancer potency of cytotoxic drugs after exposure to high-
intensity focused ultrasound in the presence of microbubbles and
hematoporphyrin. *Mol Pharm* **8**, 1408-1415 (2011).
- 28. Ethirajan M, *et al.* The role of porphyrin chemistry in tumor imaging and
photodynamic therapy. *Chem Soc Rev* **40**, 340-362 (2011).
- 29. Muller AJ, *et al.* Inhibition of indoleamine 2,3-dioxygenase, an
immunoregulatory target of the cancer suppression gene bin1, potentiates
cancer chemotherapy. *Nat Med* **11**, 312-319 (2005).
- 30. Ladomersky E, *et al.* Idol inhibition synergizes with radiation and pd-1
blockade to durably increase survival against advanced glioblastoma. *Clin*
*Cancer Res* **24**, 2559-2573 (2018).
- 31. Lemos H, *et al.* Immune control by amino acid catabolism during
tumorigenesis and therapy. *Nat Rev Cancer* **19**, 162-175 (2019).
- 32. Konieczkowski DJ, *et al.* A convergence-based framework for cancer drug
resistance. *Cancer Cell* **33**, 801-815 (2018).

- 33. Gottesman MM, *et al.* Toward a better understanding of the complexity of
cancer drug resistance. *Annu Rev Pharmacol Toxicol* **56**, 85-102 (2016).
- 34. Konermann S, *et al.* Genome-scale transcriptional activation by an engineered
crispr-cas9 complex. *Nature* **517**, 583-588 (2015).
- 35. Zou W. Immunosuppressive networks in the tumour environment and their
therapeutic relevance. *Nat Rev Cancer* **5**, 263-274 (2005).
- 36. Joyce JA, *et al.* T cell exclusion, immune privilege, and the tumor
microenvironment. *Science* **348**, 74-80 (2015).
- 37. Phuengkham H, *et al.* Nanoengineered immune niches for reprogramming the
immunosuppressive tumor microenvironment and enhancing cancer
immunotherapy. *Adv Mater* **31**, e1803322 (2019).
- 38. Xie S, *et al.* Doxorubicin-conjugated escherichia coli nissle 1917 swimmers to
achieve tumor targeting and responsive drug release. *J Control Release* **268**,
390-399 (2017).
- 39. Du JZ, *et al.* Tailor-made dual ph-sensitive polymer-doxorubicin nanoparticles
for efficient anticancer drug delivery. *J Am Chem Soc* **133**, 17560-17563
(2011).
- 40. Du JZ, *et al.* Tumor extracellular acidity-activated nanoparticles as drug
delivery systems for enhanced cancer therapy. *Biotechnol Adv* **32**, 789-803
(2014).
- 41. Li Y, *et al.* Delivery of nanomedicines to extracellular and intracellular
compartments of a solid tumor. *Adv Drug Deliv Rev* **64**, 29-39 (2012).
- 42. Doherty GJ, *et al.* Mechanisms of endocytosis. *Annu Rev Biochem* **78**, 857-
902 (2009).
- 43. Zhang X, *et al.* Persistence and recovery of zif-8 and zif-67 phytotoxicity.
*Environ Sci Technol* **55**, 15301-15312 (2021).
- 44. Wu W, *et al.* Microbiotic nanomedicine for tumor-specific chemotherapy-
synergized innate/adaptive antitumor immunity. *Nano Today* **42**, (2022).
- 45. Tannock IF, *et al.* Acid ph in tumors and its potential for therapeutic
exploitation. *Cancer Res* **49**, 4373-4384 (1989).

- 46. Stubbs M, *et al.* Causes and consequences of tumour acidity and implications
for treatment. *Mol Med Today* **6**, 15-19 (2000).
- 47. Chen B, *et al.* Current multistage drug delivery systems based on the tumor
microenvironment. *Theranostics* **7**, 538-558 (2017).
- 48. Tekade RK, *et al.* The warburg effect and glucose-derived cancer theranostics.
*Drug Discov Today* **22**, 1637-1653 (2017).
- 49. Borkowska M, *et al.* Targeted crystallization of mixed-charge nanoparticles in
lysosomes induces selective death of cancer cells. *Nat Nanotechnol* **15**, 331-
341 (2020).
- 50. Zhang J, *et al.* Immunostimulant hydrogel for the inhibition of malignant
glioma relapse post-resection. *Nat Nanotechnol* **16**, 538-548 (2021).
- 51. Hu X, *et al.* A novel modulation of structural and functional changes of mouse
bone marrow derived dendritic cells (bmdcs) by interleukin-2(il-2). *Hum*
*Vaccin Immunother* **11**, 516-521 (2015).
- 52. Robb RJ, *et al.* Heterogeneity of human t-cell growth factor(s) due to variable
glycosylation. *Mol Immunol* **18**, 1087-1094 (1981).
- 53. Smith KA, *et al.* Production and characterization of monoclonal antibodies to
human interleukin 2: Strategy and tactics. *J Immunol* **131**, 1808-1815 (1983).
- 54. Tahara H, *et al.* Antitumor effects of interleukin-12 (il-12): Applications for
the immunotherapy and gene therapy of cancer. *Gene Ther* **2**, 96-106 (1995).
- 55. Thierfelder WE, *et al.* Requirement for stat4 in interleukin-12-mediated
responses of natural killer and t cells. *Nature* **382**, 171-174 (1996).
- 56. Trinchieri G, *et al.* Natural killer cell stimulatory factor (nksf) or interleukin-
12 is a key regulator of immune response and inflammation. *Prog Growth*
*Factor Res* **4**, 355-368 (1992).
- 57. Zeh HJ, 3rd, *et al.* Interleukin-12 promotes the proliferation and cytolytic
maturation of immune effectors: Implications for the immunotherapy of
cancer. *J Immunother Emphasis Tumor Immunol* **14**, 155-161 (1993).

- 58. Chowdhury FZ, *et al.* Il-12 selectively programs effector pathways that are
stably expressed in human cd8⁺ effector memory t cells in vivo. *Blood* **118**,
3890-3900 (2011).
- 59. Rotow J, *et al.* Understanding and targeting resistance mechanisms in nscl.
*Nat Rev Cancer* **17**, 637-658 (2017).
- 60. Horcajada P, *et al.* Metal-organic frameworks in biomedicine. *Chem Rev* **112**,
1232-1268 (2012).
- 61. He C, *et al.* Nanomedicine applications of hybrid nanomaterials built from
metal-ligand coordination bonds: Nanoscale metal-organic frameworks and
nanoscale coordination polymers. *Chem Rev* **115**, 11079-11108 (2015).
- 62. Zheng H, *et al.* One-pot synthesis of metal-organic frameworks with
encapsulated target molecules and their applications for controlled drug
delivery. *J Am Chem Soc* **138**, 962-968 (2016).
- 63. Venna SR, *et al.* Structural evolution of zeolitic imidazolate framework-8. *J*
*Am Chem Soc* **132**, 18030-18033 (2010).
- 64. Saliba D, *et al.* Crystal growth of zif-8, zif-67, and their mixed-metal
derivatives. *J Am Chem Soc* **140**, 1812-1823 (2018).
- 65. Zhai L, *et al.* Ido1 in cancer: A gemini of immune checkpoints. *Cell Mol*
*Immunol* **15**, 447-457 (2018).
- 66. Inhibiting btk and ido enhances dendritic cell-mediated immune response.
*Cancer Discov* **11**, 2956 (2021).
- 67. Munn DH, *et al.* Ido in the tumor microenvironment: Inflammation, counter-
regulation, and tolerance. *Trends Immunol* **37**, 193-207 (2016).
- 68. Guo Y, *et al.* Indoleamine 2,3-dioxygenase (ido) inhibitors and their
nanomedicines for cancer immunotherapy. *Biomaterials* **276**, 121018 (2021).
- 69. Cervenka I, *et al.* Kynurenines: Tryptophan's metabolites in exercise,
inflammation, and mental health. *Science* **357**, (2017).
- 70. Stone TW, *et al.* An expanding range of targets for kynurenine metabolites of
tryptophan. *Trends Pharmacol Sci* **34**, 136-143 (2013).

- 71. van der Goot AT, *et al.* Tryptophan metabolism: Entering the field of aging
and age-related pathologies. *Trends Mol Med* **19**, 336-344 (2013).
- 72. Newman AC, *et al.* Immune-regulated ido1-dependent tryptophan metabolism
is source of one-carbon units for pancreatic cancer and stellate cells. *Mol Cell*
**81**, 2290-2302 e2297 (2021).
- 73. Bender DA, *et al.* Utilization of tryptophan, nicotinamide and nicotinic acid as
precursors for nicotinamide nucleotide synthesis in isolated rat liver cells. *Br J*
*Nutr* **59**, 279-287 (1988).
- 74. Liu H, *et al.* De-novo nad⁺ synthesis regulates sirt1-foxo1 apoptotic pathway
in response to nqo1 substrates in lung cancer cells. *Oncotarget* **7**, 62503-
62519 (2016).
- 75. Fiore A, *et al.* Kynurenine importation by slc7a11 propagates anti-ferroptotic
signaling. *Mol Cell* **82**, 920-932 e927 (2022).
- 76. Zeitler L, *et al.* Anti-ferroptotic mechanism of il4i1-mediated amino acid
metabolism. *Elife* **10**, (2021).
- 77. Thaker AI, *et al.* Idol metabolites activate beta-catenin signaling to promote
cancer cell proliferation and colon tumorigenesis in mice. *Gastroenterology*
**145**, 416-425 e411-414 (2013).
- 78. Zhang Q, *et al.* Sonodynamic therapy-assisted immunotherapy: A novel
modality for cancer treatment. *Cancer Sci* **109**, 1330-1345 (2018).
- 79. Galon J, *et al.* Approaches to treat immune hot, altered and cold tumours with
combination immunotherapies. *Nat Rev Drug Discov* **18**, 197-218 (2019).
- 80. Nam J, *et al.* Cancer nanomedicine for combination cancer immunotherapy.
*Nature Reviews Materials* **4**, 398-414 (2019).
- 81. Memar MY, *et al.* Antimicrobial use of reactive oxygen therapy: Current
insights. *Infect Drug Resist* **11**, 567-576 (2018).
- 82. Qian X, *et al.* Micro/nanoparticle-augmented sonodynamic therapy (sdt):
Breaking the depth shallow of photoactivation. *Adv Mater* **28**, 8097-8129
(2016).

- 83. Hu D, *et al.* Application of nanotechnology for enhancing photodynamic
therapy via ameliorating, neglecting, or exploiting tumor hypoxia. *View* **1**,
(2020).
- 84. Hu D, *et al.* Activatable albumin-photosensitizer nanoassemblies for triple-
modal imaging and thermal-modulated photodynamic therapy of cancer.
*Biomaterials* **93**, 10-19 (2016).
- 85. Xue Y, *et al.* Insight into cao2-based fenton and fenton-like systems: Strategy
for cao2-based oxidation of organic contaminants. *Chem Eng J* **361**, 919-928
(2019).
- 86. Jiang W, *et al.* Tumor reoxygenation and blood perfusion enhanced
photodynamic therapy using ultrathin graphdiyne oxide nanosheets. *Nano Lett*
**19**, 4060-4067 (2019).
- 87. Shi C, *et al.* Catalase-based liposomal for reversing immunosuppressive tumor
microenvironment and enhanced cancer chemo-photodynamic therapy.
*Biomaterials* **233**, 119755 (2020).
- 88. Zhou W, *et al.* Iodine-rich semiconducting polymer nanoparticles for
ct/fluorescence dual-modal imaging-guided enhanced photodynamic therapy.
*Small* **16**, e1905641 (2020).
- 89. Nguyen VN, *et al.* An emerging molecular design approach to heavy-atom-
free photosensitizers for enhanced photodynamic therapy under hypoxia. *J Am*
*Chem Soc* **141**, 16243-16248 (2019).
- 90. Jung HS, *et al.* Overcoming the limits of hypoxia in photodynamic therapy: A
carbonic anhydrase ix-targeted approach. *J Am Chem Soc* **139**, 7595-7602
(2017).
- 91. Liu Y, *et al.* Hypoxia induced by upconversion-based photodynamic therapy:
Towards highly effective synergistic bioreductive therapy in tumors. *Angew*
*Chem Int Ed Engl* **54**, 8105-8109 (2015).
- 92. Zhang K, *et al.* Light-triggered theranostic liposomes for tumor diagnosis and
combined photodynamic and hypoxia-activated prodrug therapy. *Biomaterials*
**185**, 301-309 (2018).

- 93. Luan X, *et al.* A tumor vascular-targeted interlocking trimodal nanosystem
that induces and exploits hypoxia. *Adv Sci (Weinh)* **5**, 1800034 (2018).
- 94. Long GV, *et al.* Epcadostat plus pembrolizumab versus placebo plus
pembrolizumab in patients with unresectable or metastatic melanoma (echo-
301/keynote-252): A phase 3, randomised, double-blind study. *The Lancet*
*Oncology* **20**, 1083-1097 (2019).
- 95. Cheng Q, *et al.* Selective organ targeting (sort) nanoparticles for tissue-
specific mrna delivery and crispr-cas gene editing. *Nat Nanotechnol* **15**, 313-
320 (2020).
- 96. Jinek M, *et al.* A programmable dual-rna-guided DNA endonuclease in
adaptive bacterial immunity. *Science* **337**, 816-821 (2012).
- 97. Wiedenheft B, *et al.* Rna-guided genetic silencing systems in bacteria and
archaea. *Nature* **482**, 331-338 (2012).
- 98. Lyu Y, *et al.* A photolabile semiconducting polymer nanotransducer for near-
infrared regulation of crispr/cas9 gene editing. *Angew Chem Int Ed Engl* **58**,
18197-18201 (2019).
- 99. Wolter JM, *et al.* Cas9 gene therapy for angelman syndrome traps ube3a-ats
long non-coding rna. *Nature* **587**, 281-284 (2020).
- 100. Yan L, *et al.* Coupling of n7-methyltransferase and 3'-5' exoribonuclease with
sars-cov-2 polymerase reveals mechanisms for capping and proofreading. *Cell*
**184**, 3474-3485 e3411 (2021).
- 101. Jin X, *et al.* In vivo perturb-seq reveals neuronal and glial abnormalities
associated with autism risk genes. *Science* **370**, (2020).
- 102. Frangoul H, *et al.* Crispr-cas9 gene editing for sickle cell disease and beta-
thalassemia. *N Engl J Med* **384**, 252-260 (2021).
- 103. Qi J, *et al.* Synergistic effect of tumor chemo-immunotherapy induced by
leukocyte-hitchhiking thermal-sensitive micelles. *Nat Commun* **12**, 4755
(2021).

- 104. Qi J, *et al.* Ph and thermal dual-sensitive nanoparticle-mediated synergistic
antitumor effect of immunotherapy and microwave thermotherapy. *Nano Lett*
**19**, 4949-4959 (2019).
- 105. Yang W, *et al.* Smart nanovesicle-mediated immunogenic cell death through
tumor microenvironment modulation for effective photodynamic
immunotherapy. *ACS Nano* **14**, 620-631 (2020).
- 106. Li Y, *et al.* Hsp70 decreases receptor-dependent phosphorylation of smad2
and blocks tgf-beta-induced epithelial-mesenchymal transition. *J Genet*
*Genomics* **38**, 111-116 (2011).
- 107. Lahaye X, *et al.* Hsp70 protein positively regulates rabies virus infection. *J*
*Virol* **86**, 4743-4751 (2012).
- 108. Dokladny K, *et al.* Regulatory coordination between two major intracellular
homeostatic systems: Heat shock response and autophagy. *J Biol Chem* **288**,
14959-14972 (2013).
- 109. Kalmar B, *et al.* Induction of heat shock proteins for protection against
oxidative stress. *Advanced Drug Delivery Reviews* **61**, 310-318 (2009).
- 110. Niforou K, *et al.* Molecular chaperones and proteostasis regulation during
redox imbalance. *Redox Biol* **2**, 323-332 (2014).
- 111. Wang C, *et al.* Coordination polymer-coated caco3 reinforces radiotherapy by
reprogramming the immunosuppressive metabolic microenvironment. *Adv*
*Mater* **34**, e2106520 (2022).
- 112. Madamanchi NR, *et al.* Reactive oxygen species regulate heat-shock protein
70 via the jak/stat pathway. *Arterioscler Thromb Vasc Biol* **21**, 321-326 (2001).
- 113. Chen Y, *et al.* Spatiotemporal control of engineered bacteria to express
interferon-gamma by focused ultrasound for tumor immunotherapy. *Nat*
*Commun* **13**, 4468 (2022).
- 114. Liu J, *et al.* Co-delivery of iox1 and doxorubicin for antibody-independent
cancer chemo-immunotherapy. *Nat Commun* **12**, 2425 (2021).

- 115. Mao C, *et al.* Delivery of an ectonucleotidase inhibitor with ros-responsive
nanoparticles overcomes adenosine-mediated cancer immunosuppression. *Sci*
*Transl Med* **14**, eabh1261 (2022).
- 116. Jin F, *et al.* Nir-triggered sequentially responsive nanocarriers amplified
cascade synergistic effect of chemo-photodynamic therapy with inspired
antitumor immunity. *ACS Appl Mater Interfaces* **12**, 32372-32387 (2020).
- 117. Chen L, *et al.* Tumor-targeted drug and cpg delivery system for phototherapy
and docetaxel-enhanced immunotherapy with polarization toward m1-type
macrophages on triple negative breast cancers. *Adv Mater* **31**, e1904997
(2019).
- 118. Liu Y, *et al.* An inhalable nanoparticulate sting agonist synergizes with
radiotherapy to confer long-term control of lung metastases. *Nat Commun* **10**,
5108 (2019).
- 119. Zhang Y, *et al.* Reactive oxygen species-responsive and raman-traceable
hydrogel combining photodynamic and immune therapy for postsurgical
cancer treatment. *Nat Commun* **13**, 4553 (2022).
- 120. Huang Y, *et al.* Engineered macrophages as near-infrared light activated drug
vectors for chemo-photodynamic therapy of primary and bone metastatic
breast cancer. *Nat Commun* **12**, 4310 (2021).

REVIEWER COMMENTS

Reviewer #1 (Remarks to the Author):

The manuscript is thoroughly revised by including supplementary data, additional justifications, and sentences with more clarity. There remains, however, some questions that needs to be addressed prior to publication. With appropriate revisions in the manuscripts on these aspects, we feel that the manuscript may be ready for publication.

The author claims “superior tumor targeting” of the bacteria based on the increase in bacteria CFU in tumor over time. This needs to be carefully stated since we believe that the majority of this increase is due to bacterial replication and not specifically “targeting”. In this case, the conjugated MHS won’t accumulate in the tumor as much as expected. This needs to be clarified in the manuscript to avoid overstatement.

Figure 6 (i) only has n=2 for LGG-MHI group but n=4 for LGG-MHS group. Why is that? We suggest matching the sample size if possible. N=2 is difficult to assess the data. It could be worthwhile considering moving the data to supplementary.

Thank you to the authors for describing the MHS system. The mechanisms in which MHS gets detached from the conjugated bacteria and enters cancer cells are still unclear though. It would be helpful for the readers to clarify this point.

The author claims that the efficacy against metastasis were due to systemic immune activation. Have they checked whether the bacteria colonize metastasized tumors? This is a critical control experiment to support the claim.

The bacterial CFU difference in tumor and liver seems very small compared to other studies of bacterial cancer therapy (~1000 fold difference). We suggest the authors toning down the claim of tumor targeting by LGG, and clearly point out this difference in the manuscript.

Reviewer #2 (Remarks to the Author):

The authors have made extensive revisions and included new data that address all my previous concerns.

Reviewer #4 (Remarks to the Author):

I think this work with revisions is now acceptable for publication

Reviewer #5 (Remarks to the Author):

The manuscript reported by Yu et al. presents a self-driven CRISPR/Cas9 nanosystem for TIME modulation to avoid lung metastasis and antagonize re-challenge. The nanosystem uses LGG for hypoxia targeting and ZIF-8 for sonosensitizer hematoporphyrin monomethyl ether and CRISPR/Cas9 delivery. A lot of experiments have been done and largely support that the combination of these technology shows its powerful for in vivo tumor immunosuppressive. Although the combination of these technologies is novel and the authors emphasize their specific originality from microbial CRISPR/Cas9 nanosystem and ultrasound-based dual modulation, they do not provide a clear conclusion about how the nanosystem form and how its individual component interact. This results in the unexplained dominance of each component. To better understand the system, the following comments need to be addressed. The knockout of IDO may suppress tumors, but it may also affect other functions since IDO is a functional gene. Therefore, we suggest the author can try other

CRISPR systems, such as CRISPR/Cas13 for gene knockdown in RNA level in future studies.

1. The authors have a tedious explanation on how to encapsulate biomacromolecules by MOFs and give the conclusion as “Cas9/sgRNA is partially internalized into the interior of MH and partially grafted onto the surface of MH after incubation with MH, resulting in MHS.” But there is no positive response as to what kinds of interactions responsible for the internalization and graft.
2. It is not accurate to use average pore size to explain the Cas/sgRNA penetration. Firstly, the reviewer has serious doubts about the reliability of N₂ adsorption-desorption isotherms in Fig. 2b and supplementary Fig. 2c, since ZIF-8 does not have such large pore size. Secondly, the decrease of pore size is too less to demonstrate cargoes loaded into their pores, even for small molecules, not to mention Cas9/sgRNA.
3. In fact, the morphology of ZIF-8 is greatly influenced by the encapsulated biomolecules. But the hexagonal structure change is no guaranteed. The authors should give the PXRD results to show the crystal structure consistency.
4. Except P element, it seems like other elements also have increased in the EM figures of MHS. A relative percentage of each element is required for ZIF-8, MH and MHS. Please keep the scale bars of LGG-MHS in Fig. 2h consistently, not coexist of 0.5 and 1 μm .
5. It is still confused that the grayscale of naked sgRNA in supplementary Fig. 2b was used to calculate the loading efficiency since the RNP was added. Sametime, no electrophoresis result of naked sgRNA, even RNP has been shown in supplementary Fig. 2a.
6. The deep sequencing analysis is needed in Fig 3j to avoid that the point mutation is mismatch introduced during PCR process.
7. As an important reference, the authors need to add IDO to Fig. 4 and give the corresponding discussion.
8. Please add controls to RNAseq-based KEGG analysis in Fig. 5 to explain if the gene expression profiles are conducted by LGG-MHS+US treatment only.

Response to reviewer #1

The manuscript is thoroughly revised by including supplementary data, additional justifications, and sentences with more clarity. There remains, however, some questions that needs to be addressed prior to publication. With appropriate revisions in the manuscripts on these aspects, we feel that the manuscript may be ready for publication.

Response: Thank you very much for the positive comments and recommendations. Your concerns have been addressed point by point, and the corresponding content has been added and modified in the Revised Manuscript. Please find the following detailed responses to your comments and suggestions.

1.The author claims “superior tumor targeting” of the bacteria based on the increase in bacteria CFU in tumor over time. This needs to be carefully stated since we believe that the majority of this increase is due to bacterial replication and not specifically “targeting”. In this case, the conjugated MHS won’t accumulate in the tumor as much as expected. This needs to be clarified in the manuscript to avoid overstatement.

Response: Thank you for your constructive suggestions, which will help to improve the rigor of our research. The inappropriate description has been corrected in the Revised Manuscript, which reads, “The amount of LGG was increased dramatically over time in tumors within 24 h after injection. Interestingly, LGG enrichment in the tumor was higher than in the liver at 72 h with ~ 2-fold difference in CFU, which was attributed to the more favourable hypoxic microenvironment in the tumor for LGG proliferation, which further supports that LGG has relatively better hypoxic targeting and proliferative capacity (Supplementary Fig. 6a, b).” (Line 296-301, Page 9-10, Revised Manuscript)

2.Figure 6 (i) only has n=2 for LGG-MHI group but n=4 for LGG-MHS group. Why is

that? We suggest matching the sample size if possible. $N=2$ is difficult to assess the data. It could be worthwhile considering moving the data to supplementary.

Response: Thank you for your constructive comments. Regarding the mismatch between the two groups of mice in Figure 6i, which is due to construction of the re-challenge model according to the suggestion of 2# reviewers, *i.e.* re-injection of 4T1 cells into surviving mice ($n_{\text{LGG-MHI+US}} = 2$, $n_{\text{LGG-MHS+US}} = 4$) after primary tumor treatment to assess whether the treatment stimulates durable and stable anti-tumor immunity. Since the number of surviving mice is an experimental result after primary tumor treatment rather than by manual control, it leads to mismatch in the number of mice between the two groups. Following your suggestion, the relevant data has been moved to the Supplementary Information. (**Supplementary Figure 9i, Page 17, Revised Supplementary Information**).

Supplementary Figure 9. (i) Average tumor growth curves after being treated by re-challenge. ($n_{\text{LGG-MHI+US}} = 2$, $n_{\text{LGG-MHS+US}} = 4$)

3. Thank you to the authors for describing the MHS system. The mechanisms in which MHS gets detached from the conjugated bacteria and enters cancer cells are still unclear though. It would be helpful for the readers to clarify this point.

Response: Thank you for your constructive suggestions. The description of the mechanism of the separation of MHS from LGG was added to the article as follows, “Notably, it has been shown that the acidic nature of the tumor microenvironment reduces the forces between the drug molecule and the carrier material, such as electrostatic interaction, which facilitate the release of the drug. Therefore, when LGG-MHS is enriched in the tumor hypoxic microenvironment, the decrease in pH value improves the release of MHS from LGG.” (Line 113-116, Page 4, Revised Manuscript)

In addition, confocal was used to observe the mechanism of MHS entry into cells as detailed below, “4T1 cells were seeded into CLSM-specific culture dishes at a density of 1×10^5 and incubated for 24 h at 37 °C, followed by pre-treatments of M β CD, sucrose, and amiloride for 30 min, following the medium was replaced by Cy3-labeled MHS (MHS = 100 μ g/mL), which was then co- incubated for 3h. Then, the medium was washed with PBS for 3 times, followed by cell nucleus was stained by DAPI for 20 min. To further observe the intracellular fluorescence intensity, and the fluorescence signals were measured.”(Line 62-68, Page 3, Revised Supplementary Information) Corresponding descriptions were added to the Revised Manuscript, which reads, “In order to thoroughly investigate the cellular absorption mechanism and confirm clathrin-mediated endocytosis, caveolae-mediated endocytosis, and micro-pinocytosis, three endocytosis inhibitors—sucrose, methyl-cyclodextrin (M β CD), and amiloride—were applied, respectively. The CLSM images show that endocytosis efficiency was decreased in cells pretreated with M β CD and amiloride, indicating that caveolae-dependent endocytosis were the primary routes for the endocytic uptake of MHS (Supplementary Fig. 3a, b).” (Line 191-196, Page 6, Revised Manuscript)

Supplementary Figure 3. (a) CLSM images and (b) the corresponding mean fluorescence intensity analysis of cellular uptake of Cy3-labeled MHS by 4T1 cancer-cell line after coincubation with different inhibitors.

4. The author claims that the efficacy against metastasis were due to systemic immune activation. Have they checked whether the bacteria colonize metastasized tumors? This is a critical control experiment to support the claim.

Response: Thank you for your constructive suggestions. We have replicated the distal tumor model in order to investigate whether bacteria colonize distal tumors and thus can contribute to the immune activation effect, and have added a corresponding description to the Revised Manuscript, which reads, “It is crucial to ensure that LGG can colonize distal tumors before the LGG-MHS self-driven nanosystem elicits systemic immune effects. Therefore, 4T1 cells were injected into the left side of the second breast pad of mice and the same operation was performed on the right side 7 days later to establish an *in situ* dual tumor model as primary and distal tumors, respectively. When the primary tumor size reached approximately 200-250 mm³ and the distal tumor volume was approximately 60-80 mm³, LGG was injected *via* tail vein. After 24 hours, the primary and distal tumors were harvested and homogenized for dish coating. As shown in Supplementary Figure 12a and 12b, both primary and distal tumors showed LGG colonization. The difference in CFU may be due to the different

levels of hypoxia in the primary and distal TIME.” (Line 464-472, Page 15, Revised Manuscript)

Supplementary Figure 12a. (a) Representative photographs and corresponding CFU count analysis of MRS agar plates of bacterial colonization in primary and distal tumor. ($n = 3$),

5. The bacterial CFU difference in tumor and liver seems very small compared to other studies of bacterial cancer therapy (~1000 fold difference). We suggest the authors toning down the claim of tumor targeting by LGG, and clearly point out this difference in the manuscript.

Response: Thank you for your careful review and constructive suggestions, which are critical to improving the rigor of our research. To down the claim of tumor targeting by LGG, there are some improperly descriptions that were modified in the Revised Manuscript, which reads, “revealing that the LGG-MHS complex has relatively better tumor targeting properties.” (Line 343-344, Page 11, Revised Manuscript) “demonstrating relatively good tumor targeting properties of LGG-MHS” (Line 351-352, Page 11, Revised Manuscript) “After our study, we found that LGG does have an ability to target the hypoxic microenvironment of tumors.” (Line 533-534, Page 17, Revised Manuscript) “revealing relatively better tumor targeting properties of the LGG-MHS complex.” (Line 536-537, Page 17, Revised Manuscript)

In addition, the minor differences in bacterial CFU between tumor and liver were present in the Revise Manuscript, which reads, “The amount of LGG was increased

dramatically over time in tumors within 24 h after injection. Interestingly, LGG enrichment in the tumor was higher than in the liver at 72 h with ~ 2-fold difference in CFU, which was attributed to the more favourable hypoxic microenvironment in the tumor for LGG proliferation, which further supports that LGG has relatively better hypoxic targeting and proliferative capacity (Supplementary Fig. 6a, b).” (Line 296-301, Page 9-10, Revised Manuscript)

Response to reviewer #2

The authors have made extensive revisions and included new data that address all my previous concerns

Response: It is an honor to receive your approval of this work, and we appreciate your constructive comments to help improve the quality of the research.

Response to reviewer #4

I think this work with revisions is now acceptable for publication

Response: Thank you for your constructive comments that have greatly benefited the rigor of our study. We also appreciate your affirmation of this study.

Response to reviewer #5

The manuscript reported by Yu et al. presents a self-driven CRISPR/Cas9 nanosystem for TIME modulation to avoid lung metastasis and antagonize re-challenge. The nanosystem uses LGG for hypoxia targeting and ZIF-8 for sonosensitizer hematoporphyrin monomethyl ether and CRISPR/Cas9 delivery. A lot of experiments have been done and largely support that the combination of these technology shows its powerful for in vivo tumor immunosuppressive. Although the combination of these technologies is novel and the authors emphasize their specific originality from microbial CRISPR/Cas9 nanosystem and ultrasound-based dual modulation, they do

not provide a clear conclusion about how the nanosystem form and how its individual component interact. This results in the unexplained dominance of each component. To better understand the system, the following comments need to be addressed. The knockout of IDO may suppress tumors, but it may also affect other functions since IDO is a functional gene. Therefore, we suggest the author can try other CRISPR systems, such as CRISPR/Cas13 for gene knockdown in RNA level in future studies.

Response: Thank you for your kind comments and constructive suggestions. A conclusive description of how the LGG-MHS self-driven nanosystem was formed has been added to the Revised Manuscript as follows, “Briefly, during the synthesis of ZIF-8, HMME was added dropwise to form MH through *in situ* encapsulation, and MH was incubated with CRISPR/Cas9 to produce MHS *via* the inherent dispersion force of ZIF-8 coupled with surface energy between substances adsorption of CRISPR/Cas9, and grafting of CRISPR/Cas9 by imidazole-like ligands provided by ZIF-8. Finally, MHS was electrostatically adsorbed onto the surface of LGG after being magnetically agitated with it in PBS at room temperature.” (Line 105-110, Page 4, Revised Manuscript) The relevant experimental methods are represented in the Revised Manuscript, which reads, “2-Methylimidazole (1.910 g) and zinc nitrate solution (1.314 g) were dissolved in methanol (20 mL), respectively. Hematoporphyrin monomethyl ether (HMME, 200 μ L, 2 mg/mL) was slowly added to 2-methylimidazole solution under mechanical stirring at room temperature, and after 10 min, zinc nitrate solution was added dropwise. The MH was obtained after stirring for 24 h at room temperature. Then, the MH and CRISPR/Cas9 system (mass ratio 4:1) were incubated at 37 ° C according to the methodology instructions, finally, the integration of MHS nanosystem was constructed. The obtained product was gathered by centrifugation and washed with ddH₂O for three times to remove the residuum. MHS was further stirred with LGG (PBS = 1 mL, LGG = 1×10^7 CFU, MHS = 1 mg) in PBS for 24 h to arrangement LGG-MHS.” (Line 595-603, Page 19, Revised Manuscript) Additionally, a clear conclusion about how the components of the nanosystem interact was modified in the Revised Manuscript, which reads, “Utilizing the hypoxia targeting ability of LGG, the

ultrasound (US)-controlled CRISPR/Cas9 gene editing system (MHS) was delivered to the hypoxia tumor core, thus promoting effective accumulation of MHS in tumors. Notably, it has been shown that the acidic nature of the tumor microenvironment reduces the forces between the drug molecule and the carrier material, such as electrostatic interaction, which facilitate the release of the drug. Therefore, when LGG-MHS is enriched in the tumor hypoxic microenvironment, the decrease in pH value improves the release of MHS from LGG. The as-obtained CRISPR/Cas9 system generated reactive oxygen species (ROS) upon US triggering, which induced the release of tumor-associated antigens, immunogenic cell death of tumor cells and caused DCs maturation. In addition, ROS effectively disrupted the structure of the endosomal/lysosomal membrane, allowing Cas9/sgRNA to escape from the endosomal/lysosomal and transport to the nucleus for efficient *IDO1* knockdown, reducing Treg cells to cluster in the tumor microenvironment.” (Line 110-122, Page 4, Revised Manuscript)

We are grateful for your valuable suggestions regarding gene editing tools, and we will actively adopt your suggested research ideas in our future studies, which we believe will greatly benefit the quality and content of our future studies.

1. The authors have a tedious explanation on how to encapsulate biomacromolecules by MOFs and give the conclusion as “Cas9/sgRNA is partially internalized into the interior of MH and partially grafted onto the surface of MH after incubation with MH, resulting in MHS.” But there is no positive response as to what kinds of interactions responsible for the internalization and graft.

Response: Thank you for your kind comments and constructive suggestions. Many relevant studies have shown that MOFs automatically adsorb substances into the pores when immersed in sufficient concentration *via* the inherent dispersion force of MOFs and surface energy between substances¹⁻⁴. This process certainly requires the pore size and volume of the MOFs to be larger than the substance being adsorbed. Relevant BET data show that the average pore size of MH (ZIF-8 after in situ encapsulation of HMME)

is 3.3482 nm, which is sufficient for internalization of CRISPR/Cas9. In addition, the total pore volume of MHS was also reduced relative to MH, which demonstrated the successful internalization of CRISPR/Cas9.

Related studies have shown that hydrogen bonding interactions may occur between the free carboxyl, amino or imidazole MOF ligands of MOFs and biomolecules, which would lead to CRISPR/Cas9 coupling to the surface of ZIF-8⁵. A study in which simulated drug entry into ZIF-8 found that the Zn²⁺ cations in the ZIF-8 structure exhibit tetrahedral geometry coordinated by four neighboring imidazolate groups. It is expected that the Zn²⁺ cations on the surface of the ZIF-8 structure will have two imidazolate ligands replaced by water molecules. And the relevant molecular docking results show that doxorubicin binds to the Zn²⁺ cation, thus maintaining its tetrahedral coordination geometry, possibly by replacing two water molecules acting as ligands to the cation⁶. Thus ZIF-8 provides imidazole-based ligands capable of forming hydrogen bonds with biomolecules, making CRISPR/Cas9 grafting a reality.

And lastly, the elaboration on the interaction of internalization and grafting was mentioned in the Revised Manuscript, which reads, “Briefly, during the synthesis of ZIF-8, HMME was added dropwise to form MH through *in situ* encapsulation, and MH was incubated with CRISPR/Cas9 to produce MHS *via* the inherent dispersion force of ZIF-8 coupled with surface energy between substances adsorption of CRISPR/Cas9, and grafting of CRISPR/Cas9 by imidazole-like ligands provided by ZIF-8.” (Line 105-110, Page 4, Revised Manuscript)

2. It is not accurate to use average pore size to explain the Cas/sgRNA penetration. Firstly, the reviewer has serious doubts about the reliability of N₂ adsorption-desorption isotherms in Fig. 2b and supplementary Fig. 2c, since ZIF-8 does not have such large pore size. Secondly, the decrease of pore size is too less to demonstrate cargoes loaded into their pores, even for small molecules, not to mention Cas9/sgRNA.

Response: Thank you for your careful review and constructive suggestions. It was shown that not only micropores smaller than 2 nm and mesopores of 2-50 nm exist in ZIF-8, but also interparticle mesoporosity and macroporosity between ZIF-8 particles⁷. We apologize for inappropriately using the average pore size encompassing all pore sizes and interparticle mesoporosity and macroporosity between ZIF-8 particles that under 100 nm to account for Cas9/sgRNA penetration. Therefore, we removed the reference to aperture explaining CRISPR/Cas9 penetration. In addition, the NLDFT model was used to re-detect the pore size of MH and MHS, and the relevant data are presented in the inserted data in Figure 2b and Supplementary Figure 2c. A related research showed that when the MOF material was encapsulated in situ after the substance, the presence of mesopores with a radius of 3.5 ± 0.5 nm within the MOF was detected, significantly larger than the theoretical pore size of the pure phase ZIF-8, such that the mesopores have sufficient size to accommodate biomolecules⁸. Therefore, MH was performed BET test and the relevant data showed that after in situ encapsulation of HMME, mesopores with an average pore size of 3.3482 nm were detected in MH, and their size was sufficient for the penetration of Cas/sgRNA.

In addition, pore volume was used to explain Cas9/sgRNA penetration. Relevant data showed that the total pore volume of MHS calculated by single-point method was $0.045091\text{cm}^3/\text{g}$ when the relative pressure of adsorption curve was 0.988643472 after MH incubation with Cas9/sgRNA, although it did not show a significant decrease compared with the total pore volume of MH ($0.724167\text{cm}^3/\text{g}$). However, it is further verified that part of Cas9/sgRNA penetrates into MH. The corresponding description is added to the revision, which reads “Moreover, the pore volume of MHS also showed a decrease relative to MH, demonstrating that part of the Cas9/sgRNA successfully entered the interior of ZIF-8 *via* permeation.” (Line 142-144, Page 5, Revised Manuscript)

Fig. 2. (b) N_2 adsorption-desorption isotherms and of MHS. The inset shows its corresponding BET total pore volume specific surface area and average pore size.

Supplementary Figure 2. (c) N_2 adsorption-desorption isotherms and of MH. The inset shows its corresponding BET total pore volume specific surface area and average pore size.

3. In fact, the morphology of ZIF-8 is greatly influenced by the encapsulated biomolecules. But the hexagonal structure change is no guaranteed. The authors should give the PXRD results to show the crystal structure consistency.

Response: Thank you for your constructive comments and for this reason we have performed PXRD tests on ZIF-8, MH and MHS separately. The results are shown in Figure 2e in the Revised Manuscript. The peaks of ZIF-8, MH and MHS in PXRD plots do not show great differences. The corresponding descriptions were also added to the

Revised Manuscript, which reads, “Following that, transmission electron microscopy (TEM) and Powder X-ray diffraction (PXRD) were used to examine the morphologies and structures of ZIF-8, MH and MHS, which showed no changes in nanoparticles morphology except for the slightly increase in particle size of MH and MHS compared to ZIF-8.” (Line 144-148, Page 5, Revised Manuscript)

Fig. 2. (e) PXRD of ZIF-8, MH and MHS.

4. Except P element, it seems like other elements also have increased in the EM figures of MHS. A relative percentage of each element is required for ZIF-8, MH and MHS. Please keep the scale bars of LGG-MHS in Fig. 2h consistently, not coexist of 0.5 and 1 μm .

Response: Thank you for your careful review. The relative percentages of each element are provided in the following table (Supplementary Table 1, Page 23, Revised Supplementary Information). As the data presented in the supplementary table, although the proportion of all elements except C shows an increase, the increase of P element is much greater than that of Zn element and N element.

Additionally, the scale bars in Figure 2h of the previous revision manuscript were adjusted to be consistent (Figure 2i, Page 30, Revised Manuscript).

Supplementary Table 1: Corresponding atomic fraction of Fig. 2d.

	Zn (%)	P (%)	N (%)	C (%)
ZIF-8	1.09	0.01	3.89	95.01
MH	0.95	0.01	2.67	96.37
MHS	3.88	0.56	4.95	90.61

Fig. 2. (i) Transmission electron microscopic (TEM) and corresponding elemental mappings of LGG and LGG-MHS.

5. It is still confused that the grayscale of naked sgRNA in supplementary Fig. 2b was used to calculate the loading efficiency since the RNP was added. Sametime, no electrophoresis result of naked sgRNA, even RNP has been shown in supplementary Fig. 2a.

Response: We apologize for the misspelling of the vertical coordinate labels in Figure 2b. The corresponding error has been corrected in Supplementary Figure 2b in the Revised Supplementary Information. In our experiments, the MH group did not contain Cas9/sgRNA (RNP), and the Cas9/sgRNA mass in the other groups was fixed and the variable was MH, so when MH: Cas9/sgRNA was 0, it meant that only Cas9/sgRNA was present at this time. Figure 2b shows that bare Cas9/sgRNA (*i.e.*, the 0 group in Supplementary Figure 2a) was used as a reference to calculate the grayscale values of undegraded sgRNA after incubation of different ratios of MH:Cas9/sgRNA in 10% serum for 6 h to explore the optimal ratio of MH to protect Cas9/sgRNA from degradation. Regarding the synthesis of Cas9/sgRNA at a fixed ratio was mentioned in the revised manuscript, which reads, “Then, the MH and CRISPR/Cas9 system (mass

ratio 4:1) were incubated at 37 ° C according to the methodology instructions, finally, the integration of MHS nanosystem was constructed.” (Line 599-601, Page 19, Revised Manuscript)

Supplementary Figure 2. (a) Agarose gel electrophoresis and (b) corresponding quantitative analysis of MHS nanoparticles at different MH/sgRNA ratios after incubation with serum (10% volume) for 6 h. Group 0 *i.e.* naked Cas9/sgRNA ($n = 3$). * $P < 0.05$, ** $P < 0.01$, *** $P < 0.001$, **** $P < 0.0001$.

6. The deep sequencing analysis is needed in Fig 3j to avoid that the point mutation is mismatch introduced during PCR process.

Response: Thank you very much for the kind comments and suggestions. The corresponding deep sequencing analysis of Figure 3j have been provided in Supplementary Figure 4 of the Revised Supplementary Information, and the related discussion has been referred in the Revised Manuscript, which reads, “Subsequently, next-generation sequencing (NGS) was further performed to quantify the efficiency of the *IDOI* indel, revealing a genome disruption efficiency was 19.86% and 34.91% for the MHS and MHS + US group, respectively, compared with only 6.35% for the control group (Fig. 3j, k and Supplementary Fig. 4a). Additionally, NGS reveals that the insertion and deletion mutation rates of the *IDOI* motif in the MHS + US group were 1.80% and 16.61%, respectively, while the deletion mutation rate of the *IDOI* motif in the MHS group was 7.79%, which further indicating that US-generated ROS disruption of the lysosomal membrane could significantly improve genome editing efficiency (Supplementary Fig. 4b).” (Line 238-245, Page 8, Revised Manuscript)

j Gene deep sequencing results		
IDOI 20 bp target site	CGCCATGGTGATGTACCCCAAGG	Percentage of mutated sequences
TTCCACACATACGCCATGGTGATGTA-CCCAAGGCCAGGTGT		6.74% (-1)
TTCCACACATACGCCATGGTGATGTGCCCAAGGCCAGGTGT		0.14% (A^G)
TTCCACACATACGCCATGGCGATGTACCCCAAGGCCAGGTGT		0.14% (T^C)
TTCCACACATACGCCATGGTGACGTACCCCAAGGCCAGGTGT		0.13% (T^C)
TTCCACACATACGCCATGGTGTGTACCCCAAGGCCAGGTGT		0.13% (A^G)
TTCCACACATACGCCATGGTGATGTACCCCGAGGCCAGGTGT		0.12% (A^G)

Total		19.86%

k Gene deep sequencing results		
IDOI 20 bp target site	CGCCATGGTGATGTACCCCAAGG	Percentage of mutated sequences
TTCCACACATACGCCATGGTGATGTA-CCCAAGGCCAGGTGT		9.44% (-1)
TTCCACACATACGCCATGGTGATGTA-CCAGGCCAGGTGT		0.94% (-2)
TTCCACACATACGCCATGGTGATGTACCCAGGCCAGGTGT		0.65% (+1)
TTCCACACATACGCCAT-----GGTGT		0.38% (-20)
TTCCACACATAC-----GCCAGGTGT		0.21% (-21)
TTCCACACATACGCCATGGTGATGTACCCCAAGGCCAGGTGC		0.21% (T^C)

Total		34.91%

Fig. 3. (j) Deep sequencing analysis of gene editing in 4T1 cells in the presence of MHS and **(k)** MHS + US.

**Supplementary Figure 4.** (a) Deep sequencing for targeted disruption of *IDO1* locus in control, US only, MH, MH + US, MHS and MHS + US. (b) Nucleotide deletion and insert distribution around the cut site of *IDO1* locus in control, US only, MH, MH + US, MHS and MHS + US.

7. As an important reference, the authors need to add *IDO* to Fig. 4 and give the corresponding discussion.

Response: Thank you very much for the kind comments and suggestions. *IDO* is indeed an important reference, therefore CLSM and WB were used to validate the effect of the CRISPR/Cas9 nanosystem on *IDO1* knockdown. The results are presented in Figure 3f, 3g of Revised Manuscript and Supplementary Fig. 3g, 3h of Revised Supplementary Information, respectively. Furthermore, the related discussion is also mentioned in the revised manuscript, which reads, “To investigate the gene editing efficacy of the MHS nanosystem under US irradiation, Cas9/sgRNA-mediated *IDO1* degradation was examined in 4T1 cells by employing immunofluorescence staining and Western blotting. As the results reveal that *IDO* protein expression levels were significantly reduced in the MHS and MHS + US group, indicating that Cas9/sgRNA effectively mediated the *IDO1* knockdown (Fig. 3f, g and Supplementary Fig. 3g, h).” (Line 226-230, Page 7, Revised Manuscript)

Fig. 3. (f) CLSM images and (g) corresponding mean fluorescence intensity of 4T1 cells treated with various treatments after IFN γ -stimulation, including control, US only, MH, MH + US, MHS and MHS + US, followed by staining with fluorescent anti-*IDO* antibody (red). DAPI was used to stain the nucleus of the cell (blue) ($n = 3$).

Supplementary Figure 3. (g) Western Blot and (h) corresponding quantitative analysis of IFN- γ -stimulated 4T1 cells treated with various treatments. (I = control, II = US only, III = MH, IV = MH + US, V = MHS, VI = MHS + US) ($n = 4$). * $P < 0.05$, ** $P < 0.01$, *** $P < 0.001$, **** $P < 0.0001$.

8. Please add controls to RNaseq-based KEGG analysis in Fig. 5 to explain if the gene expression profiles are conducted by LGG-MHS+US treatment only.

Response: Thank you very much for your kind comments and suggestions. We are very sorry for any misunderstanding you maybe caused due to improper description. The purpose of the RNA sequencing was to explore the role played by LGG in tumor treatment, therefore 4T1 tumor mice models were established and randomly divided into two groups including control and LGG groups (1×10^7 CFU LGG intravenously). Regarding the profile of KEGG analysis in Figure 5, it is the result of a comparative analysis of mice given LGG treatment (LGG group) or mice without any treatment (control group) after RNA sequencing (**Fig 5a-c, Page 36, Revised Manuscript**). The corresponding experimental methods have been described in the Supplementary Information, which reads, “With the approval of the Animal Ethics Committee of Shanghai Tenth People's Hospital, Tongji University School of Medicine, the study was conducted on Balb/c mice ($n = 6$). To establish 4T1 tumor bearing mouse models, Balb/c mice were subcutaneously implanted with 4T1 cells. After the tumor volume reached $\sim 200 \text{ mm}^3$, and then they were randomly divided into two groups ($n = 3$ per group), including the control and LGG groups (intravenous injection of 1×10^7 CFU LGG). At 24 h after the injection, tumor tissues were extracted, followed by nucleic acid extraction and full transcriptome sequencing.” (**Line 140-147, Page 6, Revised Supplementary Information**) In addition, a relevant discussion on clarifying that the

results of RNA sequencing only relate to LGG has provided in the Revised Manuscript, which reads, “Subsequently, six 4T1 tumor-bearing mouse models were established, which were randomly divided into LGG groups and control groups. When the tumor volume reached 200 mm³, RNA sequencing was performed on the tumors in order to investigate the potential biological mechanisms of LGG to promote therapeutic efficacy.” (Line 302-305, Page 10, Revised Manuscript)

Fig. 5 Bacterial hypoxia targeting characterization and bacterial sequencing. (a) Volcano map and (b) Heatmap of genes alteration with or without LGG treatment ($P < 0.05$, $|\text{fold change}| \geq 2$). (c) RNAseq-based KEGG analysis of differential gene expression profiles after LGG treatment.

Reference

1. Deng H, *et al.* Large-pore apertures in a series of metal-organic frameworks. *Science* **336**, 1018-1023 (2012).
2. Lykourinou V, *et al.* Immobilization of mp-11 into a mesoporous metal-organic framework, mp-11@mesomof: A new platform for enzymatic catalysis. *J Am Chem Soc* **133**, 10382-10385 (2011).
3. Mensinger ZL, *et al.* Adsorption of amyloid beta peptide by metal-organic frameworks. *ACS Omega* **5**, 32969-32974 (2020).
4. Kundu T, *et al.* Mechanical downsizing of a gadolinium(iii)-based metal-organic framework for anticancer drug delivery. *Chemistry* **20**, 10514-10518 (2014).
5. Doonan C, *et al.* Metal-organic frameworks at the biointerface: Synthetic strategies and applications. *Acc Chem Res* **50**, 1423-1432 (2017).
6. Vasconcelos IB, *et al.* Cytotoxicity and slow release of the anti-cancer drug doxorubicin from zif-8. *RSC Advances* **2**, (2012).
7. Zhang Y, *et al.* Influence of the 2-methylimidazole/zinc nitrate hexahydrate molar ratio on the synthesis of zeolitic imidazolate framework-8 crystals at room temperature. *Sci Rep* **8**, 9597 (2018).
8. Liang K, *et al.* Biomimetic mineralization of metal-organic frameworks as protective coatings for biomacromolecules. *Nat Commun* **6**, 7240 (2015).

REVIEWERS' COMMENTS

Reviewer #1 (Remarks to the Author):

The authors had addressed my comments

Reviewer #5 (Remarks to the Author):

The manuscript has been carefully revised with additional data and explanation. Based on their responsive letter, we still have some questions and suggestions. Please find the specific ones below. With appropriate point-by-point answer, the manuscript should be acceptable.

1. The authors have given a very detailed description about the interaction between different ingredients in MHS. The in-situ encapsulation of HMME in ZIF-8 is quite clear now, but the internalization of CRISPR/Cas9 is still not convincing. The authors mentions that the pores size of MH (3.3482 nm) is sufficient for the penetration of Cas9/sgRNA more than one time. According to previous study (10.1016/j.addr.2019.11.005), the size of CRISPR/Cas9 RNP complex (10 nm) is much bigger than that in MH.

2. In Fig. 3j, the actual editing efficiency of MHS should be less than 19.80% since the same editing sequence can be found in the control group. It's better to remove these data and calculate it again when doing the statistical editing efficiency. In addition, the IDO1 locus is hard to read in Supplementary Fig. 4a and please add higher resolution pictures.

Reviewer #1 (Remarks to the Author)

The authors had addressed my comments

Response: We greatly appreciated your acknowledgement of this study.

Reviewer #5 (Remarks to the Author)

The manuscript has been carefully revised with additional data and explanation. Based on their responsive letter, we still have some questions and suggestions. Please find the specific ones below. With appropriate point-by-point answer, the manuscript should be acceptable.

Response: Thank you for your positive comments and recommendations. Your concerns have been addressed point by point, please find the following detailed responses to comments and suggestions.

1. The authors have given a very detailed description about the interaction between different ingredients in MHS. The in-situ encapsulation of HMME in ZIF-8 is quite clear now, but the internalization of CRISPR/Cas9 is still not convincing. The authors mentions that the pores size of MH (3.3482 nm) is sufficient for the penetration of Cas9/sgRNA more than one time. According to previous study (10.1016/j.addr.2019.11.005), the size of CRISPR/Cas9 RNP complex (10 nm) is much bigger than that in MH.

Response: Thank you for your kind reminder, which helped us to improve the quality of our manuscript. Indeed, we measured an average pore size of 3.3482 nm for MH, which is smaller than the average size of the previously reported CRISPR/Cas9 RNP complex (10 nm). However, ZIF-8 has been shown to be a porous material with mesopores ranging from 2 to 50 nm (10.1038/ncomms8240), while the BET data in Supplementary Fig. 2c showed that MH has some mesopores larger than 10 nm. Both the mesopore and the total pore volume of MH decreased after loading RNP (**Fig. 2b of Manuscript**), suggesting that some CRISPR/Cas9 is internalized into mesopores larger than 10 nm. However, due to the small number of mesopores (>10nm), all the

CRISPR/Cas9 internalized into the inner part of MH was less, and CRISPR/Cas9 was mainly loaded on the surface of MH by grafting. Therefore, based on your kind reminder, a corresponding description has been added to the revision, which reads, “Due to the pore size limitation, only a relatively small amount of CRISPR/Cas9 has been internalized in the mesopores larger than 10 nm of the MH, while most of it will be grafted on the surface of the MH.” (Line 130-133, Page 4-5, Revised Manuscript)

2. In Fig. 3j, the actual editing efficiency of MHS should be less than 19.80% since the same editing sequence can be found in the control group. It's better to remove these data and calculate it again when doing the statistical editing efficiency. In addition, the *IDO1* locus is hard to read in Supplementary Fig. 4a and please add higher resolution pictures.

Response: Thank you for your kind comments. The gene editing efficiency of MHS and MHS + US in Fig. 3j-k has been recalculated based on your professional advice and applying appropriate statistics approach. The gene editing efficiency of MHS and MHS + US after recalculation was 15.06% and 29.19% (Fig. 3j, k, Revised Manuscript)

In addition, Supplementary Fig. 4a has been redrawn based on the original data (Supplementary Fig. 4a, Revised Supplementary Information)

Gene deep sequencing results			Gene deep sequencing results		
IDO1 20 bp target site	CGCCATGGTGATGTACCCAGGG	Percentage of mutated sequences	IDO1 20 bp target site	CGCCATGGTGATGTACCCAGGG	Percentage of mutated sequences
TTCCACACATACGCCATGGTGATGTA-CCCAGGGCCAGGTGT		6.74% (-1)	TTCCACACATACGCCATGGTGATGTA-CCCAGGGCCAGGTGT		9.44% (-1)
TTCCGCACATACGCCATGGTGATGTACCCAGGGCCAGGTGT		0.16% (A=>G)	TTCCACACATACGCCATGGTGATGTA--CCAAGGGCCAGGTGT		0.94% (-2)
TTCCACACATACGCCATGGTGATGTACCCAGGGCCAGGTGT		0.13% (T=>C)	TTCCACACATACGCCATGGTGATGTACCCAAGGGCCAGGTGT		0.65% (+1)
TTCCACACATACGCCATGGTGATGTACCCAGGGCCAGGTGT		0.13% (T=>C)	TTCCACACATACGCCAT-----GGTGT		0.38% (-20)
TTCCACACATACGCCATGGTGATGTACCCAGGGCCAGGTGT		0.12% (A=>G)	TTCCACACATAC-----GCCAGGTGT		0.21% (-21)
TTCCACACATACGCCATGGTGATGTACCCAGGGCCAGGTGT		0.12% (T=>C)	TTCCACACATACGCCATGGTGACGTACCCCAAGGGCCAGGTGT		0.21% (T=>C)
Total		15.06%	Total		29.19%

Fig. 3. (j) Deep sequencing analysis of gene editing in 4T1 cells in the presence of MHS and (k) MHS + US.